# Sex and smoking bias in the selection of somatic mutations in human bladder

Ferriol Calvet[1,2,3,6], Raquel Blanco Martinez-Illescas[1,2,3,6], Ferran Muiños[1,2], Maria Tretiakova[4], Elena S. Latorre-Esteves[4], Jeanne Fredrickson[4], Maria Andrianova[1], Stefano Pellegrini[1,2,3], Axel Rosendahl Huber[1], Joan Enric Ramis-Zaldivar[1,2], Shuyi Charlotte An[4], Elana Thieme[4], Brendan F. Kohrn[4], Miguel L. Grau[1], Abel Gonzalez-Perez[1,2,3,7], Nuria Lopez-Bigas[1,2,3,5,7 ✉] & Rosa Ana Risques[4,7 ✉]

Men are at higher risk of several cancer types than women[1]. For bladder cancer the risk is four times higher for reasons that are not clear[2]. Smoking is also a principal risk factor for several tumour types, including bladder cancer[3]. As tumourigenesis is driven by somatic mutations, we wondered whether the landscape of clones in the normal bladder differs by sex and smoking history. Using ultradeep duplex DNA sequencing (approximately 5,000×), we identified thousands of clonal driver mutations in 16 genes across 79 normal bladder samples from 45 people. Men had significantly more truncating driver mutations in *RBM10*, *CDKN1A* and *ARID1A* than women, despite similar levels of non-protein-affecting mutations. This result indicates stronger positive selection on driver truncating mutations in these genes in the male urothelium. We also found activating *TERT* promoter mutations driving clonal expansions in the normal bladder that were associated strongly with age and smoking. These findings indicate that bladder cancer risk factors, such as sex and smoking, shape the clonal landscape of the normal urothelium. The high number of mutations identified by this approach offers a new strategy to study the functional effect of thousands of mutations in vivo—natural saturation mutagenesis—that can be extended to other human tissues.

In the interplay between mutagenesis and selection, human somatic tissues evolve as mosaics of competing clones driven by mutations, many of which affect cancer genes[4–12]. Although most of these clones do not result in cancer, in some instances they constitute the first step of the evolutionary trajectory towards malignant tumours. Therefore, characterizing the clonal landscape of human tissues can provide a path to understanding the mechanisms of cancer formation and cancer risk. In the case of the bladder, men and people with smoking history have an increased risk of developing cancer, independently of other risk factors[1,2] (Supplementary Note 1). The reasons for the sex bias, and whether it is also present in the clonal landscape of the normal urothelium, are not known. The role of smoking—whether purely mutagenic or also as promoter of mutant clones—is not well understood either.

The detection of mutant clones in normal urothelium, as in other normal tissues, is challenging because clones are small and escape detection by conventional bulk sequencing methods. Here we directly probed large mixtures of clones from epithelial brushes of normal bladder using ultradeep DNA duplex sequencing (around 5,000× per sample, amounting to approximately 400,000 haploid genomes in aggregate)[13,14]. We focused on the detection of clones driven by mutations in 16 genes known to be under positive selection in bladder urothelium[10,15], and/or mutated frequently in bladder carcinomas[16,17], including the telomerase gene (*TERT*) promoter[18]. We then used the magnitude of positive selection for each gene at the sample level to regress out the effect of sex and smoking on their clonal landscape.

We also considered that the capacity to identify the driver mutations of thousands of clones in a human tissue could increase markedly our ability to uncover the functional effect of all mutations in genes relevant in tumourigenesis. Identifying all potential driver mutations in a cancer gene is key to advancing personalized cancer medicine. The selective advantage provided by spontaneous mutations in millions of cells is tested in the interplay between mutagenesis and selection in normal human tissues. We should then be able to probe the results of these natural experiments by sequencing a large enough number of cells in a somatic tissue. We explore this postulate with data from ultradeep sequencing of normal bladder. As a result, we have uncovered a powerful tool with which to quantify positive selection at site resolution.

## Mutation landscape of normal urothelium

We collected cells from either one or two regions—the upper (dome) and lower (trigone)—of the bladder urothelium from 45 deceased donors (79 samples) through a brushing of relatively large surface areas (roughly 2 cm²) at autopsy (Fig. 1a and Supplementary Note 2).

[1]Institute for Research in Biomedicine (IRB Barcelona), The Barcelona Institute of Science and Technology, Barcelona, Spain. [2]Centro de Investigación Biomédica en Red en Cáncer (CIBERONC), Instituto de Salud Carlos III, Madrid, Spain. [3]Department of Medicine and Life Sciences, Universitat Pompeu Fabra, Barcelona, Spain. [4]Department of Laboratory Medicine and Pathology, University of Washington, Seattle, WA, USA. [5]Institució Catalana de Recerca i Estudis Avançats (ICREA), Barcelona, Spain. [6]These authors contributed equally: Ferriol Calvet, Raquel Blanco Martinez-Illescas. [7]These authors jointly supervised this work: Abel Gonzalez-Perez, Nuria Lopez-Bigas, Rosa Ana Risques. ✉e-mail: nuria.lopez@irbbarcelona.org; rrisques@uw.edu

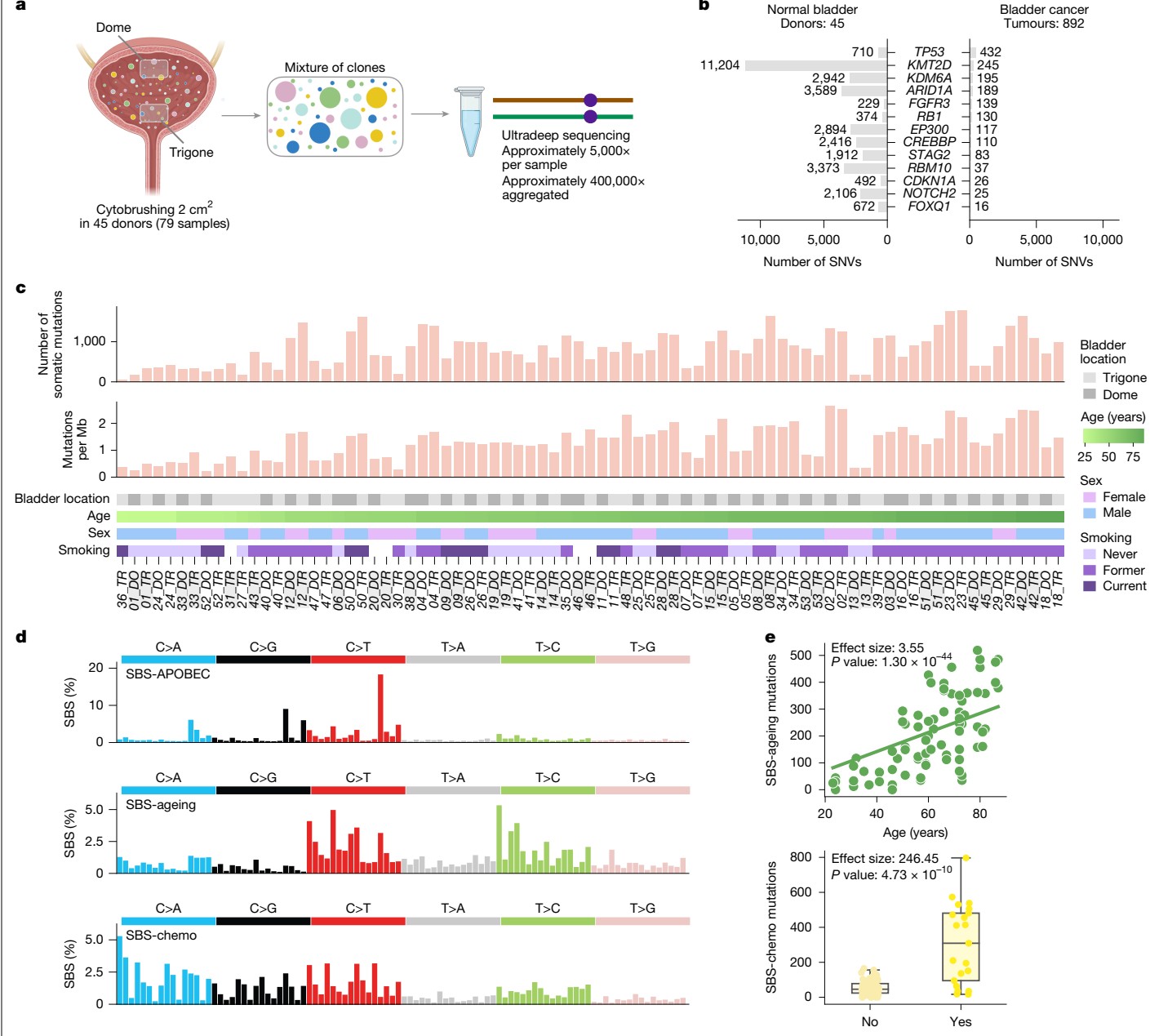

**Fig. 1 | Ultradeep DNA sequencing of normal urothelium targeting driver genes shows thousands of mutations. a**, Schematic representation of sampling and duplex DNA sequencing of polyclonal epithelial brushes from normal bladders. **b**, Number of SNVs detected in a panel of selected genes in the normal bladder in this study and comparison with the number of mutations detected in 892 tumours from bladder cancer genomics studies, obtained from intOGen[24]. **c**, Number and density of somatic mutations (SNVs, MNVs and indels up to 100 base pairs) identified in 79 samples of normal urothelium obtained from 45 donors. Bladder location (dome or trigone), age, sex and smoking history of donors are shown below the bar plots. **d**, Trinucleotide substitution profiles of the mutational signatures identified across this cohort of normal urothelium samples through de novo extraction. **e**, Top, scatter plot representing the relationship between the activity (number of mutations contributed) of SBS-ageing in samples and the age of donors; effect size, regression line and *P* value of a univariate mixed-effects linear model. Bottom, box plot representing the activity of SBS-chemo in donors exposed or not exposed to chemo/radiotherapy; effect size and *P* value of a univariate mixed-effects linear model. The boxplots show the quartiles with whiskers extending to the highest and lowest data points within 1.5 times the interquartile range; *N* = 79 samples for all plots in the panel. Mb, megabase. BioRender was used to create panel **a** (https://BioRender.com/fgnnet9).

Clinical data, including exposure to known or suspected bladder cancer risk factors such as tobacco smoking or chemotherapy, was available for most donors (Supplementary Table 1). We used ultradeep DNA duplex sequencing[13,14] (average depth ranging between 1,236 and 9,303 across samples, with a median of 5,164; Supplementary Note 2) and a newly developed and carefully calibrated computational pipeline to identify somatic mutations with high sensitivity and specificity in these polyclonal samples (Extended Data Fig. 1a–c and Supplementary Note 3). The accuracy of this technology allows the reliable detection of mutations that are present in a single DNA molecule (Extended Data Fig. 2a–c and Supplementary Note 4).

To investigate how bladder cancer risk factors might change the clonal landscape of the urothelium, we focused on 15 genes identified previously to be under positive selection in normal bladder[10,15]

and/or known to frequently drive bladder tumours[16,17] and the promoter of the gene encoding TERT, which is mutated in around 70% of bladder tumours[18–23]. This amounted to 111,876 base pairs of genomic DNA comprising the exonic regions and neighbouring intronic sites of the 15 genes (in full or selected fragments) and a region of the TERT promoter (hereinafter for simplicity, genes; Supplementary Tables 2, 3 and 4). Across samples, we identified a total of 64,278 mutations, between 54 and 1,785 (median 774) mutations per sample, including single nucleotide variants (SNVs), multiple nucleotide variants (MNVs) and indels up to 100 base pairs (Supplementary Table 5). This total number of mutations is approximately 16-fold higher than the number identified in the same genes across 892 bladder tumours (intOGen)[24] sequenced over a decade of cancer genomics (Fig. 1b,c).

The overall non-protein-affecting mutation density—mutations per megabase sequenced—was associated, as expected, with age and the exposure to certain mutagenic factors, such as several chemotherapies/radiotherapy (Supplementary Note 5). Using two orthogonal signature extraction methods[25–27], and supported by the detection of a large number of non-protein-affecting mutations, we discerned the activity of three main mutational signatures (Supplementary Note 5 and Extended Data Fig. 3a–d). These captured the activity of the APOBEC family of cytidine deaminases[28] (SBS-APOBEC; Fig. 1d), age-related mutagenesis[29] (SBS-ageing; Fig. 1d,e) and an unknown process (harder to reduce to a linear combination of mutational signatures in the COSMIC catalogue[30]) that seemed to be associated with exposure to chemotherapy (SBS-chemo; Fig. 1d,e and Supplementary Note 5). Overall, the landscape of mutational signatures and the number of mutations per megabase sequenced was similar to that identified in a previous study of normal urothelium[10] (Extended Data Fig. 2b,c), with the exception of a signature reported to be associated with tobacco smoking. Mutational signatures were correlated highly between the dome and trigone of the same donor (Supplementary Note 5).

## Driver mutations in normal bladder

The 16 genes included in the panel had been demonstrated previously to be under positive selection in normal urothelium or mutated frequently in bladder tumours. Thus, we first examined whether their mutations were positively selected in the samples included in this cohort. To this end, we used four complementary computational methods (Supplementary Note 6) on the basis of comparing features of the pattern of mutations observed in a gene with that expected assuming neutral evolution[31,32].

First, we estimated the expected number of protein-affecting mutations with different impact, that is, missense or truncating (including nonsense and essential splice-site mutations) on the basis of the number of observed synonymous mutations in each gene across samples and in each sample using a newly developed method called Omega (Fig. 2a and Supplementary Note 6). This calculation takes into account the number of sites allowing amino acid changes of each type in the protein, the frequency of trinucleotide changes observed across samples and the depth at which each individual genomic site has been sequenced. Then, using a dN/dS approach[33], Omega computes the degree of positive selection on missense (dN/dS missense) or truncating (dN/dS truncating) mutations. These values represent the excess of missense or truncating mutations—driver mutations—among those observed in the gene in the sample (Fig. 2b). In addition to the excess of truncating and missense mutations, we quantified three other signals of selection: clustering of mutations in the three-dimensional structure of the protein, functional impact bias and excess of frameshift indels (Supplementary Note 6 and Extended Data Fig. 4a). These metrics indicated that RBM10, KDM6A, STAG2, KMT2D, ARID1A, CDKN1A, TP53, EP300, NOTCH2, CREBBP, FOXQ1, KMT2C and RB1—13 of the genes included in this study—are under positive selection, driving clonal

expansion across the 79 normal urothelium samples (Extended Data Fig. 4b,c). We verified that the same is true for PIK3CA through analysis of the mutations in a hotspot affecting the histidine at position 1,047 (Extended Data Fig. 4d).

This analysis produced two additional findings. First, we discovered that truncating (both SNVs and frameshift indels) mutations of FGFR3 seem to be under negative selection (Fig. 2c and Extended Data Fig. 4b,c). Second, we observed 85 SNVs affecting the TERT promoter (across 42 samples; Fig. 2d). The TERT gene encodes telomerase, and its promoter includes two hotspots that result in the creation of two de novo ETS transcription factor binding sites[34] and are mutated frequently in human tumours[18,19,22,23]. To formally compute the strength of positive selection on the TERT promoter in our cohort, we dichotomized the mutations on the basis of whether or not they have been observed several times across 8,136 tumour whole genomes (Supplementary Note 6; Methods), that is, using their presence in tumours as a surrogate of their capacity to activate the expression of TERT. We could thus calculate a modified dN/dS value for the TERT promoter (dN/dS pTERT; Supplementary Note 6). Although we expected fewer than one activating mutation, we actually observed 56, implying that at least 55 (98.2%) of these mutations detected in the TERT promoter are drivers of clonal expansion, probably by activating the expression of telomerase in normal urothelial cells.

We next computed positive selection separately for each sample in the cohort (Extended Data Fig. 5a). The high depth of sequencing in the study guarantees that these sample-level dN/dS values can be estimated accurately and robustly (Supplementary Notes 7 and 8). The number of clones driven by mutations detected in each of the 13 genes under positive selection varies between zero and several hundreds across the samples analysed from each donor (Fig. 3a). Despite the abundance of driver mutations, the relative size of the clones is very small and thus the fraction of the urothelium covered by mutant clones of these genes is modest (less than 1%) in most cases (Extended Data Fig. 5b). Some activating TERT promoter mutations reach relatively high values of variant allele frequency (VAF), probably due to both clonal expansion and convergent evolution.

We also found that the patterns of positive selection of the dome and trigone from the same donor were strikingly similar across the entire cohort (Extended Data Figs. 5c and 6a–c and Supplementary Note 5), indicating that the evolutionary dynamics and selective forces in the two regions are equivalent. Thus, a brushing of cells from approximately 2 cm² of the urothelium provides a good representation of the clonal structure of the entire tissue.

In summary, applying a dN/dS-based approach to mutations detected across urothelial samples through ultradeep sequencing, we demonstrated that a large percentage of mutations in the profiled genes are drivers, indicating pervasive positive selection; we uncovered negative selection for truncating mutations in FGFR3; we showed positive selection for activating mutations in the TERT promoter and we made robust estimations of positive selection and number of driver mutations per gene in each sample, thus enabling the study of inter-individual differences in clonal selection (Fig. 3a, Extended Data Fig. 7a,b and Supplementary Table 6).

## Sex bias in the bladder clonal landscape

To identify a potential sex bias of the urothelial clonal landscape, we resorted to regressions that incorporated meaningful covariates, in particular age and history of smoking[1,35–37] (Supplementary Note 1). Indeed, part of this heterogeneity across donors is explained by age. The density of protein-affecting mutations increases at a higher pace than that of non-protein-affecting mutations, indicating that clones driven by mutations of the 13 genes under positive selection expand with age, as reported for other normal tissues[7–9,11,12,38,39] (Fig. 3b,c and Supplementary Note 9). In agreement with this, the dN/dS values of

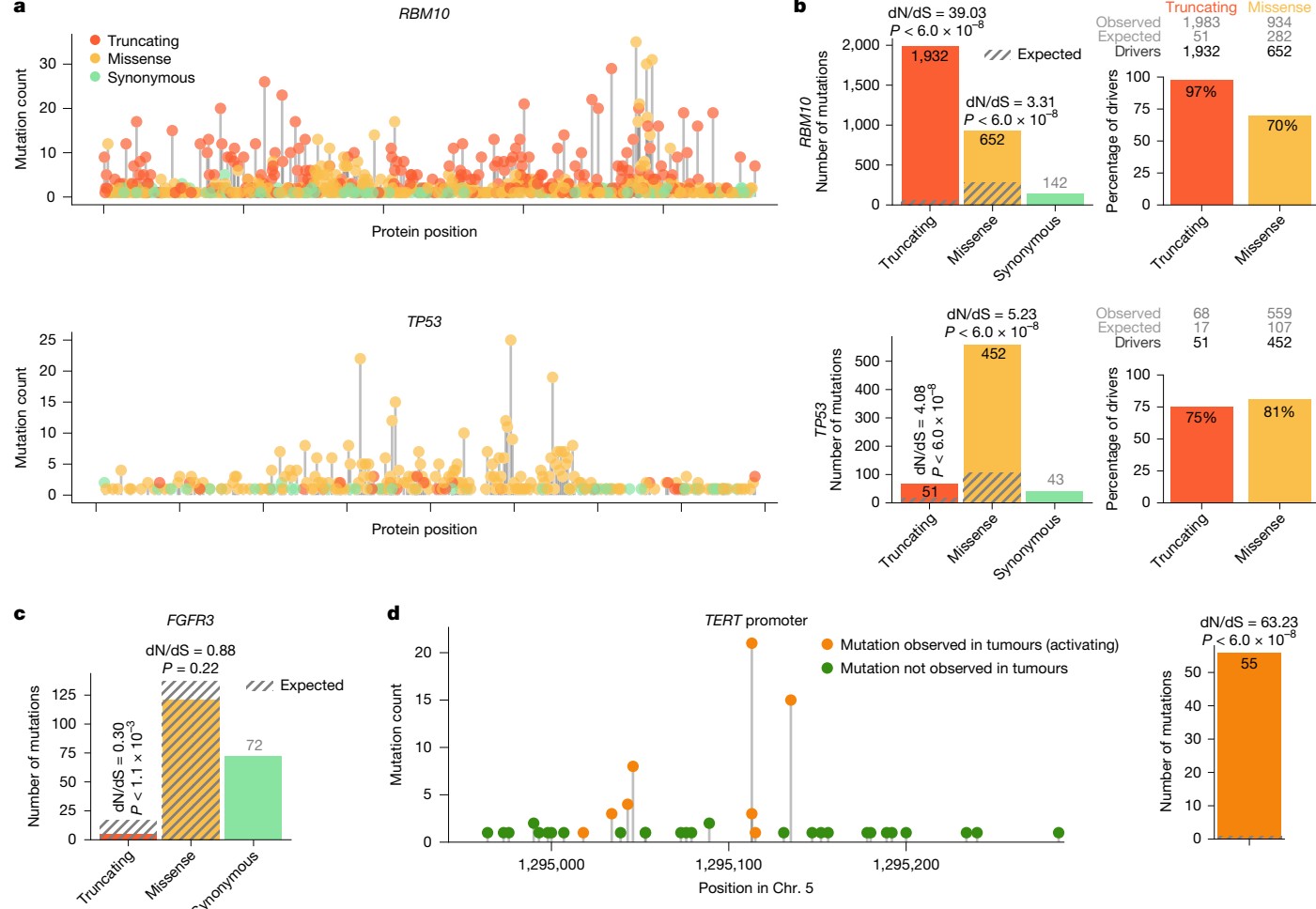

**Fig. 2 | Computing positive selection in the normal urothelium.**
**a**, Distribution of truncating, missense and synonymous somatic mutations along the coding regions of *RBM10* and *TP53*. Each needle represents the number of samples with mutations with one of the three consequences occurring on an amino acid residue. **b**, Magnitude of positive selection on truncating and missense mutations in *RBM10* and *TP53*. Left, magnitude of positive selection indicated as the dN/dS ratio on the basis of the number of observed and expected mutations of each type. Right, percentage of driver mutations (observed in excess of neutrality) among truncating and missense in *RBM10* and *TP53*. The number of observed, expected and estimated driver mutations are indicated for each gene. **c**, Magnitude of selection on truncating and missense mutations in *FGFR3*. dN/dS below 1 indicates negative selection. **d**, Magnitude of positive

selection of activating mutations in the *TERT* promoter. Left, distribution of somatic mutations along the sequence of the *TERT* promoter colour coded according to whether they are activating (seen in tumours). Right, bar representing the magnitude of positive selection on activating *TERT* promoter mutations. In dN/dS bar plots (**b**–**d**), the shaded segment in each bar represents the number of truncating or missense (or activating in the case of *TERT*) mutations expected under neutrality. The numbers above the bar detail the value of dN/dS for each type of mutation. Numbers inside bars represent the mutations in excess over the expectation, that is, the drivers. The *P* values in **b**–**d** were calculated using a dN/dS approach (Omega) described in Supplementary Note 6.

missense and truncating mutations of these genes are also associated significantly with age (Fig. 3c and Supplementary Table 7).

Using multivariate regressions, we determined that the values of dN/dS truncating of *RBM10*, *CDKN1A* and *ARID1A* are increased significantly in the urothelium of men compared with women, independently of age, smoking history, alcohol drinking history, body mass index (BMI) and exposure to chemotherapy (Fig. 3d,e, Extended Data Fig. 8a,b, Supplementary Table 7 and Supplementary Notes 9 and 10). In addition, the dN/dS truncating of *STAG2* shows a tendency, although not significant, towards higher values in men. Two of these genes (*RBM10* and *STAG2*) are in the X chromosome, whereas *CDKN1A* and *ARID1A* are in autosomes. *KDM6A* is another X chromosome gene that has been described previously with a higher number of mutations in female bladder tumours[40]. A closer inspection of X chromosome genes confirms the expectation of higher coverage across women for all three, although only *RBM10* (and *STAG2* to a lesser extent) shows a sex bias of dN/dS truncating (Supplementary Note 9). This implies that the growth

of clones driven by mutations under positive selection in these genes is different in the urothelium of men and women (Supplementary Notes 9 and 10). The number of bladder tumours with truncating mutations in *RBM10* and *CDKN1A* is also significantly higher across men than women (Extended Data Fig. 8c,d and Supplementary Note 11). In summary, clonal selection in normal bladder differs between men and women, mirrored by an increased number of mutations in *RBM10* and *CDKN1A* in male bladder tumours.

## *TERT* clones associate with age and smoking

We discovered that activating mutations in the promoter of *TERT* (that is, those observed at least twice in tumours[18–23,34]), were much more recurrent among older donors in the cohort. Specifically, all *TERT* promoter-activating mutations in the cohort, except one, occurred across donors older than 55 years (Fig. 4a,b). Furthermore, in this group, those with a history of smoking exhibited a significantly higher rate of

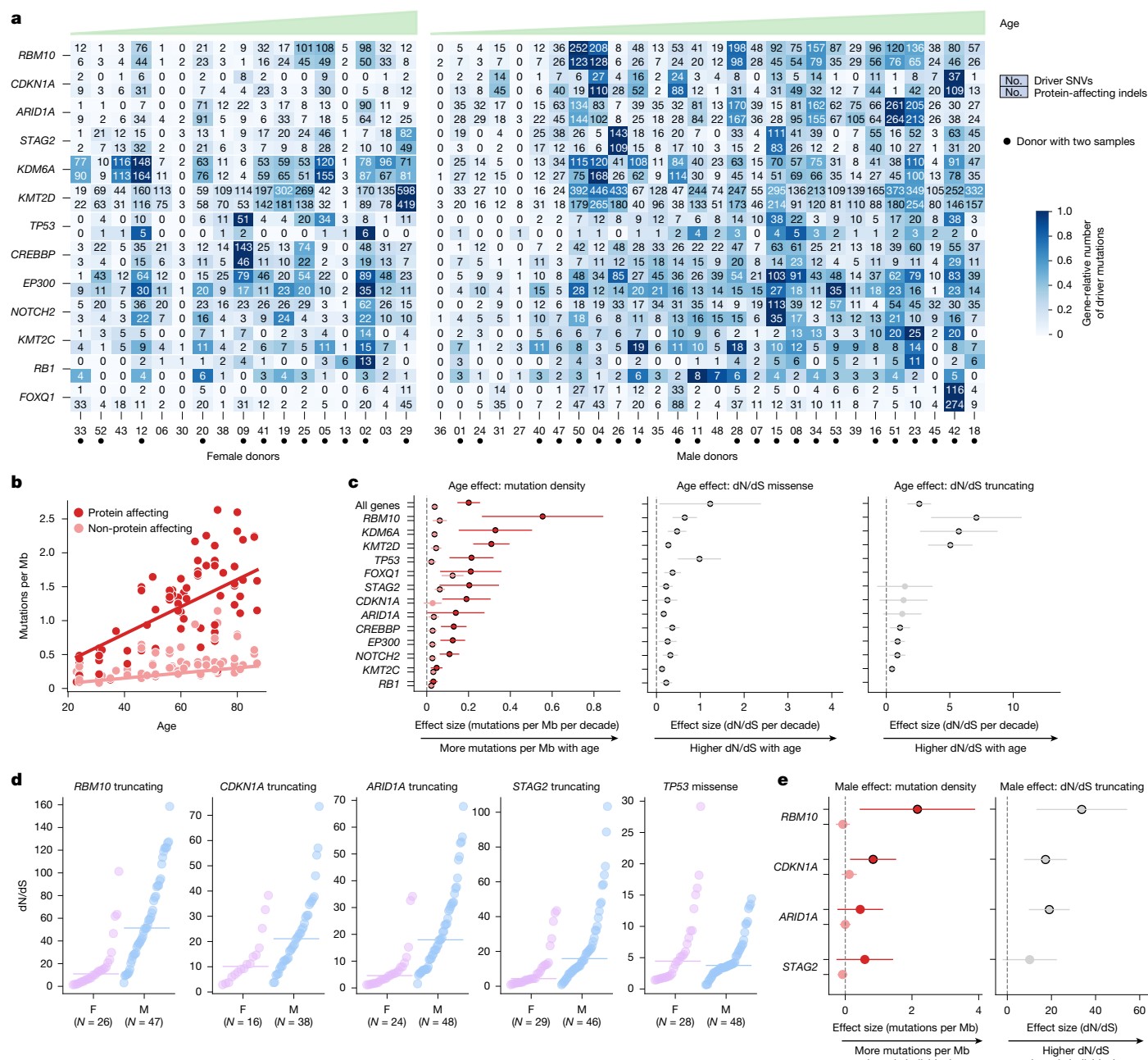

**Fig. 3 | Sex bias of the clonal landscape of normal urothelium. a**, Heatmaps with number of driver SNVs (top) and the total number of protein-affecting indels (bottom) in 13 genes across female and male donors. For donors with dome and trigone samples (marked with a dot), the number of driver SNVs in both samples has been added. Donors are sorted by age in ascending order. **b**, Relationship between age and the density (per Mb) of protein-affecting and non-protein-affecting mutations. **c**, Association of age with the protein-affecting (top coefficient) and non-protein-affecting (bottom coefficient) mutation density (left), and the magnitude of positive selection on missense (centre) and truncating (right) mutations of 13 genes (gene–mutation consequence combinations for which we could calculate dN/dS at the level of sample for at least 80% of samples were included in the calculation). **d**, Distribution of dN/dS truncating values of *RBM10*, *CDKN1A*, *ARID1A* and *STAG2* and dN/dS missense values of *TP53* among male and female donors. Horizontal lines denote the median. **e**, Association of sex with the protein-affecting (top coefficient) and

non-protein-affecting (bottom coefficient) mutation density (left), and the magnitude of positive selection on truncating mutations in *RBM10*, *CDKN1A*, *ARID1A* and *STAG2* (right). The plots represent the results of multivariate regressions accounting for age, smoking history, alcohol drinking history, BMI and exposure to chemotherapy. In all regressions (**c** and **e**), multivariate linear mixed-effects models (accounting for sex, smoking history, alcohol drinking history, BMI and exposure to chemotherapy) were used to account for the presence of several samples of the same donor, and these models yielded the effect size and *P* values. Circles represent the point estimate of the effect size of the regressions; horizontal lines represent 95% confidence intervals. Circles with dark outer circumference denote significant associations (false discovery rate (FDR) threshold of 0.2). *N* = 79 samples in **b**, **c** and **e**. All corrected *P* values for these regressions appear in Supplementary Table 7. F, female donors; M, male donors.

*TERT* promoter mutations (Fig. 4a,b). Some of these mutations showed relatively large VAF, probably caused by several expanding clones with these mutations (Extended Data Fig. 8e,f and Supplementary

Note 10). We found that the association between smoking history in older donors and the rate of *TERT* promoter-activating mutations is independent of sex, BMI and exposure to chemotherapy (Fig. 4c), as

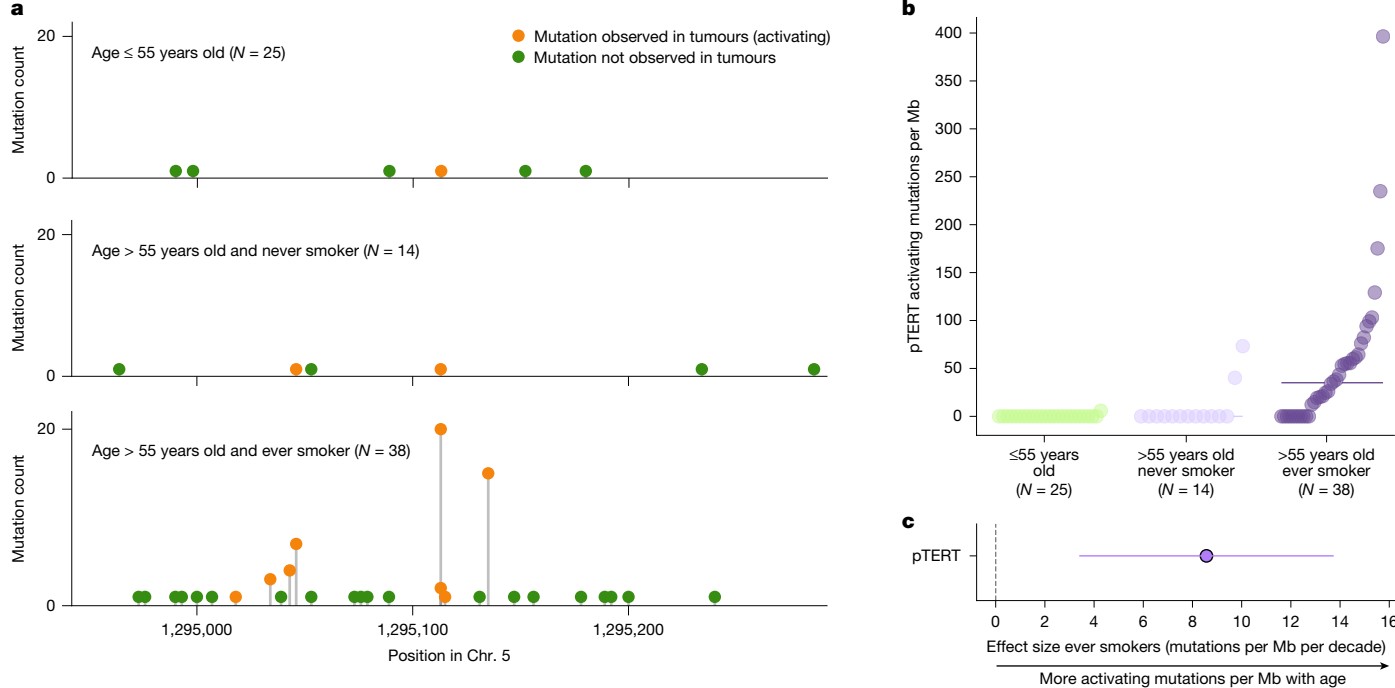

**Fig. 4 | Association of activating mutations in the *TERT* promoter with smoking. a**, Number of activating (observed in tumours) and other (not observed in tumours) mutations in the *TERT* promoter across donors younger than 55 years (top), older than 55 years with no history of smoking (middle) or older than 55 years with a history of smoking (bottom). **b**, Density (mutations per Mb) of activating *TERT* promoter mutations observed in the three groups of donors. The horizontal line denotes the median of the distribution of the frequency of mutations in the group of donors older than 55 years with a history of smoking. The sample size of the three groups was 25 (samples from donors younger than 55), 14 (samples from donors older than 55 with no smoking history)

and 38 (samples from donors older than 55 with smoking history), respectively. The two samples from a donor with unknown smoking history were excluded. **c**, Association between the frequency of activating mutations in the *TERT* promoter and the interaction between age and smoking history. A linear mixed-effects model was used to compute the *P* value. The circle represents the point estimate of the effect size of a multivariate linear mixed-effects regression, and the horizontal line represents 95% confidence intervals. The dark outer circumference denotes a significant association (FDR threshold of 0.2). The corrected *P* value appears in Supplementary Table 7. Number of samples for this analysis is the same as in **a** and **b**.

well as the depth of sequencing. The significant increase of activating *TERT* promoter mutations in smokers contrasts with the absence of association between smoking history and overall mutation density across samples (Supplementary Note 5 and Supplementary Table 7). This result indicates that tobacco carcinogens have a role in promoting the growth of clones with *TERT* promoter mutations. In summary, the rate of *TERT* promoter-activating mutations in the normal urothelium is associated significantly with the interaction of age and a history of smoking. More than 70% of bladder tumours have *TERT* promoter mutations, with a comparable frequency in smokers and non-smokers. In lung tumours, where the frequency of *TERT* promoter mutations is much lower, we find a significantly higher frequency in smokers (2.1%) compared with non-smokers (0.9%) (Supplementary Note 11). Collectively, these results indicate that exposure of the normal urothelium to tobacco carcinogens may increase the risk of bladder cancer through the promotion of clones bearing mutations of the *TERT* promoter.

## Natural human saturation mutagenesis

Experimental saturation mutagenesis (that is, the insertion of all possible mutations in a genomic element in experimental systems) has been used to identify sites or regions in proteins that are key for their structure, function or regulation, to understand their role in disease and improve the design of drugs to target them[41–45]. Recently, we introduced the concept of in silico saturation mutagenesis, exploiting more than 20 years of cancer genomics research on the natural experiments that are human tumours (or clonal expansions in normal blood) to build machine learning models that identify all their potential driver

mutations[46,47]. In contrast to tumours, whose evolutionary history entails the expansion of a single clone, with only one or two mutations observed in a cancer driver gene in most cases, in normal polyclonal samples, several mutations affecting a given gene are detected as drivers of the expansion of several clones. Therefore, sequencing normal human tissues, which can also be considered natural experiments of clonal evolution, can provide a pathway towards natural saturation mutagenesis.

To understand how far along the ultradeep sequencing of roughly 400,000 haploid genomes takes us in the path towards natural saturation mutagenesis, we first calculated the number of mutations observed in each amino acid residue of the protein-coding genes (Fig. 5a). For example, we detected mutations (missense or nonsense) on 230 (58%) of the 394 amino acid residues of *TP53* protein product covered by duplex sequencing (Fig. 5a). In normal urothelium, the frequency of mutations by gene differed from those in bladder cancers (Extended Data Fig. 9a) and the proportion of mutated residues was larger in all genes tested (Extended Data Fig. 9b), demonstrating different selective pressures and the advantage of studying mutations in normal tissue to reach saturation mutagenesis.

Next, we asked how many more genomes would need to be sequenced to observe mutations on all amino acid residues of each gene. To estimate this, we first calculated the probability to observe each mutation in a gene under neutrality, on the basis of the number of synonymous mutations and the trinucleotide mutational probabilities observed in our cohort. Thus, we obtained a theoretical curve describing the fraction of all possible mutations that are expected to be observed at different sequencing depths (theoretical kinetic of natural saturation

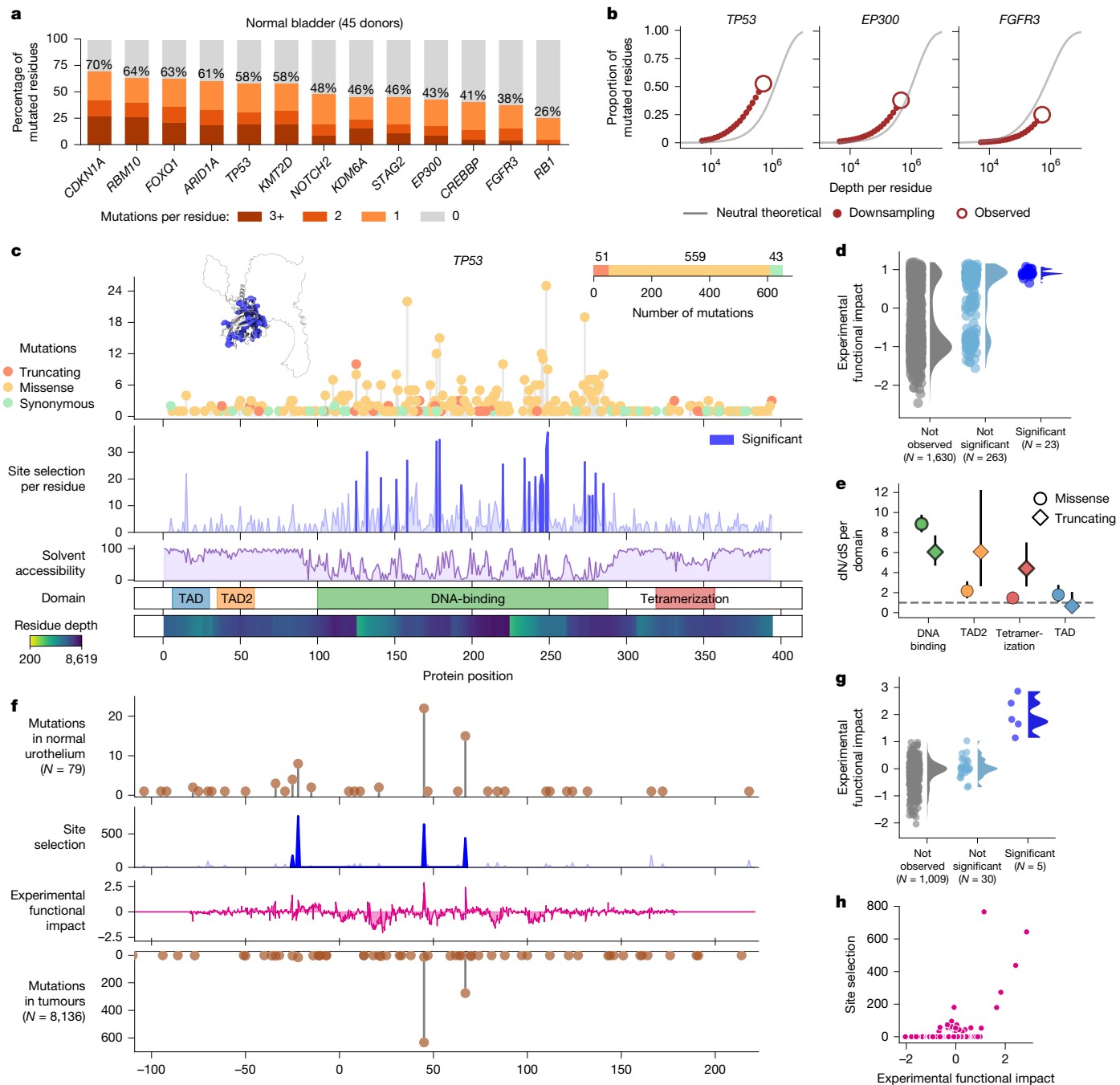

**Fig. 5 | Natural saturation mutagenesis. a**, Percentage of amino acid residues in each gene with zero, one, two or three or more mutations across the 79 samples. **b**, Theoretical and observed kinetic of natural saturation mutagenesis for *TP53*, *EP300* and *FGFR3* as cumulative depth of sequencing increases. Grey line, theoretical kinetic of saturation mutagenesis (assuming no selection). Red circle, saturation achieved across the cohort. Red dashed line, observed kinetic (obtained by downsampling). **c**, Natural saturation mutagenesis of *TP53* in normal bladder urothelium. From top to bottom, distribution of truncating, missense and synonymous mutations along the coding sequence of the gene; site selection computed for each amino acid residue of *TP53* protein product; solvent accessibility along the protein sequence; *TP53* protein product domains; duplex sequencing depth per amino acid residue. Left top, *TP53* protein product three-dimensional structure with significant site selection of residues highlighted in blue. **d**, Experimental functional impact[51] of *TP53* mutations not observed, observed, or observed with significant site selection across the

79 samples. Only mutations with experimental functional impact reported in ref. 51 are included. **e**, dN/dS truncating and dN/dS missense values for each domain of *TP53* protein product. The vertical lines represent the 95% confidence intervals of the dN/dS estimate. Solid border represents significant dN/dS values (*P* value < 0.05) according to Omega (Supplementary Note 6); *N* = 79 samples. **f**, Natural saturation mutagenesis of the *TERT* promoter. From top to bottom, distribution of mutations; site selection computed for observed mutations; experimental functional impact values of mutations in the *TERT* promoter according to ref. 55; distribution of mutations observed in the *TERT* promoter across 8,136 tumours (Supplementary Note 6). **g**, Experimental functional impact[55] of mutations not observed, observed or observed in the *TERT* promoter with significant site selection across the 79 samples. **h**, Relationship between site selection and experimental functional impact value[55] of all mutations observed in the *TERT* promoter.

mutagenesis; Fig. 5b (continuous curve), Extended Data Fig. 9c and Supplementary Note 12). The shape of this curve is determined by the difference in probability across mutations, governed by their trinucleotide contexts, with some mutations much more likely to occur than others. For example, for *TP53*, with the accumulated depth provided by the entire cohort (approximately 500,000×), only on the basis of neutral mutagenesis, we expected to have observed around 26% of all possible mutations. In theory, being able to detect every possible mutation in the gene would require an accumulated depth of around $10^7$, assuming an aggregated mutation density and mutational signatures as the samples in the study.

Nevertheless, the observation of mutations in normal urothelium is not neutral, but instead is strongly affected by selection. To obtain the actual curve of the fraction of observed mutations depending on the number of sequenced genomes (observed kinetic of natural saturation mutagenesis), we sub-sampled the number of mutations observed in each gene across the cohort, respecting their VAF but reducing the depth of sequencing at each position (dots in Fig. 5b and Extended Data Fig. 9c). The divergence in the shape of the two curves shows the importance of selection in shaping the probability to observe a given fraction of mutations in a gene. For example, as missense and truncating *TP53* mutations are under positive selection, we actually observed more mutations than predicted by the theoretical kinetic. For several genes, we observed this faster accumulation of mutations than expected under neutral mutagenesis due to positive selection (Extended Data Fig. 9c). The curve for *FGFR3* falls below the theoretical curve constructed under purely neutral mutagenesis (Fig. 5b), which supports the notion that *FGFR3* mutations are under negative selection in the normal urothelium.

Approaching natural saturation mutagenesis provides an opportunity to calculate the strength of positive selection on mutations affecting each amino acid residue (Supplementary Note 6), and for other within-gene structures such as exons or protein domains (Fig. 5c, Extended Data Figs. 9d,e, 10a and Supplementary Fig. 1). In the case of *TP53*, most sites under significant selection are in the p53 DNA-binding domain[48] (Fig. 5c), and are observed more frequently across tumours[24,49,50] (Extended Data Fig. 10b), more likely to be identified as cancer drivers[46] (Extended Data Fig. 10b) and tend to score higher on an experimental saturation mutagenesis assay[51] (Fig. 5d), as described previously in duplex sequencing analysis of *TP53* in other normal tissues[52–54]. In this and other genes that act as tumour suppressors, the mutations with significant site selection appear in buried areas of the protein (Fig. 5c, Extended Data Fig. 10c,d and Supplementary Fig. 2).

The calculation of dN/dS at the level of domains shows that only the p53 DNA-binding domain exhibits a significant excess of missense mutations (Fig. 5e). By contrast, *TP53* dN/dS truncating values present a more homogeneous distribution across exons and domains (Fig. 5e and Supplementary Fig. 2). The same is true for other genes such as *EP300* and *CREBBP* where only the histone acetyltransferase domain (HAT-KAT11) shows significant dN/dS missense values (Extended Data Fig. 9d and Supplementary Figs. 1 and 2). For *RBM10*, with a much stronger signal of truncating mutations, residues with the highest site selection are distributed more uniformly than in *TP53* (Supplementary Fig. 2). Genes affected mostly by truncating driver mutations, such as *STAG2* and *RBM10*, show a higher dN/dS truncating than dN/dS missense values for virtually all exons, except the first and last (Supplementary Fig. 2). Despite the fact that almost all genes sequenced are tumour suppressors, in all cases, a few SNVs exhibit much stronger site selection than the rest (Extended Data Fig. 10a and Supplementary Fig. 2). Whereas *FGFR3* mutations showed negative selection overall (Fig. 2c), mutations affecting a single residue (G380) were positively selected in normal urothelium. This residue did not correspond to the three most common *FGFR3* hotspots observed in bladder cancer (Supplementary Fig. 3).

*TERT* promoter mutations with significant site selection in normal urothelium appear more frequently across tumours (Extended Data Fig. 10e), and present overall higher experimental functional impact values[55] (Fig. 5f–h). In summary, natural saturation mutagenesis, brought about by the possibility of identifying mutations at extremely low VAF through ultradeep sequencing of normal tissues, opens up the possibility of directly assessing the functional impact of mutations affecting different genes in human tissues.

## Discussion

The reasons behind the wide heterogeneity in clonal landscape observed across normal human tissues from different people[7–12,38,39] and the potential role of known cancer risk factors are not understood. In this study we demonstrated the suitability of ultradeep DNA duplex sequencing[13,14] (approximately 400,000 haploid genomes) to accurately measure the magnitude of positive selection on each of 16 genes in 79 individual samples (Supplementary Note 6). The high number of mutations identified proved key to showing associations between clonal expansions and bladder cancer risk factors. Specifically, we discovered that the magnitude of positive selection on truncating mutations in *RBM10*, *ARID1A* and *CDKN1A* is significantly higher among men even when adjusting for age, smoking history, alcohol drinking history, and BMI. Importantly, mutations in *RBM10* and *CDKN1A* are also significantly more abundant in bladder cancers of men than of women (Supplementary Note 11). We can speculate that the observed differences between men and women could be related to differences in exposure to internal factors (for example, sex hormones) or extrinsic exposures that disproportionately affect men. Discerning whether the bias towards the expansion of clones bearing mutations in these and other genes is somehow related with the observed increased risk of bladder cancer across men requires further investigation. We also found a high increase of *TERT* promoter mutations among older people with a history of smoking, which may explain at least part of the link between smoking and increased bladder cancer risk[35–37], given the high frequency of *TERT* promoter mutations across bladder tumours[18–23,34]. In this cohort, we do not find a specific smoking-associated mutational signature. The high increase of activating *TERT* promoter mutations in smokers, rather than a general increase in mutation density, points to a non-mutagenic—acting rather as promoter[56–58]—role of tobacco smoking in bladder tumourigenesis. Scaling up these analyses to probe the influence of other bladder cancer risk factors requires the possibility of obtaining samples in a minimally invasive manner, which could be conceivably achieved through urine or lavages[59].

The proposal of a pathway towards the study of natural saturation mutagenesis through ultradeep error-correcting duplex DNA sequencing of large mixtures of clones from normal tissues is a very compelling one. In this study we have provided a strong argument for the plausibility of this approach, paving the way for future studies. We have shown that it is possible to obtain a fine map of the degree of selection acting on mutations at individual sites or regions of a protein throughout the normal development of tissues of people exposed to different external factors. Saturation mutagenesis—a key development towards personalized cancer medicine[60]—through ultradeep sequencing in human tissues, thus provides a cost-effective and direct alternative to complement experimental and in silico approaches. In addition, characterization of the patterns of positive (and negative) selection in different genes and human tissues will be critical to understanding how somatic evolution influences cancer risk.

## Online content

1.  Jackson, S. S. et al. Sex disparities in the incidence of 21 cancer types: quantification of the contribution of risk factors. *Cancer* **128**, 3531–3540 (2022).
2.  Doshi, B., Athans, S. R. & Woloszynska, A. Biological differences underlying sex and gender disparities in bladder cancer: current synopsis and future directions. *Oncogenesis* **12**, 44 (2023).
3.  Jha, P. Avoidable global cancer deaths and total deaths from smoking. *Nat. Rev. Cancer* **9**, 655–664 (2009).
4.  Martincorena, I. Somatic mutation and clonal expansions in human tissues. *Genome Med.* **11**, 35 (2019).
5.  Kakiuchi, N. & Ogawa, S. Clonal expansion in non-cancer tissues. *Nat. Rev. Cancer* **21**, 239–256 (2021).
6.  Martincorena, I. & Campbell, P. J. Somatic mutation in cancer and normal cells. *Science* **349**, 1483–1489 (2015).
7.  Martincorena, I. et al. High burden and pervasive positive selection of somatic mutations in normal human skin. *Science* **348**, 880–886 (2015).
8.  Martincorena, I. et al. Somatic mutant clones colonize the human esophagus with age. *Science* **362**, 911–917 (2018).
9.  Lee-Six, H. et al. The landscape of somatic mutation in normal colorectal epithelial cells. *Nature* **574**, 532–537 (2019).
10. Lawson, A. R. J. et al. Extensive heterogeneity in somatic mutation and selection in the human bladder. *Science* **370**, 75–82 (2020).
11. Yoshida, K. et al. Tobacco smoking and somatic mutations in human bronchial epithelium. *Nature* **578**, 266–272 (2020).
12. Lee-Six, H. et al. Population dynamics of normal human blood inferred from somatic mutations. *Nature* **561**, 473–478 (2018).
13. Kennedy, S. R. et al. Detecting ultralow-frequency mutations by duplex sequencing. *Nat. Protoc.* **9**, 2586–2606 (2014).
14. Schmitt, M. W. et al. Detection of ultra-rare mutations by next-generation sequencing. *Proc. Natl Acad. Sci. USA* **109**, 14508–14513 (2012).
15. Li, R. et al. Macroscopic somatic clonal expansion in morphologically normal human urothelium. *Science* **370**, 82–89 (2020).
16. Cancer Genome Atlas Research Network. Comprehensive molecular characterization of urothelial bladder carcinoma. *Nature* **507**, 315–322 (2014).
17. Robertson, A. G. et al. Comprehensive molecular characterization of muscle-invasive bladder cancer. *Cell* **171**, 540–556 (2017).
18. Vinagre, J. et al. Frequency of TERT promoter mutations in human cancers. *Nat. Commun.* **4**, 2185 (2013).
19. Campbell, P. J. et al. Pan-cancer analysis of whole genomes. *Nature* **578**, 82–93 (2020).
20. Priestley, P. et al. Pan-cancer whole-genome analyses of metastatic solid tumours. *Nature* **575**, 210–216 (2019).
21. Martínez-Jiménez, F. et al. Pan-cancer whole-genome comparison of primary and metastatic solid tumours. *Nature* **618**, 333–341 (2023).
22. Rheinbay, E. et al. Analyses of non-coding somatic drivers in 2,658 cancer whole genomes. *Nature* **578**, 102–111 (2020).
23. Sabarinathan, R. et al. The whole-genome panorama of cancer drivers. Preprint at *bioRxiv* https://doi.org/10.1101/190330 (2017).
24. Martínez-Jiménez, F. et al. A compendium of mutational cancer driver genes. *Nat. Rev. Cancer* **20**, 555–572 (2020).
25. Bergstrom, E. N. et al. SigProfilerMatrixGenerator: a tool for visualizing and exploring patterns of small mutational events. *BMC Genomics* **20**, 685 (2019).
26. Islam, S. M. A. et al. Uncovering novel mutational signatures by de novo extraction with SigProfilerExtractor. *Cell Genomics* **2**, 100179 (2022).
27. Liu, M., Wu, Y., Jiang, N., Boot, A. & Rozen, S. G. mSigHdp: hierarchical Dirichlet process mixture modeling for mutational signature discovery. *NAR Genomics Bioinforma.* **5**, lqad005 (2023).
28. Nik-Zainal, S. et al. Mutational processes molding the genomes of 21 breast cancers. *Cell* **149**, 979–993 (2012).
29. Alexandrov, L. B. et al. Clock-like mutational processes in human somatic cells. *Nat. Genet.* **47**, 1402–1407 (2015).
30. Tate, J. G. et al. COSMIC: the catalogue of somatic mutations in cancer. *Nucleic Acids Res.* **47**, D941–D947 (2019).
31. Mularoni, L., Sabarinathan, R., Deu-Pons, J., Gonzalez-Perez, A. & López-Bigas, N. OncodriveFML: a general framework to identify coding and non-coding regions with cancer driver mutations. *Genome Biol.* **17**, 128 (2016).
32. Pellegrini, S. et al. Oncodrive3D: fast and accurate detection of structural clusters of somatic mutations under positive selection. *Nucleic Acids Res.* **53**, gkaf776 (2025).
33. Jeffares, D. C., Tomiczek, B., Sojo, V. & dos Reis, M. A beginners guide to estimating the non-synonymous to synonymous rate ratio of all protein-coding genes in a genome. *Methods Mol. Biol.* **1201**, 65–90 (2015).
34. Borah, S. et al. TERT promoter mutations and telomerase reactivation in urothelial cancer. *Science* **347**, 1006–1010 (2015).
35. Dyrskjøt, L. et al. Bladder cancer. *Nat. Rev. Dis. Primers* **9**, 58 (2023).
36. Jubber, I. et al. Epidemiology of bladder cancer in 2023: a systematic review of risk factors. *Eur. Urol.* **84**, 176–190 (2023).
37. Cumberbatch, M. G. K. et al. Epidemiology of bladder cancer: a systematic review and contemporary update of risk factors in 2018. *Eur. Urol.* **74**, 784–795 (2018).
38. Yokoyama, A. et al. Age-related remodelling of oesophageal epithelia by mutated cancer drivers. *Nature* **565**, 312–317 (2019).
39. Moore, L. et al. The mutational landscape of normal human endometrial epithelium. *Nature* **580**, 640–646 (2020).
40. Hurst, C. D. et al. Genomic subtypes of non-invasive bladder cancer with distinct metabolic profile and female gender bias in *KDM6A* mutation frequency. *Cancer Cell* **32**, 701–715 (2017).
41. Kato, S. et al. Understanding the function–structure and function–mutation relationships of p53 tumor suppressor protein by high-resolution missense mutation analysis. *Proc. Natl Acad. Sci. USA* **100**, 8424–8429 (2003).
42. Giacomelli, A. O. et al. Mutational processes shape the landscape of TP53 mutations in human cancer. *Nat. Genet.* **50**, 1381–1387 (2018).
43. Mighell, T. L., Evans-Dutson, S. & O'Roak, B. J. A saturation mutagenesis approach to understanding PTEN lipid phosphatase activity and genotype-phenotype relationships. *Am. J. Hum. Genet.* **102**, 943–955 (2018).
44. Bandaru, P. et al. Deconstruction of the Ras switching cycle through saturation mutagenesis. *eLife* **6**, e27810 (2017).
45. Lue, N. Z. et al. Base editor scanning charts the DNMT3A activity landscape. *Nat. Chem. Biol.* **19**, 176–186 (2023).
46. Muiños, F., Martínez-Jiménez, F., Pich, O., Gonzalez-Perez, A. & Lopez-Bigas, N. In silico saturation mutagenesis of cancer genes. *Nature* **596**, 428–432 (2021).
47. Demajo, S. et al. Identification of clonal hematopoiesis driver mutations through in silico saturation mutagenesis. *Cancer Discov.* **14**, 1717–1731 (2024).
48. Leroy, B., Anderson, M. & Soussi, T. TP53 mutations in human cancer: database reassessment and prospects for the next decade. *Hum. Mutat.* **35**, 672–688 (2014).
49. de Bruijn, I. et al. Analysis and visualization of longitudinal genomic and clinical data from the AACR project GENIE Biopharma Collaborative in cBioPortal. *Cancer Res.* **83**, 3861–3867 (2023).
50. AACR Project GENIE Consortium AACR project GENIE: powering precision medicine through an International Consortium. *Cancer Discov.* **7**, 818–831 (2017).
51. Funk, J. S. et al. Deep CRISPR mutagenesis characterizes the functional diversity of *TP53* mutations. *Nat. Genet.* **57**, 140–153 (2025).
52. Salk, J. J. et al. Ultra-sensitive *TP53* sequencing for cancer detection reveals progressive clonal selection in normal tissue over a century of human lifespan. *Cell Rep* **28**, 132–144 (2019).
53. Matas, J. et al. Colorectal cancer is associated with the presence of cancer driver mutations in normal colon. *Cancer Res.* **82**, 1492–1502 (2022).
54. Rios-Doria, E. et al. TP53 somatic evolution in the normal endometrium of Black and White individuals. *Gynecol. Oncol.* **197**, 1–10 (2025).
55. Kircher, M. et al. Saturation mutagenesis of twenty disease-associated regulatory elements at single base-pair resolution. *Nat. Commun.* **10**, 3583 (2019).
56. Hill, W., Weeden, C. E. & Swanton, C. Tumor promoters and opportunities for molecular cancer prevention. *Cancer Discov.* **14**, 1154–1160 (2024).
57. Hill, W. et al. Lung adenocarcinoma promotion by air pollutants. *Nature* **616**, 159–167 (2023).
58. Balmain, A. Peto's paradox revisited: black box vs mechanistic approaches to understanding the roles of mutations and promoting factors in cancer. *Eur. J. Epidemiol.* **38**, 1251–1258 (2023).
59. Axelsson, T. A. et al. Diagnostic and prognostic genomic aberrations in upper tract urothelial carcinoma can be identified in focal barbotage samples. *BJU Int.* **135**, 792–801 (2025).
60. Wahida, A. et al. The coming decade in precision oncology: six riddles. *Nat. Rev. Cancer* **23**, 43–54 (2023).

# Methods

## Sample collection

Two epithelial brushes (2–3 cm$^2$) from the bladder top (dome) and the bladder floor (trigone) were obtained from 53 people without known bladder pathology and no history of bladder cancer upon autopsy (average 4 days post-mortem) at the University of Washington. Next of kin consented to autopsies and research on leftover specimens. The study of de-identified collected specimens and linked clinical history from the deceased donors was reviewed and deemed not human subjects research by the University of Washington Institutional Review Board (STUDY00016707; IRB Federal Wide Assurance number, FWA 00006878). We obtained the following relevant clinical information for all donors: age, sex, BMI, tobacco smoking history, alcohol use, previous cancer and chemotherapy exposure (Supplementary Table 1). Three donors with active or chronic inflammation and four donors with insufficient DNA were discarded. The two samples from another donor were also discarded from the study upon visual inspection of their mutational profile, resulting in a total of 45 deceased donors in this study (Supplementary Notes 2 and 3).

## Duplex DNA sequencing

**Capture panel design.** We designed a panel including ten genes identified in a previous study as being under positive selection in the normal urothelium and with more than ten mutations across the 1,647 microbiopsies analysed[10]. We added five genes that are mutated frequently in bladder tumours[30] and the *TERT* promoter—also known to be under positive selection in bladder carcinomas[22,23]. This resulted in a panel containing the entire (or almost entire) coding region of 12 genes (*ARID1A*, *NOTCH2*, *FOXQ1*, *CDKN1A*, *KMT2D*, *RB1*, *CREBBP*, *TP53*, *EP300*, *KDM6A*, *RBM10* and *STAG2*), three genes for which only selected regions were targeted because of clustering of cancer mutations in those regions and/or difficulties for capturing the full gene (*PIK3CA*, *FGFR3* and *KMT2C*) and the *TERT* promoter[10,61]. The panel was constructed by TwinStrand Biosciences and covered 111,876 base pairs, including 65,086 base pairs in coding regions and 46,790 base pairs in non-coding regions (Supplementary Tables 2 and 3 and Supplementary Note 2).

**DNA extraction, duplex library preparation and sequencing.** After centrifugation of epithelial brushes, DNA was extracted using the DNeasy Blood and Tissue (Qiagen) kit, following manufacturer's instructions with some variations (Supplementary Note 2). The DNA integrity number (DIN) was measured using Agilent 4200 TapeStation Genomic tapes. Duplex sequencing libraries were prepared using commercially available kits (TwinStrand Biosciences)[13,14,62–64] and 250 ng of genomic DNA. The DNA fragmentation step was carried out taking into account the starting DIN of samples and monitored using TapeStation. Fragmented DNA was subject to end-repair, A-tailing, ligation to duplex sequencing adaptors, library conditioning and PCR amplification, according to protocol. The PCR product was captured at 65 °C for 16–20 h. Upon PCR amplification, libraries were pooled for sequencing. Sequencing was performed with a NovaSeq 6000 at the Department of Laboratory Medicine and Pathology at University of Washington or a NovaSeq X Plus at Novogene or the Fred Hutchinson Cancer Center using 2 × 150 base-pair paired-end reads (around 115 million reads per sample; Supplementary Note 2).

Duplex DNA sequencing was successful for 34 donors on both samples. For 11 other donors, we produced duplex DNA sequencing data only for dome or trigone because of insufficient DNA or too fragmented DNA (DIN < 1.4) in the paired sample. Thus, the final number of donors with available data for at least one sample was 45 (Supplementary Table 1) and the final number of samples processed was 79 (Supplementary Table 4).

## Somatic mutation calling

We constructed a computational pipeline (deepUMIcaller) in Nextflow[65] to call mutations from duplex sequencing data on the basis of an early version of nf-core/fastquorum pipeline[66], which implements the fgbio Best Practices FASTQ to Consensus Pipeline (https://github.com/fulcrumgenomics/fgbio/blob/main/docs/best-practice-consensus-pipeline.md) and downstream variant calling by VarDictJava (https://github.com/AstraZeneca-NGS/VarDictJava). A series of filters to discard potential artefacts are included in the pipeline. Code implementing deepUMIcaller is available at (github.com/bbglab/deepumicaller). For a detailed description of the pipeline, see Supplementary Note 3.

To estimate the error rate of TwinStrand DNA duplex sequencing, we compared the density of mutations and mutational profile identified across three cord blood samples (purchased from StemCell) with that expected on the basis of colonies obtained from human haematopoietic stem cells[67,68]. The estimated error rate resulting from this analysis was around $4 \times 10^{-8}$, which is two orders of magnitude lower than the mutation density across samples in this study (Extended Data Fig. 2a and Supplementary Notes 4 and 8).

## Mutational signatures

**De novo signature extraction.** We extracted mutational signatures de novo with a Bayesian hierarchical Dirichlet process using HDP_sigExtraction pipeline (https://github.com/McGranahanLab/HDP_sigExtraction) and the R-package hdp developed by N. Roberts (https://github.com/nicolaroberts/hdp)[9] and SigProfilerExtractor (https://github.com/AlexandrovLab/SigProfilerExtractor)[26]. The HDP_sigExtraction pipeline was run with the default parameters with no previous signatures assigned. Five signatures in addition to the null signature were extracted. De novo signatures were extracted using SigProfiler using the nonnegative matrix factorization approach. The same input data were used. The upper bound for the number of signatures was set to ten; however, the most robust solution was three signatures that were similar to the three most active signatures extracted by HDP. These two sets of mutational signatures were decomposed into known COSMIC signatures when possible (Supplementary Note 5).

**Assessing biases in trinucleotide composition of the panel.** To account for biases due to the trinucleotide composition of the panel, the number of substitutions of each class was re-calculated. The rate of each substitution was calculated as the number of observed substitutions with the consequence in question divided by the number of corresponding sequenced trinucleotide sites (number of trinucleotides with this consequence in the panel weighted by sequencing depth). This mutational probability was multiplied by the number of the corresponding trinucleotide in the genome to get the expected number of substitutions of this type in the whole genome. The mutational probability of each substitution for the whole genome was calculated by dividing the expected number of substitutions with this consequence per genome by the sum of all expected substitutions per genome. Finally, the expected number of substitutions with each consequence was obtained by multiplying the probability by the total number of substitutions observed in the sample to keep the absolute number of observed mutations.

**Statistical association between mutational signatures and clinical variables.** We tested for possible associations of de novo extracted signatures with age, sex and bladder location. To do this we used linear mixed-effect regression models (Supplementary Note 5). We tested both the number of mutations attributed to the signature (counts) and the relative contribution (proportion) of each signature as a response variable. Age, sex and bladder location were used as fixed effects (separated regression was prepared for each fixed effect variable) and donors were used as random effects to control for non-independence between dome and trigone samples from the same person. *P* values

for the association were obtained using likelihood-ratio tests (ANOVA function) comparing models including and excluding the variable of interest. Associations with all other clinical variables (smoking history, drinking history, chemotherapy history, and so on) were tested in the same way but always including age, sex and bladder location in the regression together with the variable of interest.

## Positive selection

We used four methods to compute positive selection on the mutations observed across genes. One method (Omega)—a dN/dS approach to assess the strength of selection on the mutational pattern of genes—was developed de novo for this study and is described at length in Supplementary Note 6 (https://github.com/bbglab/omega). Two others, OncodriveFML and Oncodrive3D, which compute the deviation in the average functional impact and clustering in the three-dimensional structure of proteins, respectively, from those expected under neutrality, had been developed previously, and were adapted here to work on duplex sequencing data. These are also described in Supplementary Note 6. A fourth method, assessing the relative enrichment for frameshift indels observed across genes was also developed de novo for this study and is described thoroughly in Supplementary Note 6. For the *TERT* promoter, all mutations that were observed at least twice across 8,136 whole-genome sequencing tumour samples sequenced by the Hartwig Medical Foundation and the Pan-Cancer Analysis of Whole Genomes consortium (Supplementary Note 6) were considered activating. The remaining mutations were used to calculate the expected number of activating mutations under neutrality.

## Fraction of the urothelium covered by clones with driver mutations

The number of missense and truncating driver mutations of each gene in a sample was obtained from the dN/dS missense and dN/dS truncating values. From these values the number of missense and truncating mutations in excess in each sample were calculated. Specifically, the 95% confidence intervals of both dN/dS values were used to compute two extreme values of driver mutations in each sample, whereas the mean dN/dS value was used to compute an expected number of driver mutations.

This mean number was thus used to select the most likely driver mutations in the sample. To this end, mutations were ranked in descending order according to their VAF, and the top number of mutations corresponding to the expected number of driver mutations were selected. Then, for a gene $G$, we computed the fraction of genomes bearing driver mutations out of those sequenced at each genomic position covered by the DNA duplex sequencing as:

$$P(G) = \sum_{i=1}^{n} (-1)^{i-1} \sum_{|S|=i} \prod_{x \in S} p_x = \sum_x p_x - \sum_{x \neq y} p_x p_y + \cdots + (-1)^{n-1} p_{x_1} p_{x_2} \cdots p_{x_n}$$

where $p_x$ is the all-molecules VAF of a driver mutation $x$ (considering both duplex and non-duplex reads; Supplementary Note 3) and $n$ is the number of driver mutations from the previously selected set in the gene. The formula is the realization of the probabilistic principle of inclusion-exclusion assuming independence across mutations[69].

The fraction of genomes with driver mutations is equal to the fraction of cells with driver mutations under the assumption that two mutations, one in each homologous copy of the gene, are required to drive the clonal expansion. This extreme would be true in the case that all studied genes behave as classical tumour suppressors and bear deleterious mutations in both alleles. An exception to this rule are the mutations in genes in the X chromosome in men, whereby one mutation would suffice to drive the clonal expansion. In general, it is possible for some genes to be capable of driving the clonal expansion (or mutations at specific positions in a gene) with only one mutated allele. It is also possible that the other allele is affected by a large deletion or methylation

event not seen by DNA duplex sequencing. In the extreme that only one mutation per cell per gene is required, the fraction of cells with driver mutations will be double the fraction of genomes. In the plot in Extended Data Fig. 5b, we use the two-hit assumption.

## Association of the urothelium clonal structure with bladder cancer risk factors

We designed a two-step strategy to assess the influence of known bladder cancer risk factors on features of the urothelium clonal structure across samples from donors in the cohort. As dependent variables representing the urothelium clonal structure, we selected the (protein-affecting and non-protein-affecting) mutation density, and the magnitude of positive selection on genic mutations (dN/dS missense and dN/dS truncating). As the dependent variables are continuous, we chose linear regressions, specifically mixed-effects linear models, to account for two samples from the same donor and use all available samples. As independent variables, we included a list of available clinical features: age (in decades), sex, smoking history (binarized as ever or never smoker), alcohol consumption history (also binarized), BMI (re-scaled within the 0–1 interval), and exposure to chemo/radiotherapy (binarized). For the case of *TERT* promoter mutation density association, we used the interaction of age and smoking history instead of age and smoking history separately. In the first step, we applied univariate mixed-effects linear models to identify any clinical feature with a significant association with any of the dependent variables for any gene or for all genes. In the second step, we applied multivariate mixed-effects linear models to rule out confounding effects for the associations that seemed significant in the first step. Associations with FDR below 0.2 were deemed significant. We also carried out a binomial test to rule out a spurious dependence of the mutation density on group differences in terms of sequencing depth. For details, see Supplementary Notes 9 and 10.

## Tumour data

Mutations identified across 622 muscle-invasive and 105 non-muscle-invasive bladder cancer (MIBC and NMBIC, respectively) cohorts were downloaded from cBioPortal[49,70–73] together with the clinical data of the combined study. From the MIBC Beijing Genomics Institute cohort[74] only samples labelled as invasive were considered MIBC and added to the MIBC dataset. Mutations identified in a further cohort of 79 NMIBCs were obtained from the literature[40] and included as part of the NMIBC dataset. Only non-silent protein-affecting mutations in 14 of the genes included in the panel were analysed. In case of duplicated samples across MIBC and NMBIC cohorts, the mutations from the most recent published study were kept. In total, mutations across 806 bladder tumours were obtained (622 MIBC and 184 NMIBC). The mutation density per gene across the two datasets was calculated by dividing the number of observed mutations per gene by the length of its coding region. To compute a mutation density metric comparable with that used in the normal urothelium that accounts for the number of megabases sequenced, we multiplied the coding length by two times the number of samples in which that gene was sequenced, to take into account the two copies in a diploid genome. Under the assumption that, in tumours each mutation belongs to a single clonal expansion and in normal tissues we have a mixture of clones, we reasoned that the number of different tumour genomes sequenced would be equivalent to the number of genomes sequenced in normal urothelium. The mutation density per megabase was calculated multiplying the mutation density by $10^6$.

Mutations identified across 33,218 tumours (892 bladder tumours, mostly MIBC) in intOGen[24] were downloaded from intogen.org. These mutations were used to obtain the total number of mutations observed in each gene, their distribution along the sequence of the genes in the study and the percentage of sites affected by different numbers of mutations. The same data of 109,017 tumours (3,909 bladder tumours) were obtained from the GENIE project[50] to calculate the frequency of

mutations in each of the genes and the *TERT* promoter. Mutations in the *TERT* promoter were also obtained from two cohorts of tumours (included in intOGen) sequenced at the whole-genome level (Hartwig Medical Foundation[21]: *N* = 5,582 and Pan-Cancer Analysis of Whole Genomes[19]: *N* = 2,554). The classification (and score) of all possible mutations in *TP53* into drivers and passengers through in silico saturation mutagenesis was obtained from boostDM (intogen.org/boostDM). Details of analyses involving tumour mutations appear in Supplementary Note 11.

### Natural saturation mutagenesis
To compare the distribution of somatic mutations along the sequence of each of the genes in normal urothelium and bladder tumours, we first downloaded somatic mutations identified across 892 bladder tumours from the intOGen (intogen.org) platform. Although this cohort is larger than the 806 muscle-invasive and non-muscle-invasive tumours used to explore differences in mutation density, it is composed mostly of muscle-invasive carcinomas.

For each gene, we started the analysis with all genomic sites in the coding sequences and splicing sites that passed the pipeline's filtering criteria of sufficient duplex coverage across samples (Supplementary Note 3). We then mapped these DNA sites to protein positions using the Ensembl REST API[75], allowing us to determine the total number of protein positions (including amino acid residues and stop codon) covered by the panel. Then, we computed the number of mutations affecting each protein residue of every gene studied here across the 79 normal urothelium samples and the 892 bladder tumours. Next, we obtained the consequence type (missense, synonymous, nonsense or splice-affecting) of these mutations on the MANE transcript[76] of each gene from the output of the Variant Effect Predictor v.111 (ref. 77), run within the intOGen pipeline[24] (https://github.com/bbglab/intogen-plus). The method to compute the natural saturation mutagenesis kinetics in Fig. 5b is described in Supplementary Note 12. The residue-level sequencing depth shown in Fig. 5c, Extended Data Fig. 9d and Supplementary Figs. 2 and 3 was calculated as the average depth across all sites corresponding to each codon.

### Calculation of site selection
We developed a metric to measure selection per site by comparing the observed with the expected number of mutations at each site or residue. The expected number of mutations per site was obtained by distributing the expected number of mutations in a given gene under neutrality along all the possible changes in that same genomic region, with the assumption that only the mutation probability of each trinucleotide and the sequencing depth are responsible for the within-gene differences of mutation probability. We computed this value for each possible mutation in the positively selected genes including the *TERT* promoter, and for the protein-coding genes we also obtained a value per residue. A more complete explanation on this metric (and the detection of positive selection in sub-genic structures such as exons and domains) can be found in Supplementary Note 6.

### Structural representation and features
Structural models for all proteins used to run Oncodrive3D were obtained from the AlphaFold database (AlphaFold 2 v.4)[78,79]. Three-dimensional protein structures and protein structural features represented across figures were also obtained from the AlphaFold database. Solvent accessibility and secondary structure information were extracted from the AlphaFold-predicted PDB structures using PDB_Tool (https://github.com/realbigws/PDB_Tool). Structural visualizations of proteins were produced using UCSF ChimeraX[80].

### Comparison with experimental saturation mutagenesis
The results of two saturation mutagenesis experiments estimating the functional impact of mutations in the *TP53* DNA-binding domain[51]

and along the sequence of the *TERT* promoter[55] were obtained directly from supplementary tables from both papers. In the case of *TP53*, we used the transformed score. In the case of the *TERT* promoter, values calculated for the glioblastoma SF7996 cells system were used. In these comparisons, the site selection of individual mutations was used.

### Comparison of orthogonal studies of normal bladder
We obtained the mutations identified through whole-genome sequencing in clonal or quasi-clonal samples obtained from the normal bladder of donors by laser capture microdissection in a previous study[10]. We used these samples to construct the mutational profile of normal urothelium, which we compared with that reconstructed in the present study through ultradeep sequencing (Extended Data Fig. 2b). From the same previous study, we obtained the mutations identified through whole-exome sequencing. Mutations obtained from samples sequenced at depth lower than 80× were discarded. From these data, we calculated the number of mutations per megabase observed in each sample, and averaged this value across donors. A linear regression was then constructed between these values and the age of donors, and a trend line calculated. The number of mutations per megabase calculated across the samples in our cohort was overlaid upon the obtained regression. The results of this comparison are presented in Extended Data Fig. 2c.

### Reporting summary
Further information on research design is available in the Nature Portfolio Reporting Summary linked to this article.

## Data availability
Raw sequencing data for this study were deposited in dbGaP under accession number phs004105.v1.p1. The set of mutations used in all analyses presented in the paper is available at Zenodo (https://doi.org/10.5281/zenodo.15836679)[81]. Executing the code provided in the third repository mentioned below, all figures in the paper can be reproduced. Reference mutational signatures were obtained from https://cancer.sanger.ac.uk/signatures/sbs/. Tumour mutations were obtained through cBioPortal (datasets from refs. 16,17,74 at https://www.cbioportal.org/), intogen (intogen.org) and the GENIE synapse data portal (https://genie.synapse.org/) as described in Methods. Protein structural models for the entire human proteome were obtained from the AlphaFold database (https://alphafold.ebi.ac.uk/). The results of two experimental saturation mutagenesis studies on *TP53* and the *TERT* promoter were obtained from refs. 51 and 55, respectively.

## Code availability
The code of the mutation calling (deepUMIcaller) pipeline is available at (https://github.com/bbglab/deepUMIcaller). The code of a pipeline containing all analyses described in this manuscript (deepCSA) is available at (https://github.com/bbglab/deepCSA). Code needed to reproduce paper figures using the data in the Zenodo repository mentioned above is available at (https://github.com/bbglab/normal_bladder_paper).

61. Ju, Y. S. The mutational signatures and molecular alterations of bladder cancer. *Transl. Cancer Res.* **6**, S689–S701 (2017).
62. Dillon, L. W. et al. Quantification of measurable residual disease using duplex sequencing in adults with acute myeloid leukemia. *Haematologica* **109**, 401–410 (2024).
63. Woolston, D. W. et al. Ultra-deep mutational landscape in chronic lymphocytic leukemia uncovers dynamics of resistance to targeted therapies. *Haematologica* **109**, 835–845 (2024).
64. Oshima, M. U. et al. Characterization of clonal dynamics using duplex sequencing in donor-recipient pairs decades after hematopoietic cell transplantation. *Sci. Transl. Med.* **16**, eado5108 (2024).
65. Di Tommaso, P. et al. Nextflow enables reproducible computational workflows. *Nat. Biotechnol.* **35**, 316–319 (2017).
66. Ewels, P. A. et al. The nf-core framework for community-curated bioinformatics pipelines. *Nat. Biotechnol.* **38**, 276–278 (2020).

67. Osorio, F. G. et al. Somatic mutations reveal lineage relationships and age-related mutagenesis in human hematopoiesis. *Cell Rep.* **25**, 2308–2316 (2018).
68. Machado, H. E. et al. Diverse mutational landscapes in human lymphocytes. *Nature* **608**, 724–732 (2022).
69. Grinstead, C. M & Snell, J. L. *Introduction to Probability: Second Revised Edition* (American Mathematical Society, 1997).
70. Cerami, E. et al. The cBio cancer genomics portal: an open platform for exploring multidimensional cancer genomics data. *Cancer Discov.* **2**, 401–404 (2012).
71. Gao, J. et al. Integrative analysis of complex cancer genomics and clinical profiles using the cBioPortal. *Sci. Signal.* **6**, pl1 (2013).
72. Van Allen, E. M. et al. Somatic ERCC2 mutations correlate with cisplatin sensitivity in muscle-invasive urothelial carcinoma. *Cancer Discov.* **4**, 1140–1153 (2014).
73. Pietzak, E. J. et al. Genomic differences between 'primary' and 'secondary' muscle-invasive bladder cancer as a basis for disparate outcomes to cisplatin-based neoadjuvant chemotherapy. *Eur. Urol.* **75**, 231–239 (2019).
74. Guo, G. et al. Whole-genome and whole-exome sequencing of bladder cancer identifies frequent alterations in genes involved in sister chromatid cohesion and segregation. *Nat. Genet.* **45**, 1459–1463 (2013).
75. Harrison, P. W. et al. Ensembl 2024. *Nucleic Acids Res.* **52**, D891–D899 (2024).
76. Morales, J. et al. A joint NCBI and EMBL–EBI transcript set for clinical genomics and research. *Nature* **604**, 310–315 (2022).
77. McLaren, W. et al. The Ensembl variant effect predictor. *Genome Biol.* **17**, 122 (2016).
78. Jumper, J. et al. Highly accurate protein structure prediction with AlphaFold. *Nature* **596**, 583–589 (2021).
79. Varadi, M. et al. AlphaFold Protein Structure Database: massively expanding the structural coverage of protein-sequence space with high-accuracy models. *Nucleic Acids Res.* **50**, D439–D444 (2021).
80. Pettersen, E. F. et al. UCSF ChimeraX: structure visualization for researchers, educators, and developers. *Protein Sci.* **30**, 70–82 (2021).
81. Calvet, F. et al. Sex and smoking bias in the selection of somatic mutations in human bladder - Figures data. *Zenodo* https://doi.org/10.5281/zenodo.15836679 (2025).
82. Abascal, F. et al. Somatic mutation landscapes at single-molecule resolution. *Nature* **593**, 405–410 (2021).
83. Landrum, M. J. et al. ClinVar: public archive of relationships among sequence variation and human phenotype. *Nucleic Acids Res.* **42**, D980–D985 (2014).
84. Chakravarty, D. et al. OncoKB: a precision oncology knowledge base. *JCO Precis. Oncol.* https://doi.org/10.1200/PO.17.00011 (2017).

**Acknowledgements** We acknowledge the donors and their families for generously contributing their samples. We thank the University of Washington autopsy service staff including B. McGing and J. Grillo, who were instrumental in sample collection, and N. Jackson and K. Scherpelz for facilitating and supporting autopsy collection. We also thank C. Coombes for revising the manuscript, E. Jonlin for regulatory support and F. Brando for technical support in the pipeline for data analysis. We would also like to thank the nf-core community for developing the nf-core infrastructure and resources for Nextflow pipelines. Institute for Research in Biomedicine Barcelona is a recipient of a Severo Ochoa Centre of Excellence Award from the Spanish Ministry of Economy and Competitiveness (MINECO; Government of Spain) and an Excellence Institutional grant by the Asociacion Española contra el Cancer, and is supported by CERCA (Generalitat de Catalunya). R.B.M.-I. acknowledges support from a fellowship from the Ministerio de Universidades para la Formación de Profesorado Universitario (FPU21/05649). F.C. acknowledges support of fellowship PRE2022-101268, funded by MICIU/AEI /10.13039/501100011033 and by FSE+. A.R.H. is supported by an EMBO long-term fellowship (ALTF 1058-2022). J.E.R.-Z. was supported by a Postdoctoral AECC 2023 fellowship from Fundación Científica Asociación Española Contra el Cáncer (POSTD234814RAMI). This work was delivered as part of the PROMINENT team supported by the Cancer Grand Challenges partnership funded by Cancer Research UK (CGCATF-2021/100008), the NCI (OT2CA278668) and the Scientific Foundation of the Spanish Association Against Cancer, AECC. This research has been conducted using the UK Biobank Resource under application number 69794. This research was funded with two Research Grants (2020, 2023) from the Department of Laboratory Medicine and Pathology, University of Washington and support from NCI R01 CA259384 (R.A.R.). Components of Fig. 1a and Supplementary Fig. 10 were created using BioRender (https://BioRender.com/fgnnet9).

**Author contributions** F.C. designed, coded and validated the DNA duplex calling and analysis pipelines, designed and carried out data analysis, participated in data interpretation (including duplex mutation calling, computation of mutation density, study of variability between samples and individuals, and association of these metrics with bladder cancer risk factors and saturation mutagenesis), and participated in drafting and revising the manuscript. R.B.M.-I. participated in coding specific parts of the DNA duplex analysis pipeline, designed and carried out data analysis (including duplex mutation calling, computation of mutation density, estimation of positive selection, study of variability between samples and individuals, and association of these metrics with bladder cancer risk factors) and participated in data interpretation, drafting and revising the manuscript. F.M. participated in coding specific parts of the DNA duplex analysis pipeline, designed and carried out specific data analyses (including estimation of positive selection, association of these metrics with bladder cancer risk factors and saturation mutagenesis) and participated in data interpretation and in revising the manuscript and drafting supplementary information. M.T. designed, performed and supervised sample collection, retrieved clinical information, reviewed pathological findings, participated in project design and obtained funding. E.S.L.-E. performed optimization of enzymatic fragmentation, sample processing, and patient and sample data curation. J.F. performed optimization of enzymatic fragmentation, sample processing, and patient and samples data curation, contributed to the design and optimization of the capture panel, and supervised sequencing and data quality control. M.A. performed the mutational signature analysis and participated in data interpretation and in revising the manuscript and drafting supplementary information. S.P. participated in the analysis of positive selection signals, prepared plots (including saturation mutagenesis plots), and participated in data interpretation and in revising the manuscript and drafting supplementary information. A.R.H. carried out the analysis of the expected fraction of mutated urothelium and participated in data interpretation and in revising the manuscript and drafting supplementary information. J.E.R.-Z. did the epidemiological analyses with UK BioBank data and participated in data interpretation and in revising the manuscript and drafting supplementary information. S.C.A. contributed to sample processing. E.T. contributed to sample processing and data analysis. B.F.K. performed data processing and quality control data analysis, contributed to the design and optimization of the capture panel and supervised sequencing and data quality control. M.L.G. contributed to the development of the DNA duplex calling pipeline. A.G.-P. conceptualized and supervised the project, participated in the interpretation of results and led the drafting of the manuscript. N.L.-B. conceptualized and supervised the project, obtained funding, participated in the interpretation of results and led the drafting of the manuscript. R.A.R. designed the project, obtained funding, supervised sample collection, contributed to the design and optimization of the capture panel, supervised sequencing and data quality control, conceptualized and supervised the project, participated in the interpretation of results and led the drafting of the manuscript. All authors have read and approved the manuscript. F.C. and R.B.M-I. contributed equally to the manuscript and author order was decided randomly.

**Competing interests** R.A.R. is an equity holder at TwinStrand Biosciences Inc. and NanoString Technologies Inc. R.A.R. is named inventor on patent no. 11,479,807 (Methods for targeted nucleic acid sequence enrichment with applications to error corrected nucleic acid sequencing) owned by the University of Washington and licensed to TwinStrand Biosciences Inc. R.A.R. was a consultant at TwinStrand Biosciences Inc. and received research funding from a joint research grant with TwinStrand Biosciences Inc. and Ovartec GmbH. B.F.K. is an equity holder at NanoString Technologies Inc. The other authors declare no competing interests.

**Additional information**
**Correspondence and requests for materials** should be addressed to Nuria Lopez-Bigas or Rosa Ana Risques.

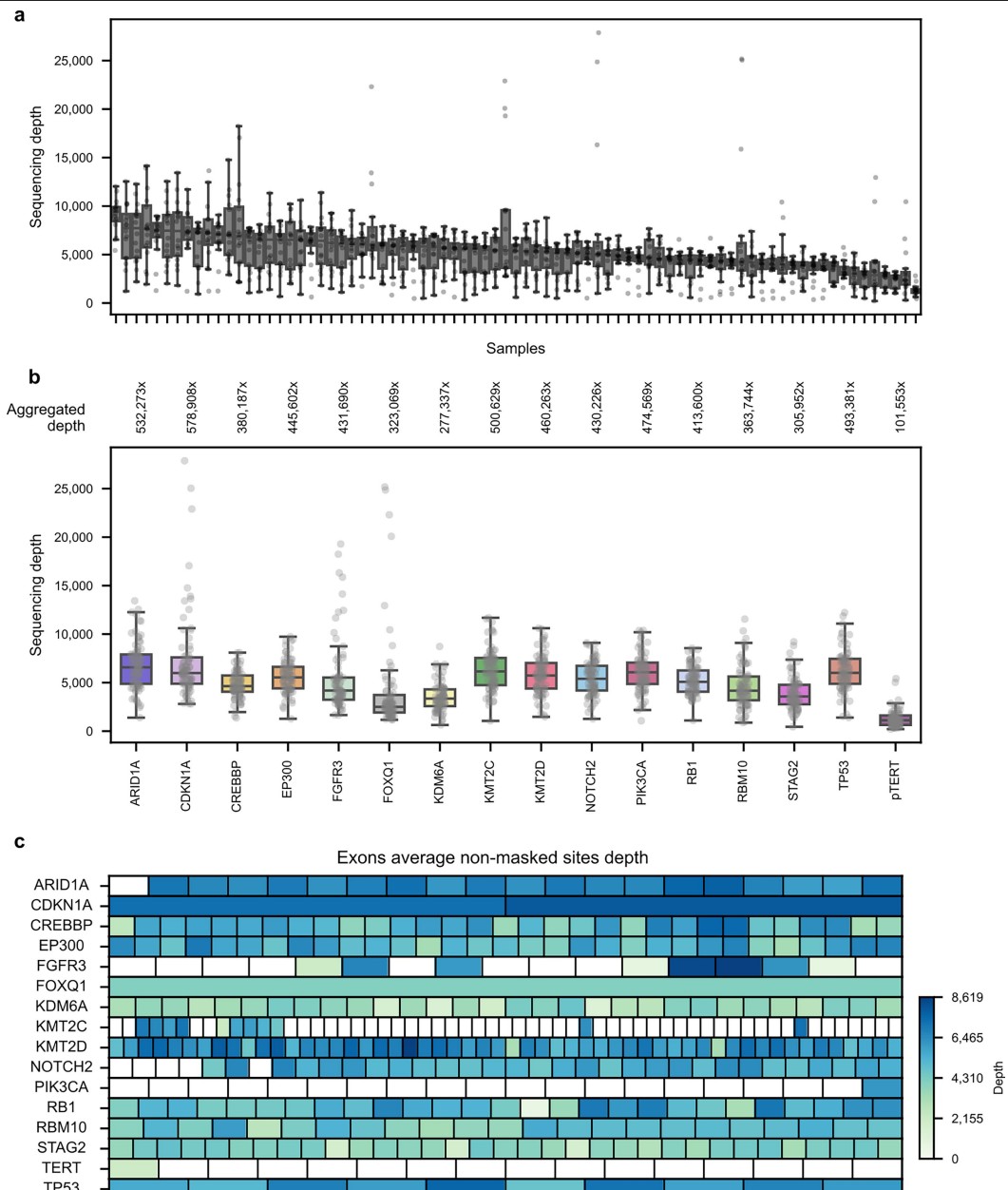

**Extended Data Fig. 1 | Ultradeep sequencing of a mixture of urothelial clones.**
a) Depth of DNA duplex sequencing across the 79 samples. The boxplots represent the distribution of sequencing depth across all genes included in the panel, while the dots represent the average sequencing depth obtained for each gene in each sample. b) Distribution of DNA duplex sequencing depth across the 16 genes. The boxplots represent the distribution of sequencing depth across all samples, and the dots represent the average sequencing depth per gene in each sample. c) Average DNA duplex sequencing depth obtained for each exon included in the panel, represented by the color of each tile according to the color scale in the colorbar on the right. White color represents the exon is not covered. Box plots in a and b display the quartiles with whiskers extending to the highest and lowest data points within 1.5 times the interquartile range. Details can be found in Supplementary Note 3.

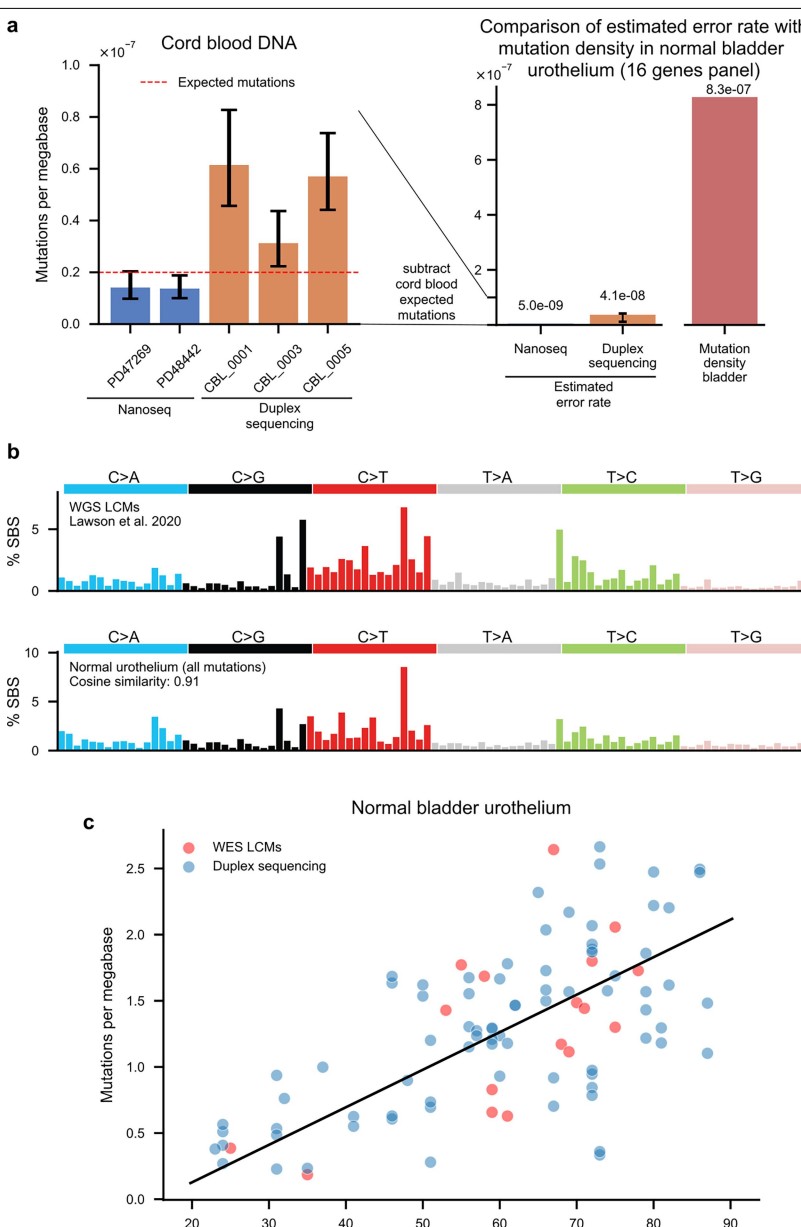

**Extended Data Fig. 2 | Error rate of the DNA duplex sequencing technology.**
a) Left panel, orange bars, mutations per sequenced nucleotide detected by the DNA duplex sequencing technology in this study using the same panel of bladder genes in cord blood DNA samples from three donors. Blue bars (for comparison), mutations per sequenced nucleotide detected by a similar technology (NanoSeq; data taken from ref. 82) in two cord blood samples (Supplementary Note 4). Each bar presents the mutation density computed for a separate cord blood sample, with vertical lines representing the Poisson 95% confidence intervals. The comparison of the number of observed mutations per sequenced nucleotide with those expected in cord blood DNA based on prior studies (dashed red line, Supplementary Note 4) yields an estimate of the error rate of the technology of ~4 × 10⁻⁸ per sequenced nucleotide. Right panel, comparison of the estimated error rate by both technologies with the mutations per sequence nucleotide across the 79 normal urothelial samples included in this study (rightmost red bar). It shows that the rate of errors of the DNA duplex sequencing technology used in the study is approximately 25 times smaller

than the mutation density detected in the normal urothelium. (N = 3 for cord blood DNA duplex sequencing). b) Mutational profile of normal urothelium obtained through two orthogonal approaches. Top panel, profile constructed using mutations detected through laser capture microdissection (LCM) of clonal or quasi-clonal samples followed by regular shallow whole-genome sequencing (data taken from ref. 10). Bottom panel, profile constructed using mutations detected in this study from ~2 cm² brushes followed by ultradeep DNA duplex sequencing. c) Relationship between the mutation density calculated in normal bladder urothelium using two orthogonal approaches. The red dots correspond to the rate of mutations detected through laser capture microdissection of clonal or quasi-clonal samples followed by regular shallow whole-exome sequencing (WES LCMs; data taken from ref. 10). The blue dots correspond to the rate of mutations computed for the 79 samples in this study from ~2 cm² brushes followed by ultradeep DNA duplex sequencing. The trend line represented in the plot was calculated from the WES LCMs samples. For more details on the error rate of the technology, see Supplementary Note 4.

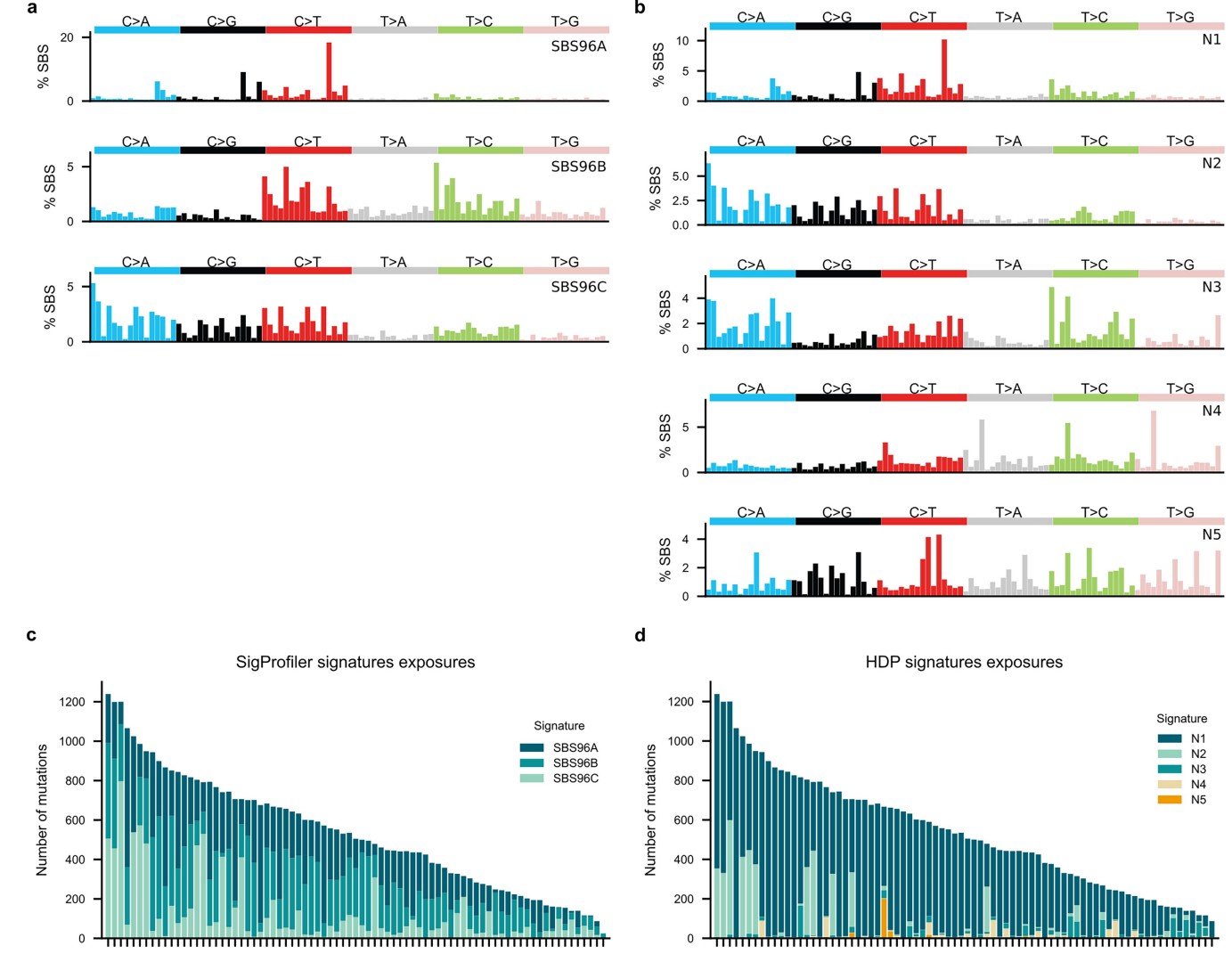

**Extended Data Fig. 3 | Mutational signatures active across the cohort.**
a) Mutational profile of the signatures identified using SigProfiler. b) Mutational profile of the signatures identified using HDP. c) Activity of the signatures identified using SigProfiler across the 79 samples. d) Activity of the signatures identified using HDP across the 79 samples. For more details on the identification of these mutational signatures and the decipherment of their etiology, see Supplementary Note 5.

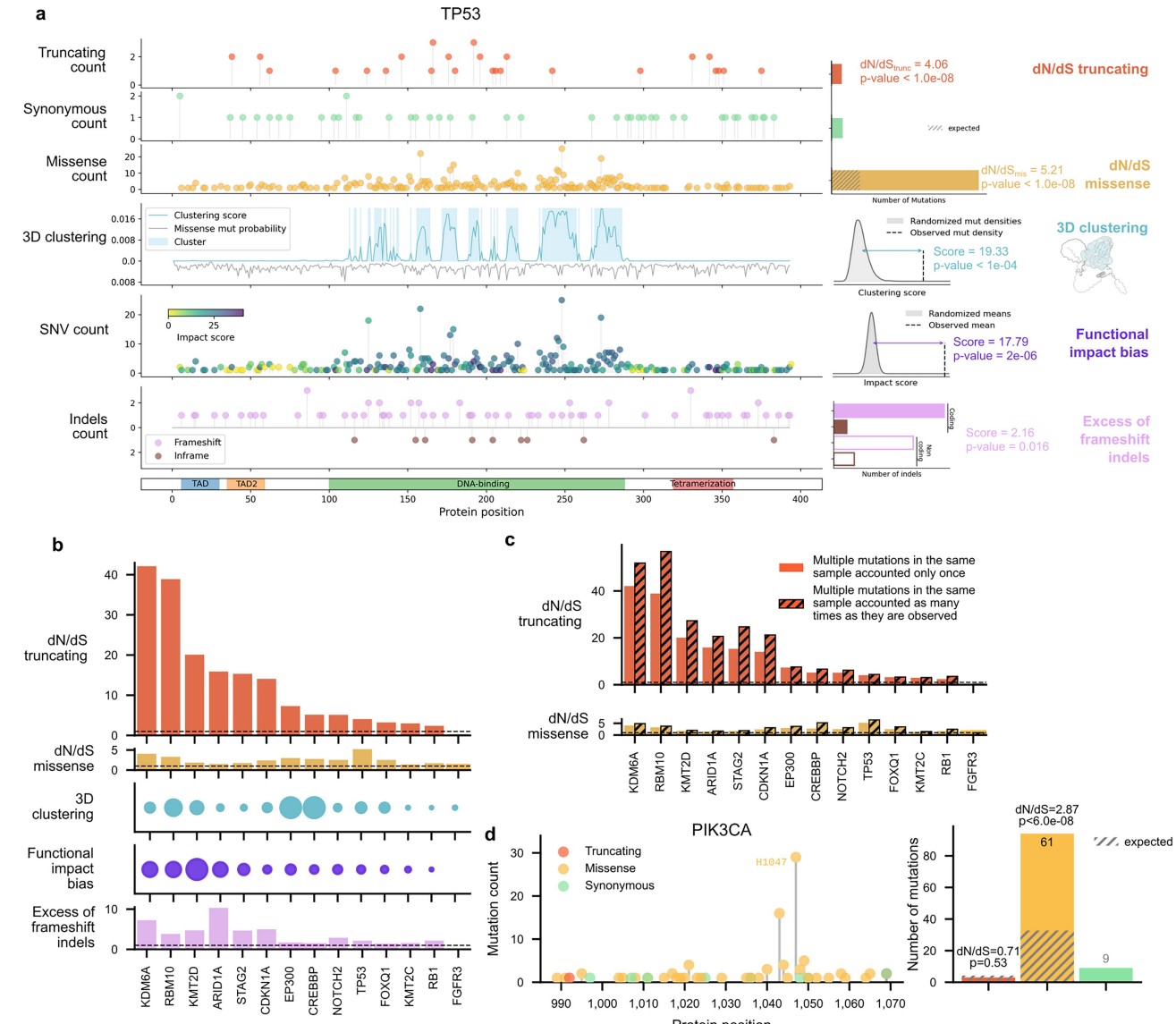

**Extended Data Fig. 4 | Calculation of positive selection.** a) Five signals of positive selection on the mutations observed in *TP53* in all samples. The first three rows show the distribution of truncating (nonsense and splice-site affecting), synonymous, and missense mutation across the coding region of *TP53*. On the right side, dN/dS truncating (top row) and dN/dS missense (third row) represent the estimation of the excess of truncating and missense mutations, respectively, over the neutral expectation calculated from the observed synonymous variants (second row). The magnitude of the neutral expectation is indicated inside of the horizontal bars with shaded diagonal lines and the p-value corresponds to the Omega implementation of dN/dS (Supplementary Note 6). Fourth row, 3D clustering score of missense mutations (blue line) compared to the neutral expectation (gray line), along with detected 3D clusters of missense mutations (filled light blue line). The right-hand panel represents the distance between the distribution of expected 3D clustering scores of the residue with the highest observed score (gray) and the observed score itself (vertical dashed lines), used to compute an empirical p-value (Supplementary Note 6). Fifth row, functional impact score of SNVs (synonymous, missense and truncating) observed in the protein. The right-hand panel represents the distance between mean expected functional impact scores (gray areas) and the observed average functional impact score (vertical dashed lines), used to compute an empirical p-value (Supplementary Note 6). Sixth row, deviation in the ratio of frameshift (purple) to inframe (brown) indels in the gene compared to the ratio of non-3n to 3n (a length multiple of three nucleotides) indels in neighboring non-coding regions (excess of frameshift

indels). The right-hand bars represent the numbers of coding frameshift and inframe indels and non-coding non-3n to 3n indels. Analytical or empirical tests used to calculate the p-values shown in the different panels are described in Supplementary Note 6. P-values for all genes computed using all methods appear in Supplementary Table 6. b) Magnitude of all signals of positive selection calculated for 14 genes on the pooled mutations of the 79 samples. Dashed lines for truncating, missense and indels indicate an equal number of observed and expected mutations of each type (dN/dS =1, that is, no selection). In 3D clustering the size of the circles is proportional to the difference between observed and expected scores. In functional impact bias, the size of the circles is proportional to the Z-score. c) Comparison of the excess of truncating mutations (dN/dS truncating, top) and the excess of missense mutations (dN/dS missense) calculated taking into account every observed mutation only once (as used in the manuscript, and represented by the unshaded bars in each plot) and taking into account the number of DNA duplex reads supporting each mutation (bars shaded with diagonal line pattern). d) Positive selection on mutations in *PIK3CA*. Left panel, needleplot representing the distribution of missense, truncating and synonymous mutations in the region of *PIK3CA* covered by sequencing reads. Right panel, magnitude of positive selection on missense mutations across the 79 samples calculated using the Omega dN/dS approach. The p-value is calculated using the Omega implementation of the dN/dS approach described in Supplementary Note 6. The p-value appears in Supplementary Table 6.

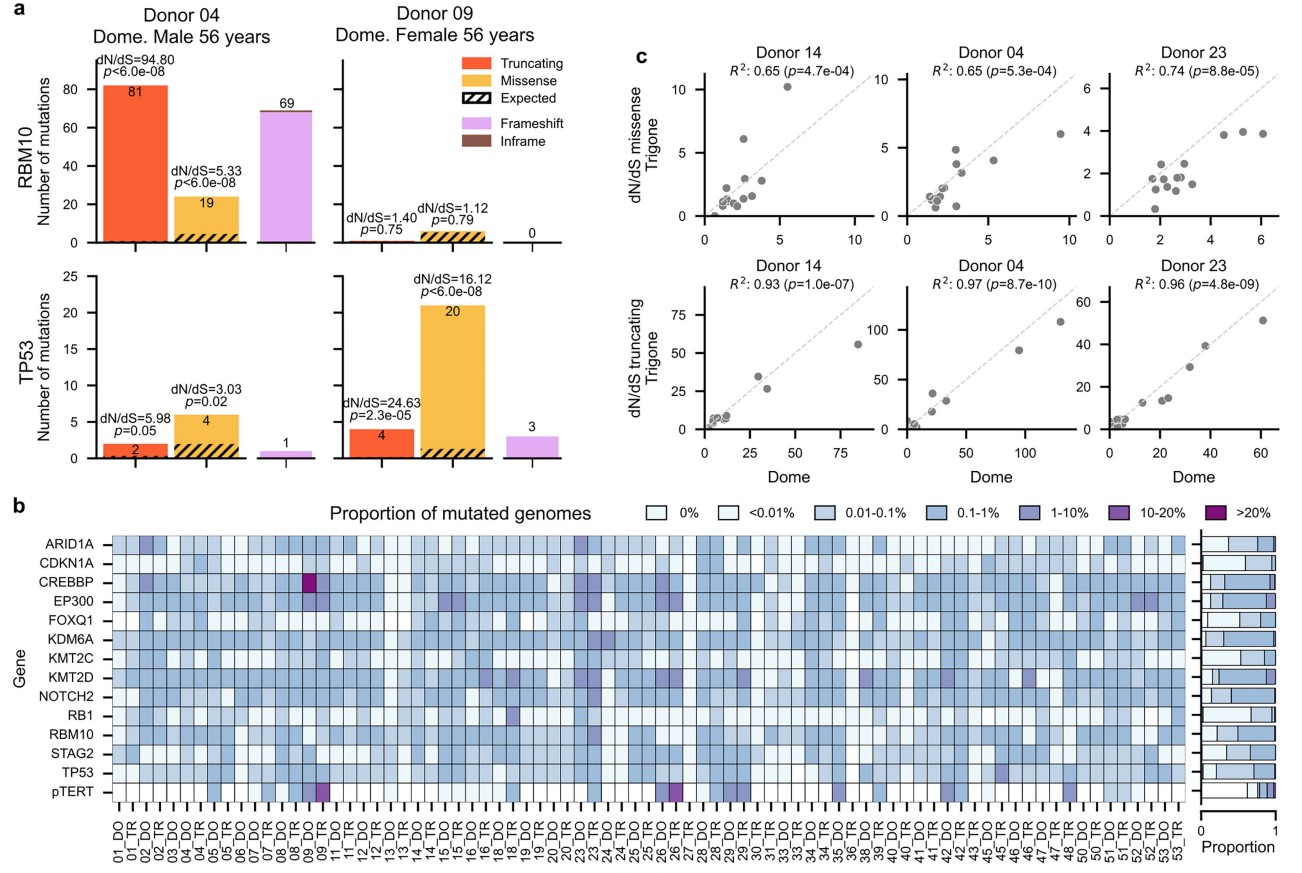

**Extended Data Fig. 5 | Calculation of positive selection at the sample level.**
a) dN/dS truncating values for *RBM10* and *TP53* in the dome of donors 04 and 09.
All legends as defined in Fig. 2. The p-values are calculated using the Omega
implementation of the dN/dS approach described in Supplementary Note 6, and
appear in Supplementary Table 6. b) Landscape of the fraction of urothelium
covered by driver mutations of each gene in the panel. The values of covered
urothelium have been discretized. In most cases, the percentage of urothelium
covered by driver mutations of each gene falls in the lowest categories.

The right-hand graph shows stacked barplots with the distribution of samples
in different categories of covered urothelium across genes. c) Agreement of the
magnitude of dN/dS missense (top) and dN/dS truncating (bottom) calculated
for all genes in the dome and trigone of donors 14, 04 and 23. R-squared ($R^2$),
Pearson's correlation coefficient of the dN/dS values calculated for the dome
and trigone samples from each donor. The p-values corresponding to the
Pearson's correlation coefficients are shown.

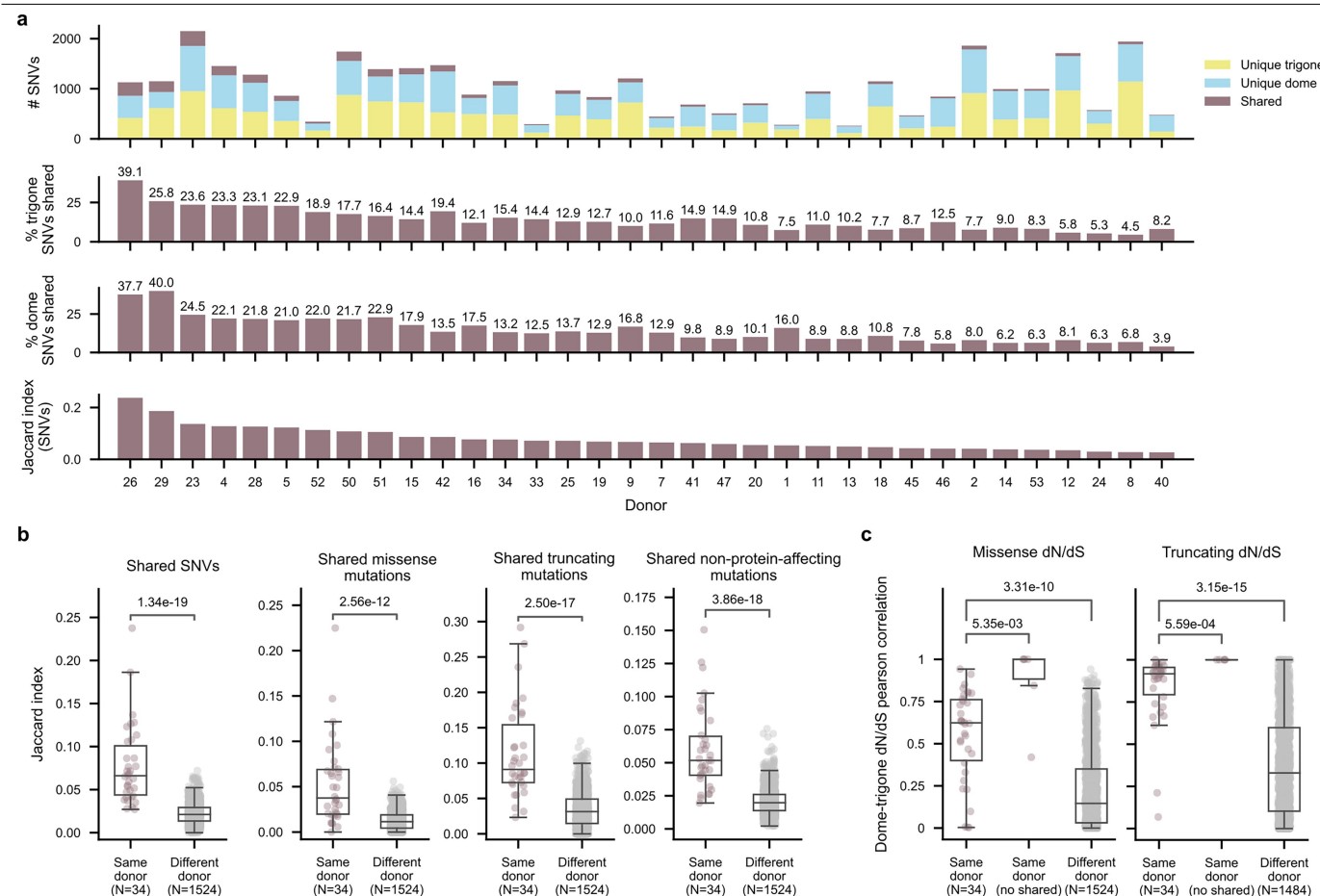

**Extended Data Fig. 6 | Similarity between dome and trigone.** a) Top, number of SNVs that are shared between the dome and trigone samples (or unique to each of them) of donors for which both areas were brushed. Bottom, percentage of SNVs found in the trigone sample of each individual that are shared with the dome sample of the same individual (top), of SNVs found in the dome that are shared with the trigone (middle), and Jaccard index measuring the overlap of the SNVs identified within both samples (bottom). b) Distribution of Jaccard Index values of SNVs (first at the left), missense mutations (second), truncating mutations (third), and non-protein affecting mutations (last to the right) shared between the dome and trigone samples of the same individual and pairs of samples from different donors. The Jaccard index obtained for any subset of mutations is significantly higher for the dome and trigone samples of the same donor (p-values from one-tailed Wilcoxon-Mann–Whitney test). N indicates the number of sample pairs. c) Comparison of the distribution of Pearson's correlation coefficients comparing dN/dS values between dome and trigone samples of the same donor (as done in Extended Data Fig. 5c) or from different donors. In the first boxplot, all mutations are included in the calculation of dN/dS values, while in the second, mutations shared between dome and trigone of the same donor are excluded. The correlation is significantly higher between dome-trigone pairs of samples of the same donor than of different donors (p-values from one-tailed Wilcoxon-Mann–Whitney test). Only pairs of samples for which Omega values of at least two genes could be computed are included in the boxplots. N indicates the number of sample pairs. Box plots in b and c display the quartiles with whiskers extending to the highest and lowest data points within 1.5 times the interquartile range.

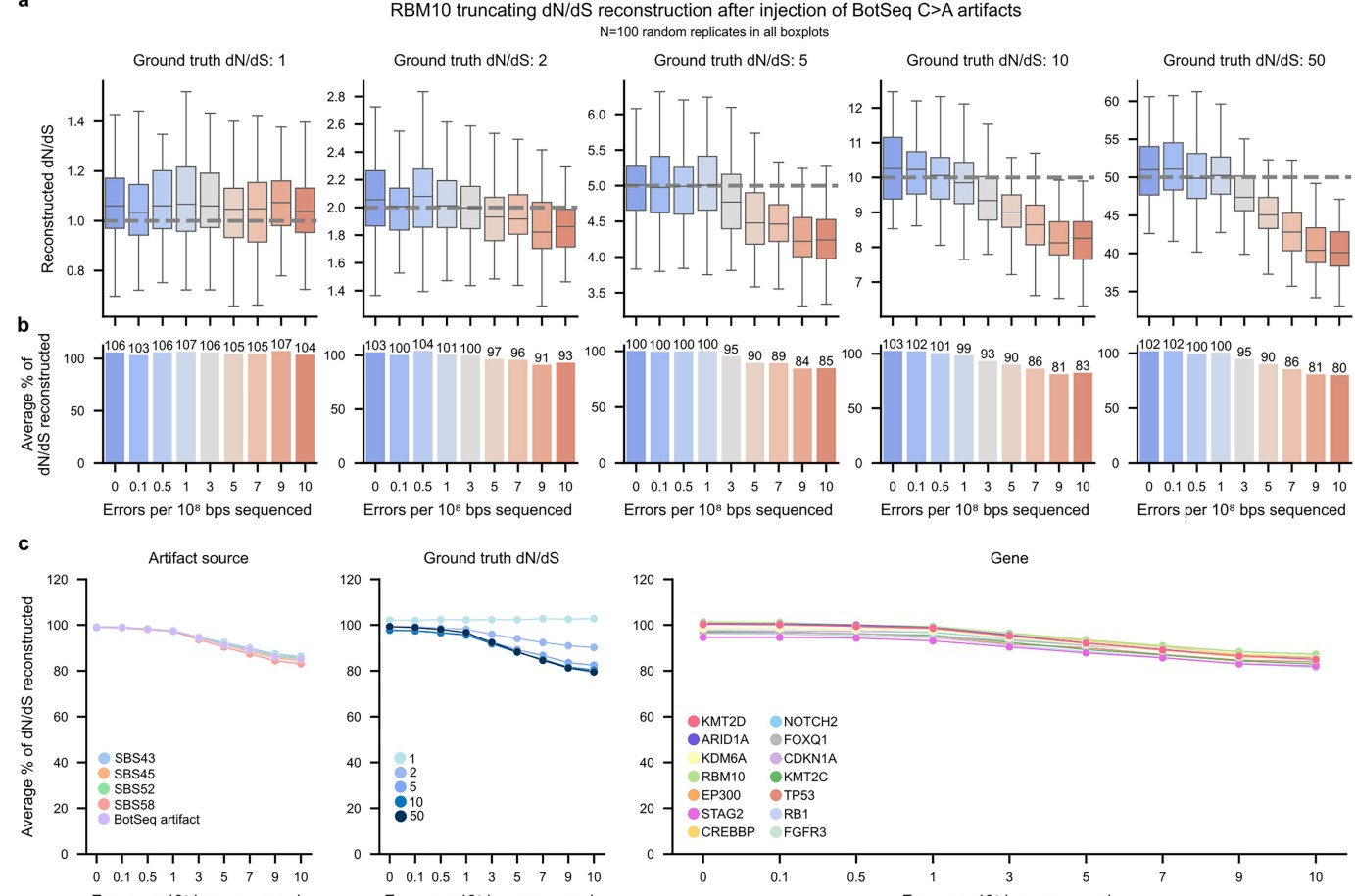

**Extended Data Fig. 7 | Tolerance of dN/dS values to errors. a)** Measurement of the tolerance of *RBM10* dN/dS truncating to artifactual mutations (artifacts) following the BotSeq mutational profile[82]. The boxplots represent the distribution of dN/dS values calculated from 100 synthetic samples with increasing rates of injected artifacts between 0 and $1 \times 10^{-7}$ (one order of magnitude higher than estimated for the technology), and for increasing values of ground truth dN/dS (between 1 and 50). The boxplots display the quartiles with whiskers extending to the highest and lowest data points within 1.5 times the interquartile range. N = 100 synthetic samples. **b)** Average percentage of *RBM10* dN/dS truncating reconstructed value across 100 synthetic samples, calculated by computing which fraction of the ground truth dN/dS in a sample is obtained upon calculation. **c)** Summary of the results of the experiment of

error tolerance. Left panel, average percentage of reconstructed ground truth dN/dS across synthetic samples (for all genes and all ground truth dN/dS explored altogether) that is calculated upon injection of increasing rates (x-axis) of different types of artifacts (color legend). Center plot, average percentage of reconstructed ground truth dN/dS across synthetic samples (for all genes and all artifacts altogether) that is calculated upon injection of increasing rates (x-axis) for different values of ground truth dN/dS (color legend). Right panel, average percentage of reconstructed ground truth dN/dS across synthetic samples (for all ground truth dN/dS explored and all artifacts altogether) for different genes (color legend), that is calculated upon injection of increasing rates (x-axis) of artifacts.

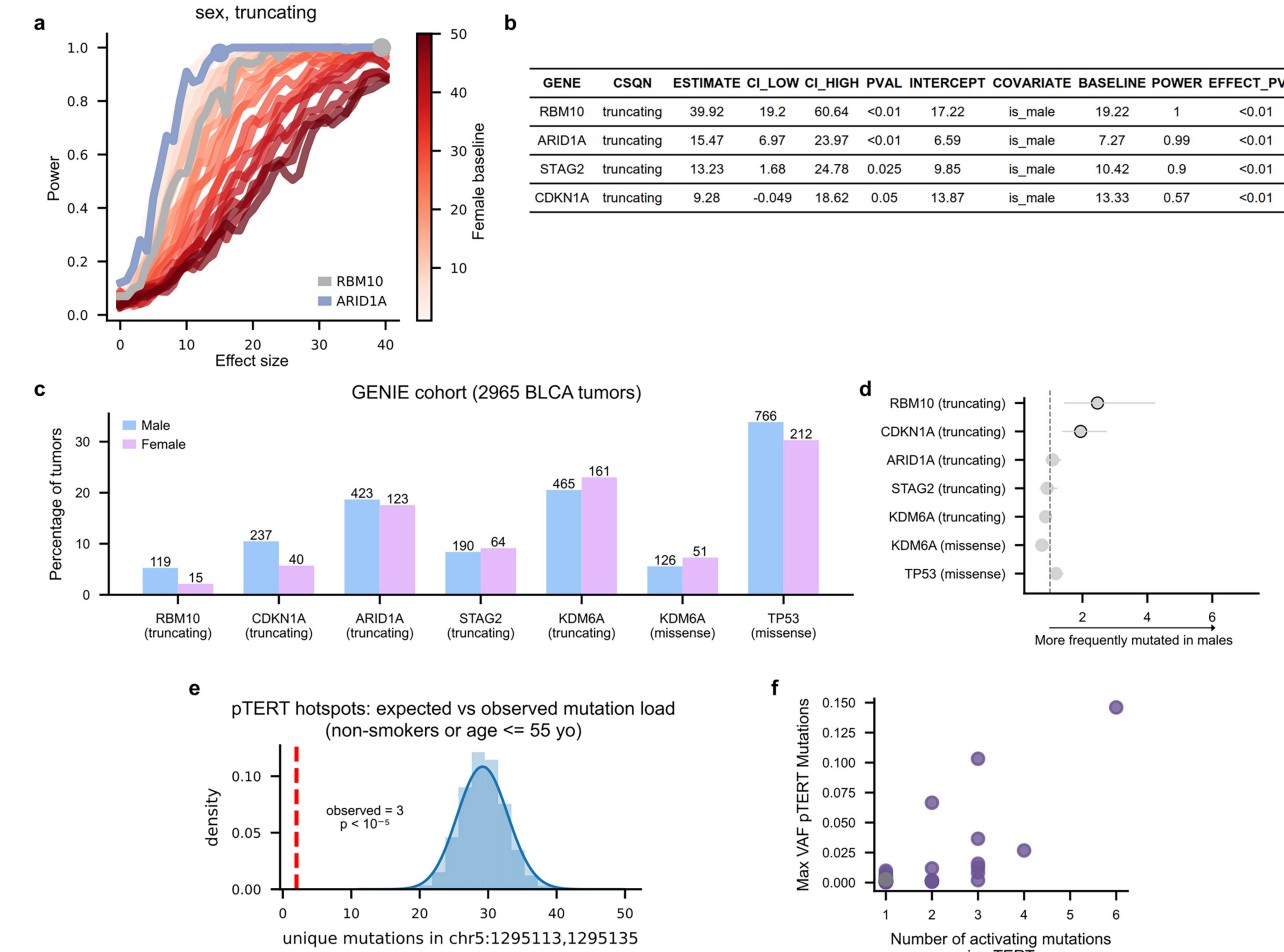

| GENE | CSQN | ESTIMATE | CI_LOW | CI_HIGH | PVAL | INTERCEPT | COVARIATE | BASELINE | POWER | EFFECT_PVAL |
|------|------|----------|--------|---------|------|-----------|-----------|----------|-------|-------------|
| RBM10 | truncating | 39.92 | 19.2 | 60.64 | <0.01 | 17.22 | is_male | 19.22 | 1 | <0.01 |
| ARID1A | truncating | 15.47 | 6.97 | 23.97 | <0.01 | 6.59 | is_male | 7.27 | 0.99 | <0.01 |
| STAG2 | truncating | 13.23 | 1.68 | 24.78 | 0.025 | 9.85 | is_male | 10.42 | 0.9 | <0.01 |
| CDKN1A | truncating | 9.28 | -0.049 | 18.62 | 0.05 | 13.87 | is_male | 13.33 | 0.57 | <0.01 |

**Extended Data Fig. 8 | Heterogeneous clonal landscape and power calculation.** a) From simulated datasets reflecting the same distributional features and data dependencies found in the study cohort, we computed the statistical power as the proportion of times the variable of interest (sex) came out significant in the univariate linear mixed-effects regression against truncating dN/dS. In this analysis the female group was picked as the baseline group. We simulated data with different ground truth female baselines (expected truncating dN/dS among females) and between-group differences (effect size). For each baseline-effect combination we can draw a power value, which are represented collectively in the form of these power profiles (see Supplementary Note 10). For the two exemplary genes *RBM10* and *ARID1A* we highlight the profile curves corresponding to their observed baselines in the cohort and the projected power given the inferred effect in the cohort. b) Table presenting a summary of the five associations found between dN/dS and sex in the study. Here we briefly define the meaning of each column. See also Supplementary Note 10 for a more in-depth account on the methodology. CSQN: Either missense or truncating, represents the specific dN/dS used as response variable in the association analysis. ESTIMATE: Coefficient of the binary variable of interest ("is_male") inferred via linear-mixed effects regression against dN/dS using the donor as a random intercept. CI_LOW, CI_HIGH: Lower and upper 95% CI bounds of ESTIMATE. PVAL: p-value associated with the variable of interest in the regression analysis. INTERCEPT: Inferred intercept in the regression analysis. BASELINE: Average CSQN-specific dN/dS value in the baseline group of samples (female). INTERCEPT and BASELINE are expected to follow closely one another. COVARIATE: The (binary) explanatory variable representing sex. POWER: Statistical power corresponding to the BASELINE and ESTIMATE in the power profile. EFFECT_PVAL: The "effect p-value" is an ad-hoc metric that we defined as the proportion of times the sex coefficient attains a value at least as high as ESTIMATE upon regression with a dataset corresponding to BASELINE and zero ground-truth effect. It can be thought of as an effect-aware false positive rate. c) Frequency of tumor samples with missense or truncating mutations of 6 genes in males and females across a cohort of 2,965 bladder carcinomas from the GENIE cohort. d) Multivariate logistic regression (including age) of sex on mutations in the 6 genes. Circles represent the point estimate of the effect size of the linear regression, and the horizontal line, the 95% confidence intervals. Circles with dark outer circumference denote significant associations (FDR threshold of 0.2). e) Distribution of expected number of mutations in the two *TERT* promoter mutational hotspots (chr5:1295113 and chr5:1295135) across donors younger than 55 years old or never smokers assuming a mutation rate equal to that observed across ever smokers older than 55 years old. The red dashed vertical line represents the actual observed number of mutations in the two hotspots across donors younger than 55 years or never smokers. The p-value was calculated empirically based on 10,000 randomizations, as described in Supplementary Note 10. f) Maximum variant allele frequency detected for activating *TERT* promoter mutations in a sample vs the number of activating *TERT* promoter mutations (i.e. mutations observed in tumors, see main text) identified in a sample. The observation of different activating *TERT* promoter mutations in the same sample indicates the existence of convergent evolution of *TERT* promoter mutations. This, in turn, suggests that the observation of mutations with large variant allele frequency may also represent multiple mutated clones with the exact mutations (convergent evolution) rather than very large clones.

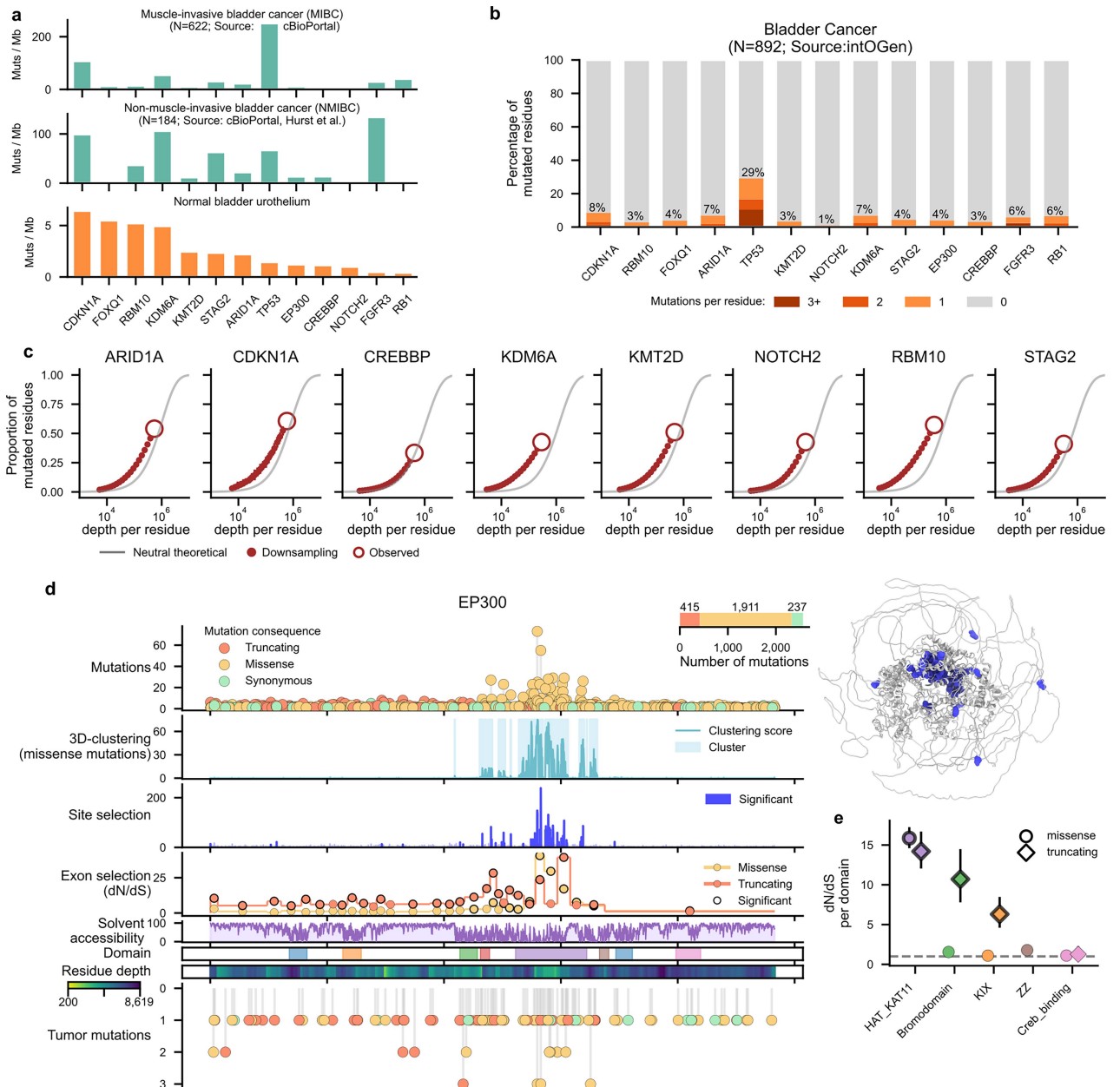

**Extended Data Fig. 9 | Calculations of natural selection mutagenesis.**
a) Comparison of the density of protein affecting mutations in 14 genes across two cohorts of bladder tumors (muscle invasive and non-muscle invasive) and in the normal urothelium of the 45 donors. Mutation density in tumors is calculated by dividing the number of observed mutations (normally 1) by the gene length in megabases (Mb). b) Percentage of amino acid residues in each gene with zero, one, two, or three or more mutations observed across 892 bladder tumor samples from the intOGen cohort. The order of the genes is as in Fig. 5a to facilitate visual comparison. c) Theoretical and observed curves of saturation mutagenesis for genes not shown in Fig. 5b. The grey dashed line represents the kinetic of saturation mutagenesis under the theoretical assumption of no selection, in which mutations are observed based only on their neutral probability of occurrence. The red circle denotes the degree of saturation achieved by probing the 79 samples in the cohort. The red dashed line is

constructed through successive depth down-samples of the current observation and represents the observed kinetic of natural saturation mutagenesis (see details in Supplementary Note 12). d) Natural saturation mutagenesis of *EP300* in normal bladder urothelium. Besides the tracks described in Fig. 5c for *TP53*, 3D clusters obtained via Oncodrive3D (second), dN/dS truncating and dN/dS missense values for each exon (fourth), and the distribution of tumor mutations (from intOGen; see Methods) along the sequence of the gene (last) have been added. These same types of plots are presented for the rest of genes in the study in Supplementary Figs. 2 and 3. Right plot, EP300 3D structure with residues with significant site selection highlighted in blue. e) dN/dS truncating and dN/dS missense values for each domain of EP300. The vertical lines represent the 95% confidence intervals of the dN/dS estimate. Solid border represents significant dN/dS values (p-value < 0.05) according to Omega (Supplementary Note 6). N = 79 samples.

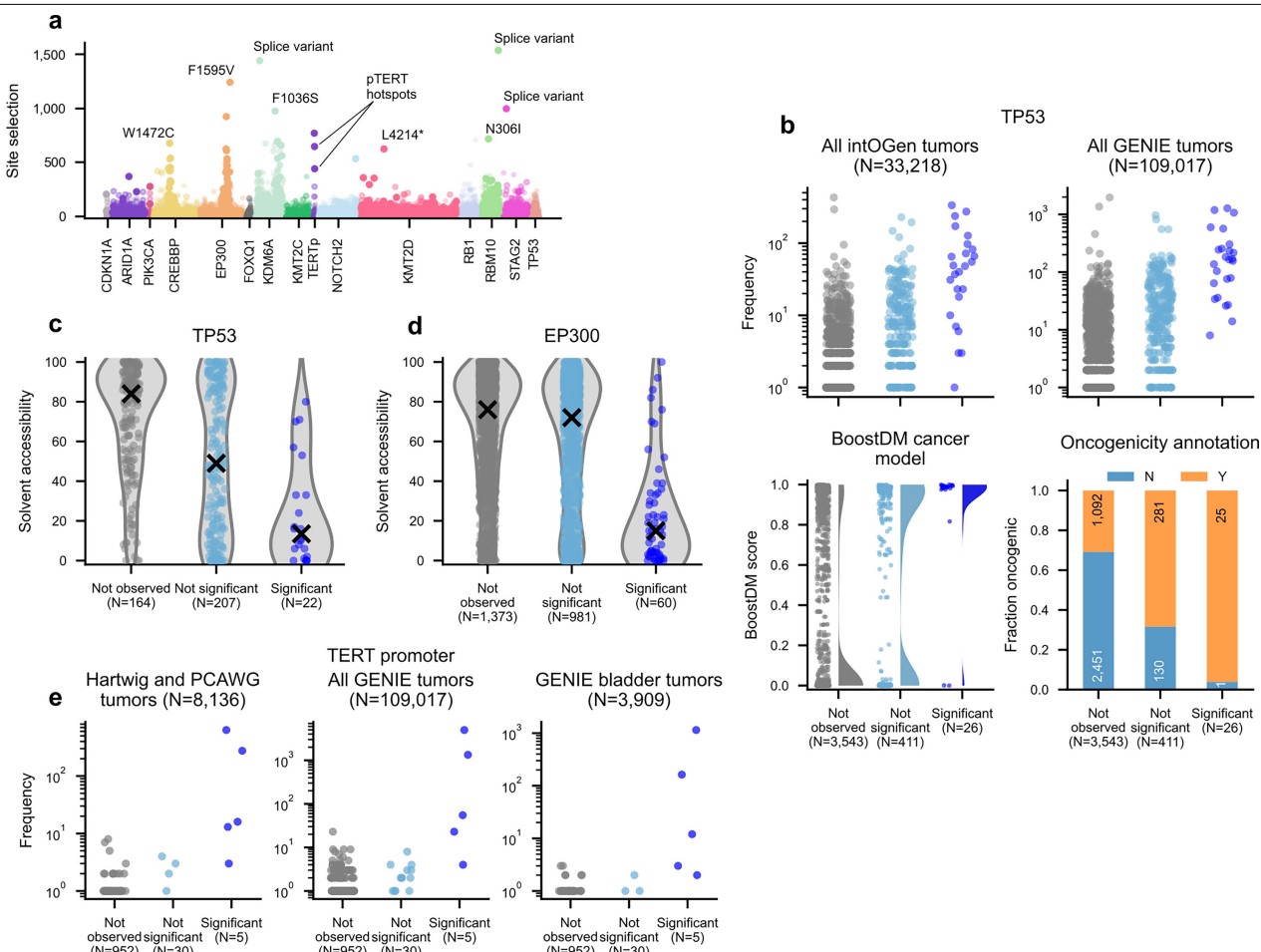

**Extended Data Fig. 10 | Application of natural saturation mutagenesis.**
a) Manhattan plot illustrating the strength of site selection for all genomic sites included in the sequencing panel. Some of the mutations in the sites with strongest selection are indicated. b) Application of site selection values to *TP53* mutations. In all plots, *TP53* tumor mutations observed in two large cohorts (intOGen, N = 33,218 and GENIE, N = 109,017) are grouped depending on whether they have been observed across bladder normal samples in this study and their site selection (i.e., not observed, observed with non-significant site selection and observed with significant site selection). Left top panel, distribution of the frequency across intOGen tumors of the three groups of mutations. Right top panel, distribution of the frequency across GENIE tumors. Left bottom panel, boostDM (machine learning models for in silico saturation mutagenesis)[46] scores of the three groups of mutations. Right bottom panel, proportion of

mutations annotated or not annotated as oncogenic in ClinVar[83] or OncoKB[84]. c,d) Distribution of the solvent accessibility of sites with mutations in each group for *TP53* (c) and *EP300* (d). e) Application of site selection values to *TERT* promoter mutations. Mutations observed in two large cohorts of tumors (or the subset of bladder tumors in GENIE) are grouped depending on whether they have been observed across normal samples and their site selection into not observed, observed with non-significant site selection and observed with significant site selection. Left, distribution of the frequency across tumors in a large cohort of whole-genome sequenced samples (N = 8,136) of the three groups of mutations. Center, distribution of the frequency across all GENIE tumors (N = 109,017) of the three groups of mutations. Right, distribution of the frequency across GENIE bladder tumors (N = 3,909) of the three groups of mutations.

# Reporting Summary

## Statistics

For all statistical analyses, confirm that the following items are present in the figure legend, table legend, main text, or Methods section.

| n/a | Confirmed | |
|---|---|---|
| ☐ | ☒ | The exact sample size (*n*) for each experimental group/condition, given as a discrete number and unit of measurement |
| ☒ | ☐ | A statement on whether measurements were taken from distinct samples or whether the same sample was measured repeatedly |
| ☐ | ☒ | The statistical test(s) used AND whether they are one- or two-sided<br>*Only common tests should be described solely by name; describe more complex techniques in the Methods section.* |
| ☐ | ☒ | A description of all covariates tested |
| ☐ | ☒ | A description of any assumptions or corrections, such as tests of normality and adjustment for multiple comparisons |
| ☐ | ☒ | A full description of the statistical parameters including central tendency (e.g. means) or other basic estimates (e.g. regression coefficient) AND variation (e.g. standard deviation) or associated estimates of uncertainty (e.g. confidence intervals) |
| ☐ | ☒ | For null hypothesis testing, the test statistic (e.g. *F*, *t*, *r*) with confidence intervals, effect sizes, degrees of freedom and *P* value noted<br>*Give P values as exact values whenever suitable.* |
| ☒ | ☐ | For Bayesian analysis, information on the choice of priors and Markov chain Monte Carlo settings |
| ☒ | ☐ | For hierarchical and complex designs, identification of the appropriate level for tests and full reporting of outcomes |
| ☐ | ☒ | Estimates of effect sizes (e.g. Cohen's *d*, Pearson's *r*), indicating how they were calculated |

*Our web collection on statistics for biologists contains articles on many of the points above.*

## Software and code

Policy information about availability of computer code

Data collection

DNA extracted from 79 samples obatained by brusing one or two sites (2-3cm2) of the bladder of 45 donors upon autopsy was sequenced using a duplex DNA sequencing technology with commercially available kits (TwinStrand Biosciences, Seattle, WA). Sample processing and the experimental protocol used in the duplex sequencing are described in detail in the Methods section and in the Supplementary Note 2.

DNA extracted from 3 cord blood samples was also obtained and sequenced using the same DNA sequencing technology. This is described in the Supplementary Note 4.

Clinical data for the 45 donors, including age, sex, BMI, tobacco smoking history, alcohol use, prior cancer, and chemotherapy exposure was also obtained.

Mutations identified across 622 muscle invasive and 105 non-muscle invasive bladder cancer (MIBC and NMBIC, respectively) cohorts were downloaded from cBioPortal together with the clinical data of the combined study. From the MIBC BGI cohort only samples labeled as invasive were considered MIBC and added to the MIBC dataset. Mutations identified in an additional cohort of 79 NMIBCs were obtained. from the literature and included as part of the NMIBC dataset. Mutations identified across 33,218 tumors (892 bladder tumors) in intOGen were downloaded from intogen.org. These mutations were used to obtain the total number of mutations observed in each gene, their distribution along the sequence of the genes in the study, and the percentage of sites affected by different numbers of mutations. The same data of 109,017 tumors (3,909 bladder tumors) were obtained from the GENIE project to calculate the frequency of mutations in each of the genes and the TERT promoter. Mutations in the TERT promoter were also obtained from two cohorts of tumors (included in intOGen) sequenced at the whole genome level (Hartwig Medical Foundation: N=5,582 and PCAWG: N=2,554). The classification (and score) of all possible mutations in TP53 into drivers and passengers via in silico saturation mutagenesis was obtained from boostDM (intogen.org/boostDM). Details of analyses involving tumor mutations appear in Supplementary Note 11.

Structural models for all proteins used to run Oncodrive3D were obtained from the AlphaFold database (AlphaFold 2 v.4), as were the structural features of proteins. Solvent accessibility and secondary structure information were extracted from the AlphaFold-predicted PDB structures using PDB_Tool (https://github.com/realbigws/PDB_Tool).

We obtained two saturation mutagenesis experiments estimating the functional impact of mutations in the TP53 DNA binding domain and along the sequence of the TERT promoter from their original publications, cited in ther Methods section.

Data analysis | Somatic mutations from the raw sequencing data were called using a computational pipeline (deepUMIcaller) implemented by us in Nextflow on the basis of an early version of nf-core/fastquorum64 pipeline, which implements the fgbio Best Practices FASTQ to Consensus Pipeline (https://github.com/fulcrumgenomics/fgbio/blob/main/docs/best-practice-consensus-pipeline.md) and downstream variant calling based on VarDictJava (https://github.com/AstraZeneca-NGS/VarDictJava). A series of filters to discard potential artifacts are included in the pipeline. The code implementing deepUMIcaller is publicly available at (github.com/bbglab/deepumicaller) upon publication. For a detailed description of the pipeline, see Supplementary Note 3.

A second pipeline, deepCSA was implemented by us also in Nextflow to automate all downstream analyses caried out on somatic mutations. This second pipeline is available at https://github.com/bbglab/deepCSA.

The analyses implemented in deepCSA include:

-Extraction of mutational signatures de novo using Bayesian hierarchical Dirichlet process using HDP_sigExtraction pipeline (https://github.com/McGranahanLab/HDP_sigExtraction) based on the R-package hdp developed by Nicola Roberts (https://github.com/nicolaroberts/hdp)9 and SigProfilerExtractor (v.1.2.1) (https://github.com/AlexandrovLab/SigProfilerExtractor).
-Four methods to compute positive selection on the mutations observed across genes are employed in this article. One of them (omega), a dN/dS approach to assess the strength of selection on the mutational pattern of genes, was developed de novo for this study and is described at length in Supplementary Note 6 (https://github.com/bbglab/omega). Two others, OncodriveFML and Oncodrive3D, which compute the deviation in the average functional impact and clustering in the three-dimensional structure of proteins, respectively, from those expected under neutrality, had been developed previously, and were adapted here to work on duplex sequencing data. These are also described in Supplementary Note 6. A fourth method, assessing the relative enrichment for frameshift indels observed across genes was also developed de novo for this study and is thoroughly described in Supplementary Note 6.
-Mixed-effects linear models were implemented to test the association between clinical variables and the clonal structure of samples, represented by the magnitude of positive selection on mutations of different genes. Associations with FDR below 0.2 were deemed significant. We also carried a binomial test to rule out a spurious dependence of the mutation density on group differences in terms of the sequencing depth. All statistical details of these models and tests carried out to assess their power are explained in Supplementary Notes 9 and 10.

We also designed a method to compute the natural saturation mutagenesis kinetics of genes across samples, described in details in Methods and Supplementary Note 12.

For manuscripts utilizing custom algorithms or software that are central to the research but not yet described in published literature, software must be made available to editors and reviewers. We strongly encourage code deposition in a community repository (e.g. GitHub). See the Nature Portfolio guidelines for submitting code & software for further information.

# Data

Policy information about availability of data

All manuscripts must include a data availability statement. This statement should provide the following information, where applicable:

- Accession codes, unique identifiers, or web links for publicly available datasets
- A description of any restrictions on data availability
- For clinical datasets or third party data, please ensure that the statement adheres to our policy

Raw sequencing data for this study were deposited in dbGaP under accession number phs004105.v1.p1. The set of mutations used in all analyses presented in the paper is available at https://doi.org/10.5281/zenodo.15836679. Executing the code provided in the third repository mentioned below, all figures in the paper can be reproduced. Reference mutational signatures were obtained from https://cancer.sanger.ac.uk/signatures/sbs/. Tumor mutations were obtained through cBioPortal (datasets from references16,17,74 at https://www.cbioportal.org/), intogen (intogen.org) and the GENIE synapse data portal (https://genie.synapse.org/) as described in Methods. Protein structural models for the entire human proteome were obtained from the AlphaFold database (https://alphafold.ebi.ac.uk/). The results of two experimental saturation mutagenesis studies on TP53 and the TERT promoter were obtained from references51,55, respectively.

# Research involving human participants, their data, or biological material

Policy information about studies with human participants or human data. See also policy information about sex, gender (identity/presentation), and sexual orientation and race, ethnicity and racism.

Reporting on sex and gender | One of the most salient results of the study is the difference in the clonal landscape of the urothelium between males and females.

Reporting on race, ethnicity, or other socially relevant groupings | N/A

Population characteristics | Described in Supplementary Note 2 and Extended Data Table 1

| Recruitment | Samples were obtained from deceased individuals without known bladder pathology and no history of bladder cancer upon autopsy at the University of Washington after obtaining consent from next-of-kin. |
|---|---|
| Ethics oversight | The study was deemed not human subjects by the Institutional Review Board at University of Washington (STUDY00016707) because research involving deceased individuals is not human subjects research by US Federal guidelines (45CFR46). |

Note that full information on the approval of the study protocol must also be provided in the manuscript.

# Field-specific reporting

Please select the one below that is the best fit for your research. If you are not sure, read the appropriate sections before making your selection.

☒ Life sciences ☐ Behavioural & social sciences ☐ Ecological, evolutionary & environmental sciences

For a reference copy of the document with all sections, see nature.com/documents/nr-reporting-summary-flat.pdf

# Life sciences study design

All studies must disclose on these points even when the disclosure is negative.

| Sample size | 79 samples from 45 deceased individuals (53 before exclusions). No prior sample size calculation was carried out; however, we conclusively demonstrate (Supplementary Note 10) that the sample size of the cohort provides the statistical power required to detect the associations reported in the manuscript. |
|---|---|
| Data exclusions | Samples from three individuals with active or chronic inflammation and four individuals with insufficient DNA were discarded. The two samples of a separate individual with cystitis and evidence of a large proportion of artifacts among mutations were also discarded. |
| Replication | All samples in the cohort were used in discovery. Nevertheless, controls with subsets of the samples were carried out. |
| Randomization | All samples in the cohort were used in all main analyses. Nevertheless, in particular analyses in the manuscript, such as the determination of the conservation of positive selection across dome and trigone samples, and the evaluation of the impact of errors on the analyses, randomization of samples or sets of mutations was carried out. |
| Blinding | All samples in the cohort were used in all main analyses, precluding the need for blinding. |

# Reporting for specific materials, systems and methods

We require information from authors about some types of materials, experimental systems and methods used in many studies. Here, indicate whether each material, system or method listed is relevant to your study. If you are not sure if a list item applies to your research, read the appropriate section before selecting a response.

## Materials & experimental systems

| n/a | Involved in the study |
|---|---|
| ☐ | ☒ Antibodies |
| ☒ | ☐ Eukaryotic cell lines |
| ☒ | ☐ Palaeontology and archaeology |
| ☒ | ☐ Animals and other organisms |
| ☒ | ☐ Clinical data |
| ☒ | ☐ Dual use research of concern |
| ☒ | ☐ Plants |

## Methods

| n/a | Involved in the study |
|---|---|
| ☒ | ☐ ChIP-seq |
| ☒ | ☐ Flow cytometry |
| ☒ | ☐ MRI-based neuroimaging |

## Antibodies

| Antibodies used | CK7. Supplier: Agilent (Dako). Clone name: OV-TL 12/30. Catalog number: M7018. Lot number: 41491921.<br>Uroplakin. Supplier: Biocare Medical. Clone name: BC21. Catalog number: AP13051 AA. Lot number: 022724A.<br>CD45. Supplier: Agilent (Dako). Clone name: 2B11&PD7/26. Catalog number: IR751. Lot number: 41718528.<br>Smooth Muscle Actin. Supplier: Cell Marque. Clone name: 1A4. Catalog number: 202M-95. Lot number: 323477. |
|---|---|
| Validation | CK7. Intended use: For in vitro diagnostic use by immunohistochemistry. Species reactivity: Human. Validation: Monoclonal Mouse Anti-Human Cytokeratin 7 consistently labels a large number of simple-, complex- and transitional epithelia, including all cell layers of urothelium (transitional epithelium) (1).<br><br>Uroplakin. Intended use: For in vitro diagnostic use by immunohistochemistry. Species reactivity: Human. Validation: UPII and UPIII may be found in the urothelial surface membrane of human renal pelvis, ureter, bladder and urethra. UPII and UPIII have also been identified as sensitive and highly specific markers for urothelial carcinoma (2-5). New mouse monoclonal antibodies to UPII, clone |

BC21, and UPIII, clone BC17, have been developed and evaluated for sensitivity in urothelial carcinoma and specificity versus normal and neoplastic tissues (6).

CD45. Intended use: For in vitro diagnostic use by immunohistochemistry. Species reactivity: Human. Validation: Anti-CD45 is a mixture of two monoclonal antibodies, clones 2B11 and PD7/26, directed against different epitopes. Clone 2B11 was clustered as anti-CD45 at the Third International Workshop and Conference on Human Leucocyte Differentiation Antigens and reacts with all the known isotypes of the CD45 family (7). Clone PD7/26 was clustered as anti-CD45RB at the Fifth International Workshop and Conference on Human Leucocyte Differentiation Antigens (8).

Smooth Muscle Actin. Intended use: For in vitro diagnostic use by immunohistochemistry. Species reactivity: Human. Validation: Anti-smooth muscle actin immunohistochemical reactivity is seen in smooth muscle cells, myofibroblasts and myoepithelial cells (9-11). The antibody was validated by Cell Marque in a collection of normal tissues and demonstrated positive staining for smooth muscle and myoepithelium.

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

# Plants

| | |
|---|---|
| Seed stocks | Report on the source of all seed stocks or other plant material used. If applicable, state the seed stock centre and catalogue number. If plant specimens were collected from the field, describe the collection location, date and sampling procedures. |
| Novel plant genotypes | Describe the methods by which all novel plant genotypes were produced. This includes those generated by transgenic approaches, gene editing, chemical/radiation-based mutagenesis and hybridization. For transgenic lines, describe the transformation method, the number of independent lines analyzed and the generation upon which experiments were performed. For gene-edited lines, describe the editor used, the endogenous sequence targeted for editing, the targeting guide RNA sequence (if applicable) and how the editor was applied. |
| Authentication | Describe any authentication procedures for each seed stock used or novel genotype generated. Describe any experiments used to assess the effect of a mutation and, where applicable, how potential secondary effects (e.g. second site T-DNA insertions, mosiacism, off-target gene editing) were examined. |

