## [Peer Review File · Nature]

Sex and Smoking Bias in the Selection of Somatic Mutations in Human Bladder

Corresponding Author: Professor Nuria Lopez-Bigas

Version 0:

Reviewer comments:

Referee #1

(Remarks to the Author)

Summary:

In this paper, Martinez-Illescas et al. perform targeted duplex sequencing of 16 genes (selected for their known status as driver genes in normal bladder urothelium and bladder cancer) with very high depth in 73 bladder urothelium samples from 42 individuals. This approach allows detection of mutations present in very small clones or even single cells without the laborious work of dissecting individual clones one by one. Thus, they effectively interrogated about half a million cells, many more than the < 1,000 bladder cancer genomes across prior cancer sequencing studies (each cancer interrogates effectively a single clone) and a prior study of bladder mutagenesis by 2,097 microbiopsies (Lawson, et al, PMID 33004514). The authors key findings are: a) identifying > 70,000 mutations that begin to approach saturation mutagenesis of the assayed genes and confirmed the driver gene status via positive selection of 13 of the genes in the panel and negative selection occurring in 1 of the genes in the panel; b) confirmation of the mutational signatures prevalent in bladder urothelium, including an APOBEC signature as well as a signature likely related to the deceased subjects' history of chemotherapy; c) similar patterns of selection in structurally disparate regions of the bladder within individuals, while different patterns of selection between individuals, suggesting that individual specific germline or environmental differences shape selection across the bladder; d) association of more mutations in four genes and in the TERT promoter in males versus females, and association of TERT promoter mutations with smoking history, which may be partly explanatory of the known higher risk of bladder cancer in males; e) identification of some contrasting patterns of mutation in normal bladder urothelium versus bladder cancer that suggests that some healthy tissue mutation/selection evolutionary paths are more prone to cancer formation.

General comments:

Overall, the paper is well written and the computational analysis was fairly rigorous. An important reference study for this paper is Lawson, et al (Science 2020, PMID 33004514). The two studies differ in two key ways: the Lawson, et al study by another group interrogated an effectively much smaller number of genomes than this study (~4,000 haploid genomes from ~2,000 biopsies from 20 individuals vs ~half a million haploid genomes in this study from 42 individuals in this study), but on the other hand, Lawson, et al interrogated many more genes than this study (a mix of ~320 cancer genes, exome-wide, and genome-wide analyses). The prior Lawson, et al study also has some overlapping findings with this study, including: identification of most of the driver genes active in urothelium which helped this study design the target panel, variation in driver genes across individuals, and APOBEC mutagenesis. This comparison indicates that the key novel findings of the current study are the very high number of mutations identified in the targeted driver genes that paint a more clear picture of mutation and selection in the bladder urothelium, identification of a few novel driver genes in normal urothelium (KMT2C, RB1, FGFR3) as well as hotspot mutations in PIK3CA and the TERT promoter, and also significantly strengthening the prior study's finding of different selection processes across individuals including the effect of sex. I believe these findings will be impactful for understanding how pervasive mutation is in the bladder and its key modifiers, and also in motivating further such studies in other tissues.

An important concern, however, is the use of a method of duplex sequencing that has been shown to incur artifacts. Specifically, the development of Nanoseq (Abascal, et al, Nature, 2021, PMID 33911282) and a recent further advancement

(Lawson, et al, biorxiv; <https://doi.org/10.1101/2024.10.30.24316422>) showed that without a process for blocking nicks, library preparation can transform lesions in one strand into mutations. The authors estimate their error rate to be between $1.6 - 4.4 \times 10^{-8}$, compared to Nanoseq's estimate of $< 5 \times 10^{-9}$. In the supplemental notes, the authors detail their development of a computational approach to filter many artifacts, and they make the argument that the level of mutations in bladder is high enough above the artifact level as to not be a concern. However, the experimental work on this was sparse and more should be done to assure that this is indeed the case. Another concern is the extent of validation of the epithelial cell types profiled.

One additional weakness of the study is profiling only known driver genes based on the study by Lawson, et al (PMID 33004514) and bladder cancer genome studies, so they would not have been able to discover entirely novel clonal driver genes with their unprecedented sequencing depth. Their panel could have been expanded further to explore more candidate genes, perhaps sacrificing some sequencing depth to conserve costs.

To summarize, this study provides a very deep interrogation of the most important driver genes in bladder urothelium, exposing processes of mutation and selection there at significantly higher resolution than before, albeit with a method of duplex sequencing that does not provide a completely clean background free of error. This study also makes a good case that this deep duplex sequencing approach could expose a world of in vivo saturation mutagenesis if applied to other tissues. This paper could merit publication in Nature in my view if they address these issues, which are further detailed below.

Introduction and abstract:

1. Describing the details of the approach and results in the introduction are redundant with the results section.

Mutational landscape of the normal bladder urothelium

1. The study used the TwinStrand duplex sequencing kit. Since recent studies (Abascal, et al., Nature 2021) have shown that duplex sequencing can incur significant artifacts that are resolved by nick repair or blocking (as performed by Nanoseq), it is important to validate the potential artifact profile of this version of duplex sequencing. The authors quantified their error rate using blood from a 1 and 12 year old children, which showed error rates between 1.6 to 4.4×10^{-8} that is significantly higher than Nanoseq achieves. The authors state this is negligible as the mutation levels in their tissues are between 2×10^{-7} to 2.5×10^{-8} . But this implies that given the upper range of the error estimate and the lower end of the mutation level estimate, up to 1 in 5 mutations may be an artifact in samples with low mutation levels. To further measure the error profiles, it would be important to profile more early age samples, for example several newborn cord blood (commercially available), perhaps also with the same panel used for the main experiments of this study, and then to perform detailed analyses of the obtained mutation burdens and patterns to prior studies as well as showing no signs of positive selection. I would also strongly suggest the authors perform targeted Nanoseq (Lawson, et al, <https://doi.org/10.1101/2024.10.30.24316422>) on these samples side-by-side to obtain a comparison as possible between the methods. In addition to that, the authors may wish to consider profiling a subset of left-over DNA samples they have left over by targeted Nanoseq to assess concordance with their duplex sequencing method. To the extent that analyses reveal discordant results and patterns, it would then be important to show whether remaining artifacts can be reliably filtered by the computational pipeline. This may also help further improve the computational pipeline.

2. Some genes found to be under positive selection in Lawson, et al (2020 Science) were not included in this study's panel. What was the reason for that?

3. Since post-mortem intervals were several days, epithelial barriers may begin to break down as seen in described for some of the FFPE slides, and it is important to validate that brushings specifically recover epithelial cells. Suppl. Note 1 shows H&E staining of the obtained cells, but it is not clear to me how this validates the identity of these cells as epithelial. Other epithelial cell-specific stains (probably most straightforward), or alternatively single-cell sequencing (most comprehensive but more difficult), should be performed either on prior samples or equivalent new post-mortem samples, since purity of cell types is a key foundation to the study.

4. The study design description should include a summary of how many subjects had cancer/chemotherapy and which types of cancer.

5. Could chemotherapy confound results of study? How do the main conclusions of the study hold up if excluding these subjects?

6. What COSMIC signatures do the three extracted signatures match or consist of? For example, it is unclear what known signatures the SBS-aging signature corresponds to.

7. What is the fractional contribution of each final extracted signature in each individual? For example, it is possible that the APOBEC signature may not be uniform across individuals.

8. Fig. 1B would be more useful if it scaled the tumor (right side) of the figure to the maximum value to see how each gene's prevalence matches between healthy tissues and tumors.

9. The discussion of the chemotherapy results should be more granular-- what class of chemotherapies, for example, since different chemotherapies produce different signatures.

Pervasive positive selection in normal urothelium:

1. The details of the various methods for detecting positive selection could be condensed, with some of the details relegated to the Methods.

Similar landscape of positive selection in dome and trigone:

1. No comments

High interindividual variability in the clonal landscape of normal human urothelium and

Association between the clonal structure of normal urothelium and bladder cancer risk

Factors

1. Are more detailed clinical data available for the subjects beyond sex, smoking/alcohol history, BMI, and chemotherapy exposure available to investigate other possible correlates with the prevalence of specific drivers? While identifying inter-individual variability is important, discovering other associations for these (beyond sex, age, and smoking with TERT mutations) would be potentially clinically impactful.

Evolutionary trajectories of bladder cancer viewed from normal urothelium

1. This section is weaker than the others, as resolving the order by which mutations occur leading up to tumors would be more formally resolved by single-cell sequencing methods that can assess co-occurrence of mutations. Figures 5A and 5B also seem redundant with Fig. 1B. As such, I would suggest this section and its figures be condensed and folded into one of the prior sections.

A pathway to natural human saturation mutagenesis

1. Is it possible for the authors to provide some mathematical framework for how to estimate how many samples would be necessary to reach saturation mutagenesis of one or a few of the genes of interest? This would help guide future studies.
2. The number of driver mutations found in bladder cancers in prior studies is significantly less than found in this study, which causes the comparisons between the tumor and normal tissue mutations across EP300 and TP53 to be sparse on the tumor side and hard to interpret the statistical significance. This comparison could use more formal statistical testing if possible.

Discussion:

1. No comments

Methods:

1. The computational pipeline divides the data into three sets of reads depending on how many duplicates are obtained per strand. While the downstream pipeline seems to analyze these separately, it is not clear how the pipeline analyzes these in a different manner. Also, which set(s) of reads were included in the final analysis? The set of reads with only one copy per strand is likely to contain many more artifacts than the higher quality read sets.
2. Some metric for cross-contamination between samples would be important to implement, since ultra-deep sequencing is sensitive to this. This may be possible using germline variants that differentiate the samples. Some very low level cross-contamination would be acceptable, as long as some threshold is justified for this.

Other figure comments:

1. Fig. 1B: would help to plot it as a stacked barplot for each gene to see what mutation types each is comprised of (missense, frameshift, stop codons, synonymous, etc).
2. Fig. 3A y-axis label should be reworded as it is not clear what it means without reading the legend.

(Remarks on code availability)

Referee #2

(Remarks to the Author)

In this study, Lopez-Bigas and colleagues report an ultradeep sequencing study of normal bladder epithelium. The key unique feature of the study is the application of duplex sequencing on bulk normal tissue at a very high depth (~7000x) compared with previous studies. The caveat is that this is only done on 15 genes + the TERT promoter, which can really only serve as a pilot study to demonstrate the potential of the technique. Overall, the study looks well executed and the methodology is generally sound. While this is the first study of this type, the findings are more on the descriptive side without very strong data to support the basis/mechanism of interindividual variability or why certain genes are more mutated in normal than tumour samples.

Below are some specific comments which should be address:

1. The title is slight inaccurate as the method does not directly allow the study of clonal selection. It is only able to infer the presence of clones, but it is not possible to determine the extent of clonal selection since the contribution of convergent evolution of somatic mutations cannot be differentiated.
2. It should be made clear in the abstract that this is a targeted sequencing study of 15 genes + the TERT promoter.
3. For the comparison of the normal bladder and bladder tumours, the aggregation of samples and description that mutations detected is 1-2 orders of magnitude higher needs to be explained in a clearer context. The comparison should be described on a per sample or "clone" context, including a range of variability between samples.
4. It is a bit confusing across the study that there are two sets of bladder tumour samples, one from Intogen (867 samples) and one from cBioportal (805 samples). It is not really clear why one is used over the other for different analyses and Figures. For example Fig 1B uses the Intogen cohort, but in the methods under "comparison with bladder tumours" only cBioportal samples are described.
5. It would be good to add whether a gene is normal/cancer associated in Extended Data Table 3.
6. In the end, was only non-protein altering mutations were used for de novo mutational signature extraction? This didn't seem to have been clearly described.
7. For estimating the distribution and number of mutations in genes (p.8) it is mentioned that all variants identified in each sample was used. Would it be better to only use non-protein coding mutations for this?

8. Is VAF of protein altering mutations also associated with age? Presumably, the mutations may be acquired in stem cells that replenish the epithelium. If that is the case, VAF of mutations under positive selection may also increase with age a kin to clonal haematopoeisis.
9. On p16 the text describing TERT promoter mutations and smoking does not match the results shown in Figure 4 (should be 4G rather than 4I).
10. The association of TERT promoter mutation with sex. Is it still significant when smoking is accounted for? Again should be Fig 4J rather than 4k).
11. For the associating of sex with other genes, there needs to be p-values reported in both the boxplot (Fig 4K) and the multivariable regression (Figure 4 L). The complete table for the multivariable regression including the significance and contribution of other factors in the male vs female comparison needs to be included in the supplementary tables.
12. On page 20, why does it say 806 bladder cancer samples when the methods says that there are 805 (p. 29). It is also confusing that Fig 5B has bladder tumours of 867 individuals. To make things clearer, the number of samples should be stated in the header in Figure 5A.
13. The statement on P.23 regarding that no genes outside the ones included in their panel as strong candidate drivers of clonal expansion is not really accurate. For example, in REF 9, RHOA and ERCC2 were found to have significant positive selection. Conversely, as the authors showed, they identified TERT mutations where previous studies of normal bladder epithelium didn't, so the profiling of the limited region reported can really only been seen as a pilot study.

(Remarks on code availability)

None of the code are current available for review.

Referee #3

(Remarks to the Author)

The study profiles somatic mutations in normal bladder urothelium using ultradeep duplex sequencing. Compared to an early study (Lawson et al. 2020), more somatic mutations, including mutations in TERT and FGFR3 were revealed. Pervasive positive selection and high variability of clonal expansions between individuals were shown, which have been well established in the field. Truncating mutations in FGFR3 under negative selection is considered novel. The authors also linked risk factors to mutagenesis in the normal tissues, and performed a saturation analysis to mutations not seen in cancers. The paper is well written, and the results are clearly presented. However, there are several issues need to be addressed.

1. Total number of individuals sequenced is only 47, making the study very underpowered to detect any association between mutations and epidemiological risk factors, such as smoking. The authors need to be cautious about their conclusions without replication.
2. TERT promoter mutations appear to be strikingly enriched in smokers in normal tissues in their cohort, but in bladder tumors, TERT mutations are not found to be associated with smoking. The relationship between TERT and smoking seems unclear and should be further validated and investigated (p value for fig 4g is not impressive). If a replication cohort is not possible, limitations in sample size and replication should be clearly acknowledged.
3. The authors claim that "We adapted and developed new computational methods to quantify multiple metrics of positive selection and showed with orthogonal approaches that positive selection is pervasive in the normal urothelium." However, it doesn't seem like there's a new computational method developed in the study. More functional annotation, such as protein structure information is added, but it should not be considered as a new method. Please clarify.
4. Are all individuals selected in the study cancer-free? Why some of them displayed chemotherapy mutational signature and received chemotherapy and radiotherapy? It is unclear if the investigators performed pathological review to ensure that all samples sequenced are indeed normal tissues.
5. Comparison to the Lawson et al 2020 paper should be further investigated and novel findings from this study should be clearly stated. For example, RBM10 and sex bias is reported. Lawson et al reported mutational signature associated with smoking, which is not mentioned in this study.
6. Statistical test is missing in several plots, for example, extended Fig 6c for RB1, where the mutation differences between tumor and normal is interesting. Would be also interesting to include some power simulation in the saturation analysis – with how many samples we will be able to observe all driver mutations for clonal expansion?
7. Fig3A is hard to visualize, with pTERT being the outlier and other genes are barely visible.

(Remarks on code availability)

Authors indicated code will be available upon publication.

Referee #4

(Remarks to the Author)

In this manuscript, Blanco et al. investigate somatic clonal selection in normal urothelium using targeted duplex sequencing in 42 individuals. This study aims to identify factors that contribute to inter-individual variation in clonal selection and expansion. While the research question is intriguing, it remains largely unanswered by the current study, most likely due to the flaws and limitations in the study design, namely the small targeted gene panel and the small cohort size. As a result, the study provides very few new insights into somatic mutagenesis and clonal selection in normal urothelium.

Major points:

1. The cohort size is too small to address the major aim of this study effectively. As I understand, the primary goal of this study is to investigate the factors contributing to inter-individual variation in clonal selection and expansion in normal urothelium. This question inherently requires a large or even epidemiological-scale cohort to be addressed. However, the current study only included a cohort of 42 individuals which is highly insufficient to draw meaningful conclusions about the inter-individual variation in clonal selection. What makes the authors reckon that the current cohort size is adequate to answer this question? The limited sample size appears to prevent the study from providing new insights into the factors influencing clonal selection in normal urothelium beyond what has already been reported. For example, Figure 3b shows the extremely heterogeneous selection/preference of driver mutations in urothelium across individuals. Similar results were also observed by Lawson et al in their study. The fascinating question of which epidemiological factors drive/associate with the inter-individual variation in driver gene selection remains unanswered here, as the study does not provide new evidence or clues. This limitation seems largely attributable to the small cohort size, which undermines the study's ability to achieve its stated objective.
2. Targeted sequencing on 16 genes: this study only did targeted sequencing on 15 genes and TRET promoter, which largely limits its power to uncover novel findings that one would expect. This constrained approach likely explains why limited new findings are reported in the paper. For instance, in the section discussing positive selection in the urothelium, all 13 genes identified under positive selection have already been reported in earlier studies. What rationale do the authors have for believing that focusing on such a limited set of 16 genes would yield new insights? This is particularly also relevant for mutational signature analysis, as the constraints imposed by the targeted panel limit the coverage of trinucleotide sites.
3. Using cytobrushing samples: The authors collected cells from bladder urothelium by brushing large surface areas. This method inevitably introduces contamination from other cell lineages, such as immune and stromal cells residing in the lamina propria. Have the authors evaluated the contamination from other cell lineages using methods beyond HE staining and pathological examination? It is critical to rigorously evaluate contamination through additional approaches, as the reported observations may be confounded by the presence of non-urothelial cell types.
4. Duplex sequencing: As has been raised by Abascal et al in their Nanoseq study previously, duplex sequencing can suffer from errors introduced during end repair, leading to a much higher error rate than the theoretically expected. How did the authors avoid/handle/improve this in their experiments? How accurate is the mutation detection in the study? This requires a thorough evaluation, but the current manuscript provides only limited information, as shown briefly in Extended Data Fig. 1D.
5. It is reasonable to assume that both germline and environmental factors contribute to somatic clonal selection in normal urothelium across individuals. However, germline variables are not considered at all in the current study. To what extent might the observed variation in clonal selection be driven by germline factors?
6. The authors reported a high proportion of shared mutations between dome and trigone, which is unexpected and potentially concerning. What are the VAFs of these shared mutations? How many of them are driver mutations? Can the authors validate these findings using orthogonal approaches? The authors appear surprised by the similar patterns of mutation and clonal selection between the dome and trigone. What reasons led the authors to anticipate differences? Are there any previously reported "geographical" differences in mutation or clonal selection patterns in bladder cancer?
7. The authors reported sporadically observed large clones in the urothelium, such as two mutations in TERT promoter with a VAF of 14% and 20% in one individual. How confident are the authors about this? Have they validated these results using orthogonal approaches?
9. Compared to previous studies, this study provides very limited new insights. Many of the findings have already been reported in earlier research on normal urothelium. For instance, the 13 positively selected genes identified here were previously described, and it is well established that TP53 and FGFR3 are more frequently mutated in bladder cancer than in normal urothelium. Furthermore, all the mutational signatures extracted in this study have been reported before, yet the smoking-related signature SBS92, identified in earlier studies, is notably absent. The lack of novel findings is likely attributable to the study's research strategy and design—specifically, the use of targeted sequencing limited to 16 genes in a small cohort of 42 individuals.

Minor points:

1. TERT is missing in Fig 4k while being mentioned in the main text.
2. I fail to see the point of displaying Figure 4A

(Remarks on code availability)

Version 1:

Reviewer comments:

Referee #1

(Remarks to the Author)

Summary:

The authors have reframed their paper to focus on their novel findings, and they have addressed my comments largely satisfactorily, with some remaining minor comments per below. The mathematical modeling estimating the depth needed to reach saturation mutagenesis I believe is novel and may be a fairly impactful contribution to the field, similar to the early studies that estimated how many samples would be needed to power future GWAS studies.

1. Extended Data Fig. 2b: the similarity should be calculated as a cosine similarity.

2. Error rate estimates: the authors estimate this as the difference between the observed mutation rate in their samples vs the observed mutation rate in NanoSeq. The authors should consider (if feasible computationally for their pipeline) also calculating an error rate per Abascal et al's method using the single-strand calls (i.e. calls observed in only one of the two strands). Specifically, multiplying the frequencies of the reverse complement single-strand calls on a per-trinucleotide context basis, and summing across all trinucleotides.
3. "Some genes found to be under positive selection in Lawson, et al (2020 Science) were not included in this study's panel. What was the reason for that?" -> I recommend adding the authors' explanation for gene choice for the panel to the methods section.
4. Supplementary Note Figure 2.2: The authors should provide more information on the number of such brushes evaluated (ideally several) and would help to have some bar graph of the % for each sample rather than just an average.
5. The methods could use more clarification why rather than only using the medium confidence sets, the high and medium confidence sets of mutations weren't pooled for downstream analyses.
6. Cross contamination analysis: The authors should more directly calculate/estimate what fraction of their final mutation calls could be due to cross-contamination based on their new germline/somatic contamination analysis. There is also a tool called VerifyBAMID2 that they could run similar to Abascal, et al, though it may not work well with the small panel size they have.

General writing comments:

- These sentences are written as a summary of the results, which is somewhat redundant with the beginning of the results section: "Ultradeep sequencing (~5,000x per sample, amounting to ~350,000 haploid genomes in aggregate) in these genomic regions has proved very powerful to accurately quantify selection at the sample level, and to count and characterize the mutations of each gene driving clonal expansions. The normal urothelium of 71 samples obtained from 41 individuals, profiled at this sequencing depth proved key to accurately compute the magnitude of positive selection for each gene at the sample level, and carry out a regression analysis that uncovered this sex bias."

- "game changer": I would rephrase this less formal term

- "We explore this postulate with the data of the normal bladder ultradeep sequencing uncovering an unprecedented power to quantify positive selection at site resolution." -> the grammar and lack of commas is a bit confusing.

- The paragraphs beginning "To explore potential differences in the magnitude of positive selection" and "A correct comparison of the association of sex with the clonal landscape" address sex biases, but then there is a separate section about sex biases. So sex bias is introduced into disparate sections. This structure will be confusing to readers.

- The paragraph beginning "The number of detected clones driven" doesn't appear to belong in the sex bias section, unless I'm confused about its purpose.

- The grammar of this sentence is unclear: "That many somatic tissues..."

(Remarks on code availability)

Referee #2

(Remarks to the Author)

In the current revision, the authors have made substantial effort in improving the robustness of their analyses. The manuscript rigorously demonstrates the technical quality of the ultra-deep targeted sequencing and analysis. As such, I accept that for their purpose of identifying sex bias in selection, their methodology is sufficient and need not be considered as a pilot.

However, I am still not convinced by the significance of the sex bias in selection for mutation in four genes. Firstly, the cohort is still relatively small with just 41 individuals. While the difference is statistically significant in this cohort, it is not certain that this can be generalised to the general population, particularly given most donors appear to be sampled due to incidental health conditions. Second, even if the sex bias is generalisable, without resolving the mechanism, it could just be an association representing some other factor not captured by the study. For example, even though smoking has been controlled, it is well known that males are generally heavier smokers than females (in terms of cigarettes per day or packs/year), could the amount of smoking contribute to the observed bias? Even with the bias shown in non-smokers, the representativeness of the cohort (5 males versus 8 females) remains questionable.

Minor comment – regarding Figure 1b, even though I understand that it is intended for descriptive purposes, the comparison is somewhat unusual, as the number of mutations have different meanings for the two cohorts. If the purpose is to demonstrate the superiority of the technique, a comparison with Lawson et al. might be more informative and appropriate.

(Remarks on code availability)

No code has been made available.

Referee #3

(Remarks to the Author)

I am satisfied with the revision, although it is still puzzling to me why TERT promoter mutations are strongly associated with smoking in normal tissue, but not the case for tumors. I think this paper is a good starting point.

(Remarks on code availability)

Referee #4

(Remarks to the Author)

I thank the authors for their responses. My noteworthy concerns during the first round of reviews were (1) the sample size was insufficient for a comprehensive investigation of inter-individual variation in the mutational landscape and clonal selection in normal urothelium; (2) the limited number of targeted genes constrained the potential for new discoveries; (3) contamination from other cell lineages in cytobrush samples; (4) higher-than-expected error rates of duplex sequencing. In the revision, the authors extensively revamped the manuscript, shifting the primary focus to sex differences in mutation and selection in normal urothelium. Given the higher incidence of bladder cancer in males for largely unknown reasons, it is important for understanding whether mutagenesis and clonal selection differ at the earliest stages of cancer initiation.

I have some follow-up comments and questions:

Regarding concern (1), although I initially expected the authors to expand the cohort in the revision, it is an acceptable move to shift the focus in revised manuscript to sex bias, a question that the current cohort size appears to have enough power to address (as shown in the revision). One of the major findings is that the positive selection on truncating mutations of RBM10, CDKN1A, STAG2 and ARID1A is significantly higher among males. Is the similar sex difference in these four genes seen in bladder cancer? Can authors speculate on how this finding is related to the increased risk of bladder cancer in males? Are there any other mutational parameters showing sex differences, e.g., overall mutational burden, mutational burden of certain mutational signatures, total driver mutation burden, etc?

For the concern (2), the authors claim that the aim of the study is not to identify new genes under positive selection, but rather to understand sex differences in the selection of known genes. This is reasonable, but it still feels like a missed opportunity that the study doesn't leverage the available samples and technology to explore additional findings. For the selection of these 16 genes, beyond sex differences, did authors identify anything that potentially associated with inter-donor variation in selection of mutations in these genes, as obvious inter-donor variation can be seen in Fig 3A regardless of sex?

For the concern (3), the authors provided representative IHC staining images showing the cytobrush sampling predominantly collects urothelial cells with minimal contamination from other lineages. This is a stronger piece of evidence than previously included. How many samples were subjected to IHC staining? Can the authors provide a range of contamination besides representative IHC images.

Regarding concern (4), the authors re-evaluated the error rate of duplex sequencing in this study by sequencing 3 cord blood samples with the same gene panel. They observed the expected average mutation burden for newborn blood, which supports the validity of their approach. However, as shown in Extended Data Fig. 2, the error rate of duplex sequencing remains higher than that of NanoSeq. Have the authors performed NanoSeq on these same cord blood samples for comparison? I understand that this may be challenging, given that the NanoSeq protocol is still in-house.

(Remarks on code availability)

Version 2:

Reviewer comments:

Referee #1

(Remarks to the Author)

The authors have addressed all my comments, and I don't have further ones.

(Remarks on code availability)

Referee #2

(Remarks to the Author)

I have no further comments on the manuscript.

(Remarks on code availability)

Referee #4

(Remarks to the Author)

All of my comments have been addressed satisfactorily. I have no further questions.

(Remarks on code availability)

We are grateful to the four reviewers who revised our paper. Their comments and criticisms have been extremely helpful to improving our manuscript. On the one hand, they motivated us to carry out new analyses aimed at clarifying and extending certain results. On the other hand, they were key to completely restructure and re-write the manuscript, clarifying the main results and streamlining the messages.

Below, we list and briefly explain these new analyses motivated by the reviewers' comments, and their outcome, and we highlight the novel findings of our study.

New analyses

1. Low error rate and high tolerance of the calculation of positive selection.

On the basis of several comments, we recalculated the rate of errors of the DNA duplex sequencing technology employed in our study using cord blood samples from three donors, and with the same gene panel used on bladder samples. The rate of errors is similar to that estimated in the previous version ($\sim 4 \times 10^{-8}$). We also compared the mutational profile and the mutation rate of our technology with that of whole-exome sequencing of laser capture microbiopsies from Lawson et al 2020, which are also normal urothelium obtained during autopsy. This analysis showed remarkable similarities in mutation profile and mutation rate, further indicating that the error rate of our approach is low.

Furthermore, a systematic exploration of the effect of errors (in the range between 10^{-9} and 10^{-7}) on the calculation of dN/dS across our samples revealed a very mild reduction. The metric used to estimate the clonal landscape of the samples is, thus, robust to different types and levels of sequencing errors.

2. The power of high depth per sample

We now explain more clearly the importance of sequencing at high depth per sample to accurately compute selection metrics at sample level. To clarify this we performed a down-sampling study, which demonstrates that sequencing at lower depths would result in a higher rate of missing dN/dS values, effectively reducing the power of the cohort to identify the sex bias.

3. The power of the cohort

We carried out a careful estimation of the power of the cohort to calculate the observed association between the magnitude of positive selection on several genes and sex. This analysis, based on simulations of cohorts of the same size and sex composition and dN/dS distribution features, with variable effect size and baseline value, incontrovertibly demonstrated that the cohort has enough power to identify the observed associations, and further confirms that the high sex bias observed in the four genes is very unlikely to be obtained by chance.

4. Smoking drives the expansion of TERT promoter mutations in normal bladder

We performed new analyses to clarify the association between TERTp mutations with age and smoking. We observed activating TERTp mutations only in older individuals (>55 y.o.)

and almost exclusively in smokers, and conclude that the association between TERTp mutations and smoking is highly significant and robust.

5. High proportion of epithelial cells

As suggested by two reviewers we did an immunohistochemistry study of urothelium brushes obtained and processed exactly as the samples collected for the study. This revealed a very low level of contamination from hematopoietic cells (~3.8%), and no stromal contamination.

6. Application of saturation mutagenesis

On the basis of several comments, we implemented a calculation of the kinetic of saturation mutagenesis, that is, the dependence of the number of genic sites with mutations on the depth of sequencing, under the assumption of neutrality. This kinetic curve revealed the number of haploid genomes that need to be sampled to observe all possible mutations in each gene.

7. Measuring selection at single sites

The large number of driver mutations detected in the study provides the power to compute selection metrics at subgenic resolution, including at the level of individual sites. We implemented a novel calculation of positive selection per individual site or mutation, as well as for sub-genic elements, such as exons or protein domains. The results demonstrate the power of the ultradeep sequencing in normal tissues to identify all functional sites in genes and proteins of interest.

Novel findings of our study

The new streamlined manuscript highlights better the novel findings of our study:

1. The clear sex bias of driver truncating mutations in four genes (RBM10, CDKN1A, STAG2 and ARID1A) in the normal urothelium
2. The discovery that activating TERT promoter mutations appear under strong positive selection in the normal urothelium, and associated with smoking and older age
3. First demonstration of the feasibility of neutral saturation mutagenesis and its applications

REFEREE COMMENTS

OUR RESPONSE

QUOTES FROM THE MANUSCRIPT OR SUPPLEMENTARY NOTES

Referees' comments:

Referee #1 (Remarks to the Author):

Summary:

In this paper, Martinez-Illescas et al. perform targeted duplex sequencing of 16 genes (selected for their known status as driver genes in normal bladder urothelium and bladder cancer) with very high depth in 73 bladder urothelium samples from 42 individuals. This approach allows detection of mutations present in very small clones or even single cells without the laborious work of dissecting individual clones one by one. Thus, they effectively interrogated about half a million cells, many more than the < 1,000 bladder cancer genomes across prior cancer sequencing studies (each cancer interrogates effectively a single clone) and a prior study of bladder mutagenesis by 2,097 microbiopsies (Lawson, et al, PMID 33004514). The authors key findings are: a) identifying > 70,000 mutations that begin to approach saturation mutagenesis of the assayed genes and confirmed the driver gene status via positive selection of 13 of the genes in the panel and negative selection occurring in 1 of the genes in the panel; b) confirmation of the mutational signatures prevalent in bladder urothelium, including an APOBEC signature as well as a signature likely related to the deceased subjects' history of chemotherapy; c) similar patterns of selection in structurally disparate regions of the bladder within individuals, while different patterns of selection between individuals, suggesting that individual specific germline or environmental differences shape selection across the bladder; d) association of more mutations in four genes and in the TERT promoter in males versus females, and association of TERT promoter mutations with smoking history, which may be partly explanatory of the known higher risk of bladder cancer in males; e) identification of some contrasting patterns of mutation in normal bladder urothelium versus bladder cancer that suggests that some healthy tissue mutation/selection evolutionary paths are more prone to cancer formation.

We thank the reviewer for their accurate summarization of our work.

General comments:

Overall, the paper is well written and the computational analysis was fairly rigorous. An important reference study for this paper is Lawson, et al (Science 2020, PMID 33004514). The two studies differ in two key ways: the Lawson, et al study by another group interrogated an effectively much smaller number of genomes than this study (~4,000 haploid genomes from ~2,000 biopsies from 20 individuals vs ~half a million haploid genomes in this study from 42 individuals in this study), but on the other hand, Lawson, et al interrogated many more genes than this study (a mix of ~320 cancer genes, exome-wide, and genome-wide analyses). The prior Lawson, et al study also has some

overlapping findings with this study, including: identification of most of the driver genes active in urothelium which helped this study design the target panel, variation in driver genes across individuals, and APOBEC mutagenesis. This comparison indicates that the key novel findings of the current study are the very high number of mutations identified in the targeted driver genes that paint a more clear picture of mutation and selection in the bladder urothelium, identification of a few novel driver genes in normal urothelium (KMT2C, RB1, FGFR3) as well as hotspot mutations in PIK3CA and the TERT promoter, and also significantly strengthening the prior study's finding of different selection processes across individuals including the effect of sex. I believe these findings will be impactful for understanding how pervasive mutation is in the bladder and its key modifiers, and also in motivating further such studies in other tissues.

It is important to clarify that the sex bias in clonal selection in normal urothelium reported in our manuscript is a novel result not described in Lawson et al 2020 (Science 2020, PMID 33004514) or any other manuscript to the best of our knowledge. To be precise, in the paper by Lawson et al 2020 sex differences are mentioned only in the paragraph quoted below, in a section describing differences in driver preference across individuals, acknowledging the lack of power (9 female and 6 male individuals) in their study to observe sex differences.

“It is unclear whether these differences are driven by variability in environmental exposures or by the genetic background of each individual. No clear evidence of pathogenic germline mutations was found in these genes (22). KDM6A and RBM10 are both located on the X chromosome, and KDM6A is known to escape X-chromosome inactivation, with some evidence suggesting that both KDM6A and RBM10 are more-frequently mutated in males across cancer types (27). However, in our limited cohort, KDM6A appears to be more-frequently mutated in women than men, which is in line with previous observations in non-muscle-invasive bladder cancers (28). Larger cohorts would be required to establish robust associations between epidemiological factors and differences in somatic mutation rates and selection.”

These anecdotal observations are illustrated in this figure (Figure 2G in the Lawson et al., manuscript).

[Redacted figure]

Our study is the first to report a sex bias in the number of truncating mutations across four genes (ARID1A, RBM10, STAG2 and CDKN1A), and this finding is robustly supported by the data. [Redacted text]

The impact of these two elements in the robustness of the results presented in our study are discussed below at length.

Nonetheless, we agree with the reviewer that the novelty of the sex bias in the clonal landscape of the urothelium was not clearly presented in the previous version of our manuscript. To solve this problem, we have extensively re-written it, including changing the title to “Sex differences in selection of somatic mutations in normal human bladder”, and streamlining the Introduction, Results and Discussion to clarify the prior lack of evidence on this bias and that determining its existence is one of the main goals of our study.

An important concern, however, is the use of a method of duplex sequencing that has been shown to incur artifacts. Specifically, the development of Nanoseq (Abascal, et al, Nature, 2021, PMID 33911282) and a recent further advancement (Lawson, et al, biorxiv; <https://doi.org/10.1101/2024.10.30.24316422>) showed that without a process for blocking nicks, library preparation can transform lesions in one strand into mutations. The authors estimate their error rate to be between $1.6 - 4.4 \times 10^{-8}$, compared to Nanoseq’s estimate of $< 5 \times 10^{-9}$. In the supplemental notes, the authors detail their development of a computational approach to filter many artifacts, and they make the argument that the level of mutations in bladder is high enough above the artifact level as to not be a concern. However, the experimental work on this was sparse and more should be done to assure

that this is indeed the case. Another concern is the extent of validation of the epithelial cell types profiled.

One additional weakness of the study is profiling only known driver genes based on the study by Lawson, et al (PMID 33004514) and bladder cancer genome studies, so they would not have been able to discover entirely novel clonal driver genes with their unprecedented sequencing depth. Their panel could have been expanded further to explore more candidate genes, perhaps sacrificing some sequencing depth to conserve costs.

To summarize, this study provides a very deep interrogation of the most important driver genes in bladder urothelium, exposing processes of mutation and selection there at significantly higher resolution than before, albeit with a method of duplex sequencing that does not provide a completely clean background free of error. This study also makes a good case that this deep duplex sequencing approach could expose a world of in vivo saturation mutagenesis if applied to other tissues. This paper could merit publication in Nature in my view if they address these issues, which are further detailed below.

We thank the reviewer for presenting these general comments on our study. We have now clarified all of them in the new version of the manuscript and, as a result, we are confident that the results are more clearly presented and all potential caveats have been satisfactorily addressed.

Here we summarize the steps we have taken to address each of these concerns, which are detailed below in response to each comment by the reviewer.

1. Rate of errors of the technology

We recalculated the rate of errors of the DNA duplex sequencing technology employed in our study using cord blood samples from three donors, and with the same genic panel used on bladder samples. The rate of errors is similar to that estimated in the previous version ($\sim 4 \times 10^{-8}$). We also compared the mutational profile and the mutation rate of our technology with that of whole-exome sequencing of laser capture microbiopsies from Lawson et al 2020, which are also normal urothelium obtained during autopsy. This analysis showed remarkable similarities in mutation profile and mutation rate, further indicating that the error rate of our approach is low. Furthermore, a systematic exploration of the effect of errors (in the range between 10^{-9} and 10^{-7}) on the calculation of dN/dS across our samples revealed a very mild reduction. The metric used to estimate the clonal landscape of the samples is, thus, robust to different types and levels of sequencing errors.

2. Fraction of epithelial cells

An immunohistochemistry study of urothelium brushes obtained and processed exactly as the samples collected for the study revealed a very low level of contamination from hematopoietic cells ($\sim 3.8\%$), and no stromal contamination.

3. Processing only known bladder drivers

This study did not aim to discover new genes under positive selection in the normal urothelium. Its two main aims (clarified in the new version of the manuscript) were the study of a potential sex bias in the clonal landscape of the normal urothelium and the exploration of natural saturation mutagenesis. Both aims required ultradeep sequencing (we reached ~5,000x in individual samples) to accurately and thoroughly identify driver mutations across samples. Focusing on the genes known to be under positive selection in normal bladder or bladder tumor allowed us to achieve this depth of sequencing, advancing both objectives. Through a down-sampling study, we demonstrated that sequencing at lower depths would result in a higher rate of missing dN/dS values, effectively reducing the power of the cohort to identify the sex bias.

Introduction and abstract:

1. Describing the details of the approach and results in the introduction are redundant with the results section.

We agree with this comment. The Introduction of the manuscript has been streamlined to focus on the goals of the study, while a description of the approach (also abridged) is presented in Results.

Mutational landscape of the normal bladder urothelium

1. The study used the TwinStrand duplex sequencing kit. Since recent studies (Abascal, et al., Nature 2021) have shown that duplex sequencing can incur significant artifacts that are resolved by nick repair or blocking (as performed by Nanoseq), it is important to validate the potential artifact profile of this version of duplex sequencing. The authors quantified their error rate using blood from a 1 and 12 year old children, which showed error rates between 1.6 to 4.4 x 10⁻⁸ that is significantly higher than Nanoseq achieves.

The authors state this is negligible as the mutation levels in their tissues are between 2x10⁻⁷ to 2.5x10⁻⁸. But this implies that given the upper range of the error estimate and the lower end of the mutation level estimate, up to 1 in 5 mutations may be an artifact in samples with low mutation levels. To further measure the error profiles, it would be important to profile more early age samples, for example several newborn cord blood (commercially available), perhaps also with the same panel used for the main experiments of this study, and then to perform detailed analyses of the obtained mutation burdens and patterns to prior studies as well as showing no signs of positive selection. I would also strongly suggest the authors perform targeted Nanoseq (Lawson, et al, <https://doi.org/10.1101/2024.10.30.24316422>) on these samples side-by-side to obtain as direct of a comparison as possible between the methods. In addition to that, the authors may wish to consider profiling a subset of left-over DNA samples they have left over by targeted Nanoseq to assess concordance with their duplex sequencing method. To the extent that analyses reveal discordant results and patterns, it would then be important to show whether remaining artifacts can be reliably filtered by the computational pipeline. This may also help further improve the computational pipeline.

Following the reviewer’s suggestion, we re-calculated the error rate of the DNA duplex sequencing technology employed in our study directly on three cord blood samples with the same panel used in this study. The results are very similar to those previously obtained on pediatric blood samples on the previous version of the manuscript. In the new version of the manuscript, the result of these experiments are presented in Extended Data Figure 2 (copied below), and discussed in the Supplementary Note 4 (quoted below).

Extended Data Figure 2a

Extended Data Figure 2. Error rate of the DNA duplex sequencing technology

a) Left panel, orange bars, mutation rate detected by the DNA duplex sequencing technology in this study using the same panel of genes in cord blood samples from three donors. The comparison of this observed mutation rate with that expected in these samples yields an estimate of the error rate of the technology, as $\sim 4 \times 10^{-8}$ per sequenced base pair. Blue bars, mutation rate detected by other similar technology (NanoSeq; ref; data taken from ref) in two cord blood samples (Supplementary Note 4). Right panel, comparison of the estimated error rate by both technologies with the mutation rate detected across the 71 normal urothelial samples included in this study (rightmost red bar). It shows that the rate of errors of the DNA duplex sequencing technology used in the study is approximately 25 times smaller than the mutation rate detected in the normal urothelium.

Supplementary Note 4

Estimation of the error rate of the technology in cord blood samples

We estimated the error rate intrinsic to the duplex sequencing technology employed to detect mutations in this study. To that end, we constructed DNA duplex libraries using the same kits and the same capture panel as in this study for three cord blood samples obtained from newborns (StemCell; Extended Data Fig. 3x). The samples were sequenced to depths of 2985x, 4608x and 4308x. We identified 35, 26 and 47 SNVs in these three samples, accounting for mutation rates of 6.1×10^{-8} , 3.1×10^{-8} and 5.7×10^{-8} mutations per sequenced base (Extended Data Fig. 2a). We obtained the average number of mutations

observed in newborn blood from two studies^{18,19}, based on the expansion of clones from hematopoietic stem cells. According to these two studies, the number of somatic SNVs in cord blood is between 60 and 105, yielding an expected mutation rate between 2×10^{-8} and 3.5×10^{-8} .

Comparing the rate of mutations detected in these three samples with that expected in newborn blood, we estimated that the rate of errors introduced by the technology using these three samples is not greater than 4.1×10^{-8} . This is the value shown in Extended Data Figure 2a.

We also compared the mutational profile and the mutation rate of our technology with that of whole-exome sequencing of laser capture microbiopsies from Lawson et al 2020, which are also normal urothelium obtained during autopsy. This analysis showed remarkable similarities in mutation profile and mutation rate (Extended Data Fig. 2b,c, copied below; Supplementary Note 4, quoted below), indicating that the error rate of our approach is low.

Extended Data Figure 2b,c

b) Mutational profile of normal urothelium obtained through two orthogonal approaches. Top panel, profile constructed using mutations detected through laser capture microdissection of clonal or quasi-clonal samples followed by regular shallow whole-genome sequencing

(data taken from ref). Bottom panel, profile constructed using mutations detected in this study from $\sim 2\text{cm}^2$ brushes followed by ultradeep DNA duplex sequencing.

c) Relationship between the mutation rate calculated in normal bladder urothelium using two orthogonal approaches. The red dots correspond to the rate of mutations detected through laser capture microdissection of clonal or quasi-clonal samples followed by regular shallow whole-exome sequencing (LCM+WES; data taken from ref). The blue dots correspond to the rate of mutations computed for the 71 samples in this study from $\sim 2\text{cm}^2$ brushes followed by ultradeep DNA duplex sequencing. The trend line represented in the plot was calculated from the LCM+WES samples.

Supplementary Note 4

Comparison of ultradeep sequencing with other technology

To have an orthogonal comparison of the error rate we can compare the mutation pattern and mutation rate of our study with that of normal bladder urothelium from autopsies obtained through laser capture microdissection of clonal or quasi-clonal structures (ref). In this case the sequencing approach is bulk whole-genome sequencing, which allows the comparison of duplex and bulk genome sequencing in the same tissue and type of samples.

This comparison revealed strikingly similar mutational profiles (Extended Data Fig. 2b), suggesting the absence of a large number of artifacts with a different profile across the mutations identified in the duplex sequencing of brushes of normal urothelium. In addition, the mutation rate of the normal urothelium in our study revealed a very close match with the values obtained in the laser microdissection study (Extended Data Fig. 2c), again pointing to a low proportion of errors in our study. Moreover, the distribution of mutation rate across both cohorts is very similar, as is the linear relationship between mutation rate and age, which would be incompatible with high error rates across the samples in our cohort.

Based on the reviewer's comment, we also reviewed the potential error rate across all samples in the cohort. We visually compared the mutational profile of samples with low and high error rate, as well as the mutational profile of all samples in this cohort probed through ultradeep sequencing with the normal urothelium samples obtained by Lawson et al. (2020) using laser capture microdissection and sequenced at the whole-genome level. Moreover, we compared the distribution of mutation rate calculated for normal urothelium samples using these two orthogonal technologies. All these analyses (described in Supplementary Note 4, quoted below) suggest that the level of error across samples is not sufficient to alter neither their mutational profile, nor their mutation rate.

Supplementary Note 4

Mutational profile of samples with different mutation rate

This comparison is based on the error rate calculated in cord blood, a scenario in which we do not expect positive selection, in contrast to what we observe in the normal urothelium of

adults. In addition, the type of sampling is different in both scenarios. Thus, to understand whether this rate of errors posed a problem in its comparison with different mutation rates across samples, we inspected the mutational profile of the two samples with the lowest mutation rate (01_DO and 27_TR), and of two among those with the highest mutation rate, with the mutational profile of the pooled cohort. We reasoned that a higher prevalence of artifacts in the samples with lower mutation rate should be reflected through the observation of a more different mutational profile. However, the comparison reveals that the profile of these samples is very similar to that of the pooled cohort (cosine similarity = 0.83 and 0.74, respectively; Supplementary Note Fig. 4.1).

Supplementary Note Figure 4.1. Mutational profile of the pooled cohort (top) and the two samples with the lowest (left) and highest (right) mutation rate. Cosine similarity with the mutational profile of the cohort is shown for each sample.

Moreover, we expect that the presence of artifactual mutations across samples in the cohort would result in the extraction of a mutational signature representing them. As discussed in Supplementary Note 5, this is not the case.

Furthermore, we systematically explored the effect of an error rate comparable to or higher than that estimated in cord blood on the estimation of dN/dS values across genes (Extended Data Fig. 6a-c and Supplementary Note 8, copied below). We found that error rates similar to or higher than those estimated in cord blood cause only a mild reduction in the values of dN/dS computed across genes. Moreover, it is not possible that these dN/dS values are spuriously generated by sequencing errors.

Extended Data Figure 6a-c

Extended Data Figure 6. Tolerance of dN/dS values to errors

a) Measurement of the tolerance of RBM10 dN/dS truncating to artifactual mutations (artifacts) following the BotSeq mutational profile (ref.). The boxplots represent the distribution of dN/dS values calculated from 100 synthetic samples with increasing rates of injected artifacts between 0 and 1×10^{-7} (one order of magnitude higher than estimated for the technology), and for increasing values of ground truth dN/dS (between 1 and 50).

b) Average percentage of RBM10 dN/dS truncating reconstructed value across 100 synthetic samples, calculated by computing which fraction of the ground truth dN/dS in a sample is obtained upon calculation.

c) Summary of the results of the experiment of error tolerance. Left panel, average percentage of reconstructed ground truth dN/dS across synthetic samples (for all genes and all ground truth dN/dS explored altogether) that is calculated upon injection of increasing rates (x-axis) of different types of artifacts (color legend). Center plot, average percentage of reconstructed ground truth dN/dS across synthetic samples (for all genes and all artifacts altogether) that is calculated upon injection of increasing rates (x-axis) for different values of ground truth dN/dS (color legend). Right panel, average percentage of reconstructed ground truth dN/dS across synthetic samples (for all ground truth dN/dS explored and all artifacts altogether) for different genes (color legend), that is calculated upon injection of increasing rates (x-axis) of artifacts.

Supplementary Note 8

Tolerance of positive selection calculations to errors

We wondered how artefacts detected in the samples (as they occur randomly across genomic positions) impact the magnitude of positive selection estimated for different genes. We wanted to assess how the dN/dS estimates calculated with Omega across samples could be impacted by artifactual mutations (herein 'errors') generated with the error rates estimated from cord blood.

We simulated synthetic catalogs of mutations (i.e., synthetic samples) whereby two signals are combined: ground truth mutations (obtained through sampling of the mutational profile of a real bladder sample) distributed across synonymous and non-synonymous sites to obtain a pre-defined dN/dS value, and error mutations. The baseline synthetic samples only contained "true" mutations sampled from the mutational profile of the cohort. In different simulations, we then injected increasing levels of artifacts sampled from mutational profiles representing different sources of errors.

Artifacts' mutation probability vectors

We tested the tolerance of Omega to known frequent sequencing artifacts (SBS43, SBS45, SBS52, SBS58 in the Cosmic reference catalog of mutational signatures; ref), and to the BotSeq mutational profile in cord blood reported in Abascal et al. (Nature 2021), which is enriched in artifactual C>A mutations common to this technology (artifacts hereafter) (Supplementary Note Fig. 8.1).

Results of the experiment

We calculated the dN/dS of each of the genes probed in this study across a set of synthetic samples with a given ground truth dN/dS (between 1 and 50), and a rate of errors (between 10^{-9} and 10^{-7} per base pair sequenced) injected on top of the true mutations. This calculation was replicated across 100 synthetic samples. As an example, Supplementary Note Figure 8.2a shows the reconstructed dN/dS truncating for RBM10 after receiving injections of increasing artifactual mutations generated in accordance with the BotSeq C>A mutational profile (ref; Supplementary Note Fig. 8.1). We observed that increasing rates of errors decrease the estimated dN/dS. However, even in the most extreme case of injecting one artifactual mutation every 107 base pairs sequenced, Omega is capable of reconstructing more than 80% of the ground truth dN/dS (Extended Data Fig. 6a,b).

Extended Data Figure 6c shows the overall reconstruction of dN/dS across 5 artifact sources (see Supplementary Note Fig. 8.1), different ground truth values of dN/dS and genes. As expected, artifactual mutations do not have an effect in the dN/dS estimates when the ground truth value is 1 (i.e. no selection). With increasing values of ground truth dN/dS, the reduction in dN/dS increases, although it is comparable across scenarios of high selection (i.e. ground truth dN/dS of 5, 10 and 50). Despite minor differences, the trend of underestimation of dN/dS across artifact sources and genes is maintained.

2. Some genes found to be under positive selection in Lawson, et al (2020 Science) were not included in this study's panel. What was the reason for that?

Lawson et al sequenced 1,674 bladder microbiopsies from 20 individuals and identified 17 genes under positive selection but only 10 had more than 10 mutations across all microbiopsies. Based on these results, we designed a panel for DNA duplex sequencing including these 10 genes plus 4 more that are frequently mutated in bladder tumors (data from COSMIC), and the TERT promoter, also known to be under positive selection in bladder carcinomas (see, for example, <https://www.nature.com/articles/s41586-020-1965-x>).

3. Since post-mortem intervals were several days, epithelial barriers may begin to break down as seen in described for some of the FFPE slides, and it is important to validate that brushings specifically recover epithelial cells. Suppl. Note 1 shows H&E staining of the obtained cells, but it is not clear to me how this validates the identity of these cells as epithelial. Other epithelial cell-specific stains (probably most straightforward), or alternatively single-cell sequencing (most comprehensive but more difficult), should be performed either on prior samples or equivalent new post-mortem samples, since purity of cell types is a key foundation to the study.

To assess the fraction of epithelial cells in the brushes of the urothelium obtained at autopsy, we used immunohistochemistry staining with the following antibodies for specific cell types: cytokeratin 7 (CK7) for epithelial cells, uroplakin II for urothelial cells (specifically the upper layer of urothelial cells called umbrella cells), CD45 for lymphocytes, and actin for muscle cells (Supplementary Note 2). We observed that the urothelial brushes contained predominantly normal epithelial cells, as demonstrated by almost uniformly positive CK7 staining. Umbrella cells were also observed, as expected due to the superficial sampling of the tissue. The average percentage of CD45 positive lymphocytes was low (3.8%) and actin staining was negative indicating minimal contamination with blood cells and no contamination with stroma. Overall, these results confirm that the sampling of urothelium with brushes mostly collects epithelial cells, thus validating this approach for the study of clonal expansions in bladder urothelium.

Supplementary Note Figure 2.2

Supplementary Note Figure 2.2. Determination of cell composition of urothelial brushes by immunohistochemistry. A representative urothelial brush from the bladder dome was fixed in Cytolyt, spun down, and embedded in paraffin to make a cell block. Slides sectioned from the cell block were stained with hematoxylin-eosin (H&E) to determine cell morphology, and immunostained with the following antibodies for specific cell types: cytokeratin 7 (CK7) for epithelial cells, uroplakin II specific for upper layer of urothelial cells (strongest in umbrella cells), CD45 for lymphocytes (scattered cells highlighted by red arrows) and actin for muscle cells. Representative pictures at 20x and 40x are shown.

4. The study design description should include a summary of how many subjects had cancer/chemotherapy and which types of cancer.

This information has now been included in Extended Data Table 1.

5. Could chemotherapy confound results of study? How do the main conclusions of the study hold up if excluding these subjects?

We thank the reviewer for this suggestion. Note that the exposure to chemotherapy is included as a covariable in all regressions. Nevertheless, following this comment, we repeated the regression of sex on dN/dS values excluding the individuals who received chemotherapy. We found that the same four genes, RBM10, STAG2, ARID1A and CDKN1A, appeared significant. The results of this analysis are shown in Supplementary Note 9, quoted below.

Supplementary Note 9

Exclusion of samples exposed to chemotherapy

We also checked whether the samples belonging to subjects previously exposed to chemotherapy could be biasing the sex association with dN/dS, although this variable was taken into account in the multivariate analysis when significant in the univariate analysis. In the regression calculated in the resulting smaller cohort, we obtained a significant association between the dNdS and sex for ARID1A, CDKN1A and RBM10. STAG2, although with a similar effect size did not appear significant in this smaller cohort, probably due to a reduced power with a smaller dataset (Supplementary Note Fig. 9.8).

Supplementary Note Figure 9.8. Effect of removing samples exposed to chemotherapy or with low mutation rate on the association between sex and dN/dS.

Scatter plots showing the effect size (x-axis) and false discovery rate (FDR, y-axis, in logarithmic form) of the multivariate association between dN/dS and sex when excluding subjects with a chemotherapy history in the genes where we report a sex association. FDR thresholds of 0.1 and 0.2 are depicted in the figure as grey dashed lines, as well as an effect size of 0.

6. What COSMIC signatures do the three extracted signatures match or consist of? For example, it is unclear what known signatures the SBS-aging signature corresponds to.

Following the reviewer’s comment, we have now expanded the description of the exploration of the etiology of the three mutational signatures extracted from the cohort of normal urothelium samples. This is described in Supplementary Note 5, quoted below.

Supplementary Note 4

Mutational signatures identified in the cohort

We carried out a de novo mutational signature extraction from the mutational profiles observed across samples in the cohort. To this end, we used two different methods: a Bayesian hierarchical Dirichlet process (HDP)^{23,24} and a nonnegative matrix factorization-based algorithm (SigProfiler)^{25–28}. While the extraction using SigProfiler yielded three signatures (SBS96A-C; Extended Data Fig. 2), we obtained six signatures using HDP (N1-N6; Extended Data Fig. 4). HDP signatures N1-N3 showed very similar profiles to SBS96A-C (cosine similarity = 0.99, 0.99, 0.98, Supplementary Note Fig. 5.2), with the remaining N4-N6 contributing showing very little activity across the cohort (less than 7% each), with the exception of a few samples (Extended Data Fig. 4).

Supplementary Note Figure 5.2. Comparison of de novo signatures extracted by two independent methods. Values in the heatmap represent the cosine similarity of each pair of signatures.

Decomposition of extracted signatures on the reference catalog

Formally, SBS96A (N1, according to HDP) can be represented as a linear combination of COSMIC signatures SBS2 (23.7%), SBS13 (19.7%) and SBS40a (56.5%) with a cosine similarity 0.93 (Supplementary Note Fig. 5.4, left). Therefore, we named it SBS-APOBEC (as the etiology of SBS40a is unknown). APOBEC mutagenesis and its associated signatures have already been described in normal bladder urothelium²², where it was shown to be specific to some clones, as well as in bladder tumors²⁷. In our data it was found across almost all samples, probably due to the wide brushing collection that leads to the combined analysis of a mixture of clones (see main manuscript), although its contribution varies between individuals (Extended Data Fig. 3).

The second most active signature in the cohort, SBS96B (N2), is very similar to SBS5 (cosine similarity=0.91). Formally, SBS96B can be represented as a linear combination of COSMIC signatures SBS5 (92%) and SBS97 (8%) with a cosine similarity 0.94 (Supplementary Note Fig. 5.4, center). Since SBS5 is known to be clock-like²⁹, we checked the association of the activity of SBS96B (N2) with the age of donors. This analysis showed that its activity (measured in absolute number of mutations) strongly correlates with age (Fig. 1d in the main manuscript). We thus refer to SBS96B (N2) as SBS-aging.

The third signature, SBS96C (N3) has not been previously described in normal bladder. Attempts to decompose it on the COSMIC catalog give a preponderance to SBS3, a flat signature associated with homologous recombination deficiency, which is absent from this cohort. We found no other meaningful decomposition of SBS96C (Supp. Note Fig. 5.3, right). However, we noticed a strong association of this signature with prior exposure to chemotherapy (Fig. 1c of the main manuscript). It is very difficult to associate this signature with a particular chemotherapy, as all exposed donors (11) received combinations of multiple agents (Extended Data Table 1).

Supplementary Note Figure 5.3. Decomposition of de novo extracted signatures in linear combination of cosmic signatures using deconstructSigs. The x-axis represents the signatures allowed for the reconstruction of the mutational profile of the cohort in each

case. The top plots show the cosine similarity between the observed and reconstructed profiles. The bottom plots show the activity of each cosmic signature in the decomposition.

7. What is the fractional contribution of each final extracted signature in each individual? For example, it is possible that the APOBEC signature may not be uniform across individuals.

The activity across individuals of all mutational signatures identified in the cohort is presented in Extended Data Figure 3a-d, copied below.

We observed activity of APOBEC (SBS-APOBEC) across all samples in the cohort, probably due to the fact that the study is based on a brushing of the urothelium that produces a mixture of thousands of clones.

Extended Data Figure 3

Extended Data Figure 3. Mutational signatures active across the cohort

- Mutational profile of the signatures identified using HDP.
- Activity of the signatures identified using HDP across the 71 samples.
- Mutational profile of the signatures identified using SigProfiler.
- Activity of the signatures identified using SigProfiler across the 71 samples.

For more details on the identification of these mutational signatures and their decipherment of their etiology, see Supplementary Note 5.

N1, N2 and N3, roughly correspond to SBS96A (SBS-APOBEC), SBS96B (SBS-aging) and SBS96C (SBS-chemo).

8. Fig. 1B would be more useful if it scaled the tumor (right side) of the figure to the maximum value to see how each gene's prevalence matches between healthy tissues and tumors.

We agree with the reviewer that showing how the prevalence of mutations between healthy tissues and tumors matches or differs is interesting. We show this in Extended Data Figure 9a, copied below. The goal of Fig. 1b is to show the different scale in the number of mutations identified in normal urothelium across our cohort and in bladder tumors in over a decade of cancer genomics. This is the reason why we present it with comparable scales on both sides.

Extended Data Figure 9a

Extended Data Figure 9. Calculations of natural selection mutagenesis

a) Comparison of the rate of protein-affecting mutations in 14 genes across two cohorts of bladder tumors (muscle invasive and non-muscle invasive) and in the normal urothelium of the 41 individuals.

9. The discussion of the chemotherapy results should be more granular-- what class of chemotherapies, for example, since different chemotherapies produce different signatures.

We agree with the reviewer that an analysis separated by type of chemotherapeutic agent could provide more insight into the mutational footprint of each of them. Nevertheless, only 12 individuals in the cohort received chemotherapy, with very different regimens and combinations (Extended Data Table 1). Therefore, associating SBSC, which we find active across individuals who received chemotherapy/radiotherapy to the exposure to a given drug is not possible in this cohort. We now explain this in the Supplementary Note 5 devoted to the mutational signature analysis.

Pervasive positive selection in normal urothelium:

1. The details of the various methods for detecting positive selection could be condensed, with some of the details relegated to the Methods.

We agree with the reviewer. The second section of the Results has been completely re-written. The details of all methods used to compute positive selection are now included in Methods and Supplementary Note 6.

Similar landscape of positive selection in dome and trigone:

1. No comments

High interindividual variability in the clonal landscape of normal human urothelium
and

Association between the clonal structure of normal urothelium and bladder cancer risk
Factors

1. Are more detailed clinical data available for the subjects beyond sex, smoking/alcohol history, BMI, and chemotherapy exposure available to investigate other possible correlates with the prevalence of specific drivers? While identifying inter-individual variability is important, discovering other associations for these (beyond sex, age, and smoking with TERT mutations) would be potentially clinically impactful.

We agree with the reviewer that detecting such associations would be extremely interesting. The lack of more clinical data for these individuals, and the size of the cohort prevents us to do that. We decided to focus our analysis on sex, as this is the most significant risk factor of bladder cancer (as we show in Supplementary Note 1), and we also show results of association between smoking and TERT promoter mutations. Nevertheless, as stated in the Discussion section,

We envision that the demonstration of the usefulness of ultradeep sequencing to investigate the role of sex and smoking in shaping the clonal landscape of normal urothelium and to pursue natural saturation mutagenesis will set the stage for future studies on these and other cancer risk factors across other human tissues, involving larger cohorts of donors. Scaling up this analysis requires the possibility to obtain samples from different tissues in a minimally invasive manner. In the case of bladder, this could conceivably be achieved through urine or lavages.

Evolutionary trajectories of bladder cancer viewed from normal urothelium

1. This section is weaker than the others, as resolving the order by which mutations occur leading up to tumors would be more formally resolved by single-cell sequencing methods that can assess co-occurrence of mutations. Figures 5A and 5B also seem redundant with Fig. 1B. As such, I would suggest this section and its figures be condensed and folded into one of the prior sections.

We completely agree with the reviewer's assessment. This section has been removed in the new version of the manuscript, and its contents largely reduced and merged within the

positive selection section and the Discussion. While the aim of Figure 1b is showing the difference in scale of the number of mutations identified in the normal urothelium across individuals in this cohort, the former Figure 5a (Extended Data Fig. 9a in the current version) presents the relative mutation rate observed for different genes across normal and tumor samples, which cannot be appreciated in Figure 1b. Former Figure 5b (Figure 5a in the current version) illustrates the fraction of sites in each protein with one or more mutations identified across normal and tumor samples (an information which cannot be inferred from the other two figures). The aim of the latter figure is to provide a first approximation to the question of how close we are to reaching saturation mutagenesis of these genes in tumor and normal samples.

To facilitate their comparison, we copy below Figures 1b and 5a and Extended Data Figure 9a.

Figure 1b

Figure 1. Ultradeep DNA sequencing of normal urothelium targeting driver genes reveals thousands of mutations.

b) Number of SNVs detected in a panel of selected genes in the normal bladder in this study and comparison with the number of mutations detected in 892 tumors from bladder cancer genomics studies, obtained from intOGen.

Figure 5a

Figure 5. Natural saturation mutagenesis

a) Percentage of amino acid residues in each gene with zero, one, two, or three or more mutations observed across the 71 samples.

Extended Data Figure 9a

Extended Data Figure 9. Calculations of natural selection mutagenesis

a) Comparison of the rate of protein-affecting mutations in 14 genes across two cohorts of bladder tumors (muscle invasive and non-muscle invasive) and in the normal urothelium of the 41 individuals.

A pathway to natural human saturation mutagenesis

1. Is it possible for the authors to provide some mathematical framework for how to estimate how many samples would be necessary to reach saturation mutagenesis of one or a few of the genes of interest? This would help guide future studies.

This is a very interesting suggestion. Following this comment, we have carried out a new analysis, which is now described in the last section of Results (quoted below) and illustrated in Figure 5b (copied below). The result of this analysis is that, depending on the gene, the sequencing of between 10^7 - 10^8 haploid genomes are required for complete saturation. In this study, we have reached $\sim 3.5 \times 10^5$.

Figure 5b

Figure 5. Natural saturation mutagenesis

b) Theoretical and observed curves of saturation mutagenesis for TP53 and FGFR3. The black dashed line represents the kinetic of saturation mutagenesis under the theoretical assumption of no selection, in which mutations are observed based only on their neutral probability of occurrence. The red circle denotes the degree of saturation achieved by probing the 71 samples in the cohort. The red dashed line is constructed through

successive down-samples of the current observation and represents the observed kinetic of natural saturation mutagenesis.

Results extract

Next, we asked how many more genomes would need to be sequenced to observe mutations on all amino acid residues of each of these genes. To estimate this, we first calculated the probability to observe each mutation in the gene under neutrality –due to different mutational processes. Thus, we obtained a theoretical curve describing the fraction of all possible mutations that are expected to be observed at different sequencing depths (theoretical kinetic of natural saturation mutagenesis; Fig. 5b continuous curve; Extended Data Fig. 9c; Supplementary Note 11). The shape of this curve is determined by the difference in probability across mutations, governed by their trinucleotide contexts, with some mutations much more likely to occur than others. For example, for TP53, with the accumulated depth provided by the entire cohort (~350.000x), only on the basis of neutral mutagenesis, we expect to have observed less than half of all possible mutations. In theory, being able to detect every possible mutation in the gene would require an accumulated depth of $\sim 10^7$.

Nevertheless, the observation of mutations in normal urothelium is not neutral, but instead strongly affected by selection. To obtain the actual curve of the fraction of observed mutations depending on the number of sequenced genomes (observed kinetic of natural saturation mutagenesis), we subsampled the number of mutations observed in each gene across the cohort respecting their variant allele frequency, but reducing the depth of sequencing at each position (Fig. 5b circle and dots; Extended Data Fig. 9c). The divergence in the shape of the two curves reveals the importance of selection in shaping the probability to observe a given fraction of mutations in a gene. For example, since missense and truncating TP53 mutations are under positive selection, we have actually observed more mutations than predicted by the theoretical kinetic. For several genes, we observed this faster accumulation of mutations than expected under neutral mutagenesis due to positive selection. Interestingly, the curve for FGFR3 falls below the theoretical one, constructed under purely neutral mutagenesis (Fig. 5c), which supports the notion that FGFR3 mutations are under negative selection in the normal urothelium.

2. The number of driver mutations found in bladder cancers in prior studies is significantly less than found in this study, which causes the comparisons between the tumor and normal tissue mutations across EP300 and TP53 to be sparse on the tumor side and hard to interpret the statistical significance. This comparison could use more formal statistical testing if possible.

The reviewer is correct that there is an important difference in scale in the number of mutations identified in these genes across tumor and normal (in this study) samples, as shown in Figure 1b. Moreover, this actual degree of difference varies between genes and depending on the type of bladder carcinoma (Extended Data Fig. 9a). These figures (and Fig. 5a) illustrate the importance of recurring to natural saturation mutagenesis. The aim of

former Figures 5c and 5d (Extended Data Fig. 9d,e in the current version) are to visually illustrate the differences in the distribution of mutations in two exemplary genes across normal and tumor samples, as well as to provide evidence of the power of natural saturation mutagenesis.

Extended Data Figure 9d, e

Extended Data Figure 9. Calculations of natural selection mutagenesis

d) Natural saturation mutagenesis of EP300 in normal bladder urothelium. Besides the tracks described in Figure 5c for TP53, 3D clusters obtained via Oncodrive3D (second), dN/dS truncating and dN/dS missense values for each exon (fourth), and the distribution of mutations along the sequence of the gene (last) have been added. The right hand plots are the same described in Figure 5c.

e) Same as d, for RBM10.

Discussion:

1. No comments

Methods:

1. The computational pipeline divides the data into three sets of reads depending on how many duplicates are obtained per strand. While the downstream pipeline seems to analyze these separately, it is not clear how the pipeline analyzes these in a different manner. Also, which set(s) of reads were included in the final analysis? The set of reads with only one copy per strand is likely to contain many more artifacts than the higher quality read sets.

As the reviewer states, three sets of duplex reads are obtained by the pipeline on the basis of the number of duplicates per strand. A high confidence set includes only mutations supported by duplex reads with at least three duplicates per strand; a mid confidence set includes only mutations supported by duplex reads with at least two duplicates per strand; while a low confidence set, includes mutations identified by duplex reads supported by one duplicate per strand. While the pipeline produces these three sets of mutations, which we analyze and compare for diagnostics purposes, we only use for downstream analysis reads in the mid confidence set. We have now clarified this point in Supplementary Note 3 (quoted below). To substantiate our selection, we present, as part of this Supplementary Note, the mutational profile of the mutations obtained from the three sets of reads. Their comparison shows that while the profiles of the high and medium sets are very similar, the profile of low is notably different and probably enriched for artifacts. Since high and medium appear to yield results of similar quality, we have decided to use the medium set, as it comprises more mutations.

2. Some metric for cross-contamination between samples would be important to implement, since ultra-deep sequencing is sensitive to this. This may be possible using germline variants that differentiate the samples. Some very low level cross-contamination would be acceptable, as long as some threshold is justified for this.

Following this comment we have now checked SNPs in the genes under study, and verified that there is no degree of discernible contamination between individuals. This is presented in Supplementary Note 3, quoted below.

Supplementary Note 3

Cross-contamination analysis

We explored the level of potential contamination between samples in the cohort. To this end, we identified all germline variants in each of the samples that were not removed by the common SNPs filter. That is, we focused on non-common germline variants. In all, we identified 509 of these non-common germline variants across the 41 samples in the cohort. Of these, 140 were identified in other samples as somatic mutations, that is, below the 0.3 variant allele frequency threshold. The majority of these 140 mutations were identified at very low variant allele frequencies (below 0.01). The only exception was a mutation in one

hotspot in the TERT promoter, which was identified as a SNP across several samples and was detected as a somatic mutation with a variant allele frequency above 0.01 in 8 samples. In this case this mutation is not a true SNP but a common mutation that is present at high VAF in some samples. In addition, to suspect the existence of a cross-contamination, all germline variants in the source sample should be observed as somatic mutations with similar variant allele frequencies in the contaminated sample. Therefore, we next look at the fraction of germline variants across samples that are identified as somatic mutations across other samples (Supplementary Note Fig. 3.9).

In no sample do we find a majority of the germline variants of other samples detected as somatic mutations. The greater number of germline variants of a sample identified as somatic in another sample (4 samples, 2 donors) is 8, but in each of the 4 cases the source sample possesses more than 100 germline variants, making it extremely unlikely that the eight mutations detected are the consequence of a cross-contamination.

Supplementary Note Figure 3.9. Germline variants identified as somatic mutations in other samples.

The values in the heatmap represent the number of germline variants of a sample (in the corresponding column) that have been identified as somatic mutations in another sample (in the row). The total number of germline variants in each sample appear at the bottom of each column.

Other figure comments:

1. Fig. 1B: would help to plot it as a stacked barplot for each gene to see what mutation types each is comprised of (missense, frameshift, stop codons, synonymous, etc).

Thanks for the suggestion. While this visualization is a good idea, the purpose of Fig. 1b is just to show the difference in scale of the number of mutations identified in the normal

urothelium across individuals in this cohort. Instead, we have included a stacked barplot for each gene with the mutation type in the saturation mutagenesis plots (Fig 5c and Extended Data Fig 9 d and e, and supplementary figure 1).

2. Fig. 3A y-axis label should be reworded as it is not clear what it means without reading the legend.

This figure is no longer in the manuscript

Referee #2 (Remarks to the Author):

In this study, Lopez-Bigas and colleagues report an ultradeep sequencing study of normal bladder epithelium. The key unique feature of the study is the application of duplex sequencing on bulk normal tissue at a very high depth (~7000x) compared with previous studies. The caveat is that this is only done on 15 genes + the TERT promoter, which can really only serve as a pilot study to demonstrate the potential of the technique. Overall, the study looks well executed and the methodology is generally sound. While this is the first study of this type, the findings are more on the descriptive side without very strong data to support the basis/mechanism of interindividual variability or why certain genes are more mutated in normal than tumour samples.

We thank the reviewer for their description of our work, although we respectfully disagree with their assessment that it can *only serve as pilot study to demonstrate the potential of* ultradeep sequencing. Novel and robust results are presented in this manuscript, namely the sex bias in selection for mutations in four genes, the identification of driver TERT promoter mutations in normal bladder and their clear association with age and smoking, and the results of saturation mutagenesis and site selection metrics.

Focusing on 16 genes (15 protein-coding genes and the TERT promoter) with known positive selection in normal urothelium and tumors is a strength of the study, rather than a caveat. Since the aim is to discern the existence of a sex bias in the clonal landscape of the normal urothelium, the correct study design calls to probe genes under positive selection across individuals. Genes evolving under neutrality would not be meaningful for this goal.

For the saturation mutagenesis part and the measurement of selection at the level of individual sites, it also only makes sense when focusing on genes known to be under positive selection in the tissue, as is in these cases that it makes sense to identify which specific sites are under positive selection. Approaching saturation mutagenesis in genes under neutral selection in the tissues would only follow a pattern of neutral mutagenesis.

Below are some specific comments which should be address:

1. The title is slight inaccurate as the method does not directly allow the study of clonal selection. It is only able to infer the presence of clones, but it is not possible to determine the extent of clonal selection since the contribution of convergent evolution of somatic mutations cannot be differentiated.

The title has now been changed to “Sex differences in selection of somatic mutations in normal human bladder”.

We respectfully disagree with the statement that the method does not directly allow the study of clonal selection. While it is impossible through bulk ultradeep duplex sequencing to distinguish whether the same mutation observed twice corresponds to a bigger clone or two independent clones, redundant mutations are excluded from the calculation of positive selection (see Methods section, Supplementary Note 5 and Extended Data Fig. 4). The dNdS (and other positive selection metrics) is computed for unique mutations. The dNdS calculated for each gene, therefore represents the minimum number of expanding clones driven by mutations of this gene observed in each sample.

We also compute the value of dNdS that would correspond to a scenario in which all observed mutations corresponded to independent clones (i.e., assuming complete convergent evolution). These values which are, of course, greater than the default dNdS values explained above are shown just for comparison purposes in Extended Data Figure 4g, copied below. However, for all calculations, we employed the conservative dNdS values, calculated on unique mutations.

Extended Data Figure 4g

Extended Data Figure 4. Calculation of positive selection

g) Comparison of the excess of truncating mutations (dN/dS truncating, top) and the excess of missense mutations (dN/dS missense) calculated taking into account every observed mutations only once (as used in the manuscript, and represented by the unshaded bars in each plot) and taking into account the number of DNA duplex reads supporting each mutation (bars shaded with diagonal pattern).

Importantly, most of the genes profiled in our study act as loss-of-function, and therefore, we don't expect a high degree of recurrence of individual mutations. In agreement with this expectation, most of the mutations we identify have extremely low VAFs (Supplementary Note 3). Actually, the two values of dNdS described above are not very different for most genes. These results, taken together, suggest that most mutations we observe across samples represent a single clone. The exception to this trend are the mutations hitting the two hotspots in the TERT promoter, where we find samples with large VAFs, probably reflecting a higher degree of convergent evolution into the same exact site.

2. It should be made clear in the abstract that this is a targeted sequencing study of 15 genes + the TERT promoter.

We agree with this comment. In the current version of the abstract this is made clear.

3. For the comparison of the normal bladder and bladder tumours, the aggregation of samples and description that mutations detected is 1-2 orders of magnitude higher needs to be explained in a clearer context. The comparison should be described on a per sample or "clone" context, including a range of variability between samples.

If the reviewer refers to Figure 1b, this is only intended for descriptive purposes, to illustrate the difference in scale in the number of mutations identified in these genes across normal samples in our cohort and across tumor samples in over a decade of cancer genomics studies. The aim is precisely to illustrate that while when sequencing tumors, only the mutations in one clone are probed, hundreds to thousands of clones may be observed through ultradeep sequencing of normal tissues. In other words, this is a comparison of the yield of both approaches in terms of "raw" somatic mutations.

A comparison similar to the one proposed by the reviewer is illustrated in Extended Data Figure 9a. In this plot, rather than "raw" mutations, we compare the rate of mutations observed across all genes in normal and tumor samples (of two types of bladder carcinoma). This rate takes into account the length of each gene and the depth of sequencing (i.e., the number of haploid genomes probed in each case, set to two in the case of the tumors). Each mutation in each sample is counted only once, even if multiple reads support it. For the sake of simplicity we have pooled the mutations observed for each gene across all tumor and normal samples. Otherwise, the comparison would be harder to interpret, with tumors showing 1 or 0 for each sample.

Extended Data Figure 9a

Extended Data Figure 9. Calculations of natural selection mutagenesis

a) Comparison of the rate of protein-affecting mutations in 14 genes across two cohorts of bladder tumors (muscle invasive and non-muscle invasive) and in the normal urothelium of the 41 individuals.

4. It is a bit confusing across the study that there are two sets of bladder tumour samples, one from Intogen (867 samples) and one from cBioportal (805 samples). It is not really clear why one is used over the other for different analyses and Figures. For example Fig 1B uses the Intogen cohort, but in the methods under “comparison with bladder tumours” only cBioportal samples are described.

We agree with the reviewer that the number of tumors of different cohorts were not clearly presented in the manuscript. We use the larger cohort obtained from IntOGen comprising 892 bladder cancer samples to carry out all comparisons where all tumors are considered together. The only exception is the comparison of the mutation rate across samples of normal urothelium, and muscle invasive and non-muscle invasive bladder carcinomas separately. The reason to use a different cohort of tumors for this comparison is that non-muscle invasive tumors are underrepresented in IntOGen. Therefore, for this comparison we resorted to cohorts of muscle invasive and non-muscle invasive bladder tumors from cBioportal. This comparison is now shown in Extended Data Fig. 9a.

5. It would be good to add whether a gene is normal/cancer associated in Extended Data Table 3.

Following this comment, we have now added this information in the extended data table (this is now Extended Data Table 2).

6. In the end, was only non-protein altering mutations were used for de novo mutational signature extraction? This didn't seem to have been clearly described.

All mutations are used to identify mutational signatures shown in Figure 1 and described in the first section of Results. A thorough explanation of the rationale behind the use of this

set of mutations and a comparison with the profile of (sparse) non-coding mutations obtained in the ultradeep sequencing are presented in Supplementary Note 4.

7. For estimating the distribution and number of mutations in genes (p.8) it is mentioned that all variants identified in each sample was used. Would it be better to only use non-protein coding mutations for this?

The first section of results and Figure 1 (p.8 of previous version of the manuscript) provide a description of the number of mutations identified in the study per gene and also per sample. All mutations (protein affecting and non-protein affecting) are used here. Nevertheless, we did compare the mutational profile of non-coding mutations and all mutations identified in the panel to understand if the pervasive positive observed on coding mutations in these genes could affect the calculation of mutational signatures and mutation profiles. This analysis is described in Supplementary Note 5, in the section quoted below.

Supplementary Note 5

Moreover, the mutational profiles obtained from all or only non-protein-affecting mutations captured by the panel, are very similar to that obtained from whole-genome mutations from LCMs (cosine similarities, 0.91 and 0.95, respectively; Supplementary Note Fig. 5.6).

Supplementary Note Figure 5.6. Comparison of mutational profiles in normal urothelium. The top row presents the profile of mutations obtained through whole-genome sequencing of LCMs of normal urothelium. The middle row presents the profile of mutations (both protein- and non-protein-affecting) observed in 14 genes across all samples probed in this study using duplex DNA sequencing. The bottom row presents the profile of non-protein-affecting mutations observed across all samples using the study panel. For the middle and bottom rows, the cosine similarity to the profile in the top row is shown.

To further explore the second problem, we directly compared, for each sample, the mutation rate and mutational profile computed using all mutations with those obtained using only non-protein-affecting mutations. For this analysis, we computed, for each sample, the excess of protein-affecting mutations (dN/dS ; see section on calculation of positive selection, below) with respect to the expectation under neutrality as a measure of the overall magnitude of positive selection acting on the mutations of the 15 genes. Despite the existence of a strong positive correlation between the rate of protein affecting mutations and all mutations across samples (Supplementary Note Fig. 5.7a), the higher the excess of mutations over the neutral expectation (higher dN/dS), the higher the degradation of the cosine similarity of their non-protein affecting-to-all mutational profile (Supplementary Note Fig. 5.7b). This is expected, as higher dN/dS values indicate a preponderance of protein affecting mutations. This may impact the identification of mutational signatures active in these samples. Nevertheless, the aforementioned comparison between the non-protein-affecting mutations captured by the panel and that obtained from whole-genome mutations from LCM suggests that this deviation is not large enough to cause a great distortion in the profile.

Supplementary Note Figure 5.7. Effect of pervasive positive selection on the mutational profile of normal urothelium samples.

a) Correlation of the mutation rate observed for all sites and only for non-protein-affecting sites across the genes in the panel. The dots represent samples, and they are colored following the excess of protein-affecting mutations observed in each of them (dN/dS). b) Relationship between the similarity of the mutational profile computed for all mutations and only protein-affecting mutations (cosine similarity, y-axis) and the excess of protein-affecting mutations observed in each sample (dN/dS , x-axis). The downward trend implies that the higher the magnitude of positive selection observed in the sample the more dissimilar (lower cosine similarity) the profile of protein-affecting mutations and that of all

mutations. Both plots show that a high magnitude of positive selection may cause a distortion in the profile of mutations observed in a sample with respect to that expected under purely neutral mutagenesis.

We reasoned that de novo extraction of mutational signatures using only non protein affecting mutations would be the best way to minimize the effect of selection on the observed mutational profile. However, this substantially reduces the number of mutations available for the calculation of mutational signatures. Both SigProfiler and HDP yielded 2 signatures as the most robust solution of the extraction: one of these was similar to SBS-treatment and the second one appeared to be a combination of SBS-age and SBS-APOBEC. This similarity of the mutational signatures extracted using all mutations and only non-protein affecting mutations suggests that the two aforementioned factors do not introduce an important bias in the mutational profile of samples.

8. Is VAF of protein altering mutations also associated with age? Presumably, the mutations may be acquired in stem cells that replenish the epithelium. If that is the case, VAF of mutations under positive selection may also increase with age a kin to clonal haematopoeisis.

The reviewer is correct that the VAF of driver mutations increase with age as the clones they drive expand (and new driver mutations are acquired). The VAF of the mutations observed in these normal samples are in general very small, with many mutations identified in a single DNA duplex read (see Supplementary Note Fig. 3.8), making the analysis of the growth of VAF difficult for the majority of mutations. Nevertheless, the proposed analysis of the change in VAF with age or sex is actually similar to the analysis of increase in dN/dS values used in the study. As clones expand, and VAF increases, the probability of detecting the mutations through ultradeep sequencing increases (even if only one or few molecules are detected per clone). In other words, the analysis of the association of dN/dS with age or sex is equivalent to that of VAF.

9. On p16 the text describing TERT promoter mutations and smoking does not match the results shown in Figure 4 (should be 4G rather than 4I).

We have now performed new analyses and provide more detailed results on TERT promoter mutations and its relation to age and smoking, which are shown in new panels of figure 4 (copied below).

Figure 4

Figure 4. Association of functional mutations in the TERT promoter with smoking

a) Needleplots representing the frequency (count across samples) of functional (observed in tumors) and non-functional (not observed in tumors) mutations in the TERT promoter across donors younger than 55 years old (top), older than 55 with no history of smoking (middle), or older than 55 with a history of smoking (bottom).

b) Rate of TERT promoter functional mutations observed in the three groups of donors. The horizontal line denotes the median of the distribution of the rate of mutations in the group of donors older than 55 with a history of smoking.

c) Association between the rate of functional mutations in the TERT promoter and the interaction between age and smoking history. A linear mixed-effects model was used to account for the presence of multiple samples of the same donor. The circle represents the point estimate of the effect size of the linear regression, and the horizontal line, the confidence intervals. The dark outer circumference denotes a significant association (FDR threshold of 0.2).

d) Distribution of expected number of mutations in the two TERT promoter mutational hotspots across donors younger than 55 years old or never smokers assuming a mutation rate equal to that observed across ever smokers. The red dashed vertical line represents the actual observed number of mutations in the two hotspots across never smokers.

10. The association of TERT promoter mutation with sex. Is it still significant when smoking is accounted for? Again should be Fig 4J rather than 4k).

The new detailed analysis of TERT promoter mutations shown in panels of the new Figure 4 show that TERT promoter mutations are clearly associated with age and smoking, even when accounting for sex. We do not find significant sex differences in TERT promoter mutations.

11. For the associating of sex with other genes, there needs to be p-values reported in both the boxplot (Fig 4K) and the multivariable regression (Figure 4 L). The complete table for the multivariable regression including the significance and contribution of other factors in the male vs female comparison needs to be included in the supplementary tables.

We agree with the reviewer. A complete table with the detailed results of all the regressions are now provided in Extended Data Table 6. In the different figure panels we also include p-values when space allows it, otherwise we indicate in the figure legend that they are available in the Extended Data Table 6.

12. On page 20, why does it say 806 bladder cancer samples when the methods says that there are 805 (p. 29). It is also confusing that Fig 5B has bladder tumours of 867 individuals. To make things clearer, the number of samples should be stated in the header in Figure 5A.

We thank the reviewer for pointing out this typo. All analyses shown in the main Figures are carried out on the intOGen bladder cohort (ref), comprising 892 tumors. The analysis comparing muscle invasive and non-muscle invasive bladder tumors with normal samples (Extended Data Fig. 9a) is based on other cohorts obtained via cBioPortal. Other analyses are based on cohorts from the GENIE project. Following the reviewers recommendation, we now state the number of tumors in the cohort in each relevant figure.

13. The statement on P.23 regarding that no genes outside the ones included in their panel as strong candidate drivers of clonal expansion is not really accurate. For example, in REF 9, RHOA and ERCC2 were found to have significant positive selection. Conversely, as the authors showed, they identified TERT mutations where previous studies of normal bladder epithelium didn't, so the profiling of the limited region reported can really only be seen as a pilot study.

We agree with the reviewer about the possibility that other genes outside the panel are clonal drivers of normal urothelium. Nevertheless, we disagree with the statement that this is only a pilot study. As explained above, the aim of this study was not the discovery of new genes under positive selection, rather to answer whether sex may affect the clonal landscape of the normal urothelium. To that end, we focused on genes previously known to be under positive selection in this normal tissue or in bladder tumors, as the most logical candidates to probe the heterogeneity of the clonal landscape of the urothelium among males and females. Our study has provided an answer to this question. In addition, we

report the discovery of diver TERT promoter mutations in normal bladder for the first time, and their association with smoking. Moreover we provide detailed results of the power of saturation mutagenesis in clonal driver genes to measure positive selection at site resolution.

Referee #2 (Remarks on code availability):

None of the code are current available for review.

Referee #3 (Remarks to the Author):

The study profiles somatic mutations in normal bladder urothelium using ultradeep duplex sequencing. Compared to an early study (Lawson et al. 2020), more somatic mutations, including mutations in TERT and FGFR3 were revealed. Pervasive positive selection and high variability of clonal expansions between individuals were shown, which have been well established in the field. Truncating mutations in FGFR3 under negative selection is considered novel. The authors also linked risk factors to mutagenesis in the normal tissues, and performed a saturation analysis to mutations not seen in cancers. The paper is well written, and the results are clearly presented. However, there are several issues need to be addressed.

1. Total number of individuals sequenced is only 47, making the study very underpowered to detect any association between mutations and epidemiological risk factors, such as smoking. The authors need to be cautious about their conclusions without replication.

Motivated by this and other comments by reviewers we performed a formal power analysis (presented in Supplementary Note 10). This power analysis together with the two main findings presented in this study related to the main bladder risk factors (i) the sex bias of driver truncating mutations in four genes and ii) the association of TERT promoter mutations with smoking) demonstrate that the cohort is sufficiently powered to yield robust results.

More specifically, we carried out a robust experiment to estimate the power of our cohort to test the differences of dN/dS values for different genes between males and females. This study is described in Supplementary Note 10 (quoted below) and its results are illustrated in Extended Data Figure 8 (copied below), and it demonstrates that this cohort provides enough power to detect these associations, given the distribution of dN/dS values across samples and the effect sizes calculated from the multivariate regressions for all genes. We also performed a similar experiment to estimate the power of the study to detect bias of TERT promoter mutations with smoking. In conclusion, the cohort presented in the study is sufficiently powered to detect the biases with sex and smoking reported. We acknowledge

that there may be other associations with lower effect size that are pending to be discovered and will require bigger cohorts.

Extended Data Figure 8

Extended Data Figure 8. Power profile for sex association analysis with truncating dN/dS

From simulated datasets reflecting the same distributional features and data dependencies found in the study cohort, we computed the statistical power as the proportion of times the variable of interest (sex) came out significant in the univariate linear mixed-effects regression against truncating dN/dS. In this analysis the female group was picked as the baseline group. We simulated data with different ground truth female baselines (expected truncating dN/dS among females) and between-group differences (effect size). For each baseline-effect combination we can draw a power value, which are represented collectively in the form of these power profiles (see Supp. Note 10). For the two exemplary genes RBM10 and STAG2 we highlight the profile curves corresponding to their observed baselines in the cohort and the projected power given the inferred effect in the cohort.

Supplementary Note 10

Power to discover sex associations

For all the genes and consequence type combinations we conducted a univariate linear mixed-effects regressions with our data and compared the results with the theoretical power and false positive rate that would correspond in each case given the baseline and inferred effect of the variable of interest. We also computed the proportion of simulations with the same baseline and zero ground truth effect that yielded an effect higher or equal than the one inferred, which we refer to as “effect p-value”: it plays the role of a false positive rate metric whereby we assess the likelihood that effects as high as the observed are reached just by chance due to the data composition.

In Supp. Note Table 10.1 we show the results of the comparison for sex association for those genes that we deem robust associations upon univariate, multivariate and FDR correction in the study: ARID1A, CDKN1A, RBM10 and STAG2 in the context of truncating dN/dS and CDKN1A also in the context of missense dN/dS. The power of the tests is

moderate or high and none of the 100 simulations with ground truth zero of effect (meaning no difference between males and females) reach an effect size as big as the one observed, thus indicating that these results are very unlikely to be spurious (p -value <0.01) (Supp. Note Fig. 10.5).

Supplementary Note Figure 10.5. Boxplots represent the distributions of effects for the corresponding baseline values matching each gene when the ground truth effect is zero. Dots represent the reconstructed effect after univariate linear mixed-effects regression for ARID1A, CDKN1A, RBM10 and STAG2 for dN/dS truncating and CDKN1A for dN/dS missense.

This has two important implications regarding our study design: 1) the cohort is adequate to identify associations as strong as the ones observed in these four genes, 2) the structure of the cohort makes it very unlikely that such strong effects show in the regression analysis just by chance, which further reinforces the conclusion that these four associations are real.

Supplementary Note Table 10.1

GENE	CSQN	ESTIMATE	CI_LOW	CI_HIGH	PVAL	INTERCEPT	COVARIATE	BASELINE	POWER	EFFECT_PVAL
RBM10	truncating	36.43	14.50	58.36	0.00	18.58	is_male	20.41	0.91	<0.01
STAG2	truncating	15.07	3.73	26.41	0.01	7.78	is_male	8.12	0.80	<0.01
ARID1A	truncating	14.34	6.23	22.46	0.00	6.73	is_male	7.41	0.77	<0.01
CDKN1A	truncating	13.14	2.61	23.66	0.01	12.88	is_male	12.66	0.57	<0.01
CDKN1A	missense	1.34	0.11	2.58	0.03	2.08	is_male	2.10	0.45	<0.01

This analysis conclusively shows that the cohort analyzed in our study provides the statistical power to detect the associations observed between the magnitude of positive selection in four genes and sex.

2. TERT promoter mutations appear to be strikingly enriched in smokers in normal tissues in their cohort, but in bladder tumors, TERT mutations are not found to be associated with smoking. The relationship between TERT and smoking seems unclear and should be

further validated and investigated (p value for fig 4g is not impressive). If a replication cohort is not possible, limitations in sample size and replication should be clearly acknowledged.

We have now extended the analysis of TERT promoter mutations and their association with age and history of smoking. We only found driver TERT promoter mutations in older individuals (>55 years old), and almost exclusively across donors with smoking history. The association between TERT promoter mutations and age and smoking is very significant. These results are described in a new section of the results and shown in Figure 4 (copied below). Supplementary Note 10 also presents a formal calculation of the power of the cohort to identify this difference.

Figure 4

Figure 4. Association of functional mutations in the TERT promoter with smoking
 a) Needleplots representing the frequency (count across samples) of functional (observed in tumors) and non-functional (not observed in tumors) mutations in the TERT promoter across donors younger than 55 years old (top), older than 55 with no history of smoking (middle), or older than 55 with a history of smoking (bottom).
 b) Rate of TERT promoter functional mutations observed in the three groups of donors. The horizontal line denotes the median of the distribution of the rate of mutations in the group of donors older than 55 with a history of smoking.

c) Association between the rate of functional mutations in the TERT promoter and the interaction between age and smoking history. A linear mixed-effects model was used to account for the presence of multiple samples of the same donor. The circle represents the point estimate of the effect size of the linear regression, and the horizontal line, the confidence intervals. The dark outer circumference denotes a significant association (FDR threshold of 0.2).

d) Distribution of expected number of mutations in the two TERT promoter mutational hotspots across donors younger than 55 years old or never smokers assuming a mutation rate equal to that observed across ever smokers. The red dashed vertical line represents the actual observed number of mutations in the two hotspots across never smokers.

3. The authors claim that “We adapted and developed new computational methods to quantify multiple metrics of positive selection and showed with orthogonal approaches that positive selection is pervasive in the normal urothelium.” However, it doesn’t seem like there’s a new computational method developed in the study. More functional annotation, such as protein structure information is added, but it should not be considered as a new method. Please clarify.

We use four methods to quantify positive selection across genomic elements and samples based on SNVs (OncodriveFML, Oncodrive3D, Omega) and one additional method based on indels (frameshift enrichment). Two of these methods, OncodriveFML (<https://genomebiology.biomedcentral.com/articles/10.1186/s13059-016-0994-0>), and Oncodrive3D (<https://www.biorxiv.org/content/10.1101/2025.01.11.632354v3>) had been previously developed to calculate positive selection on tumor somatic mutations and have been adapted to work on ultradeep sequencing data by accounting for differences in sequencing coverage at different genomic positions in a gene and between different samples. These adaptations are described in detail in Supplementary Note 6. Omega (<https://github.com/bbgilab/omega>) and frameshift enrichment have been developed specifically for this study, and their design and implementation are described also in the same supplementary note.

These methods to detect positive selection in the pattern of mutation in a gene across samples exploit deviations of this pattern from that expected under neutrality. These may be deviations in several features, such as the number of protein-affecting mutations, the distribution of mutations in the three-dimensional structure of the protein, or a bias in the functional impact of observed mutations. The aim of the methods described in the section mentioned by the reviewer do not annotate features of observed mutations, but rather implement different statistical approaches to compute deviations of several of these features of the pattern of mutations and test their significance.

In addition, in this new version we describe a new approach to obtain metrics of selection, at the level of individual site or protein residue. Also described in detail in Supplementary Note 6.

4. Are all individuals selected in the study cancer-free? Why some of them displayed chemotherapy mutational signature and received chemotherapy and radiotherapy? It is unclear if the investigators performed pathological review to ensure that all samples sequenced are indeed normal tissues.

All individuals included in the study had no prior history of bladder cancer (Supplementary Note 2). The absence of bladder cancer at the time of autopsy was confirmed by the pathologist prior to the sample collection. Nevertheless, some of them received chemotherapy and/or radiotherapy for the treatment of tumors in other anatomical sites. All this information is available in Extended Data Table 1.

5. Comparison to the Lawson et al 2020 paper should be further investigated and novel findings from this study should be clearly stated. For example, RBM10 and sex bias is reported. Lawson et al reported mutational signature associated with smoking, which is not mentioned in this study.

It is important to clarify that the sex bias in clonal selection in normal urothelium reported in our manuscript is a novel result not described in Lawson et al 2020 (Science 2020, PMID 33004514) or any other manuscript to the best of our knowledge. To be precise, in the paper by Lawson et al 2020 sex differences are mentioned only in the following paragraph, in a section describing differences in driver preference across individuals, acknowledging the lack of power (9 female and 6 male individuals) in their study to observe sex differences. RBM10 is only referenced as being in the X-chromosome and citing another paper that mentioned differences in tumors, but they do not describe any sex difference in RBM10 mutations in normal tissue.

It is unclear whether these differences are driven by variability in environmental exposures or by the genetic background of each individual. No clear evidence of pathogenic germline mutations was found in these genes (22). KDM6A and RBM10 are both located on the X chromosome, and KDM6A is known to escape X-chromosome inactivation, with some evidence suggesting that both KDM6A and RBM10 are more-frequently mutated in males across cancer types (27). However, in our limited cohort, KDM6A appears to be more-frequently mutated in women than men, which is in line with previous observations in non-muscle-invasive bladder cancers (28). Larger cohorts would be required to establish robust associations between epidemiological factors and differences in somatic mutation rates and selection.

These anecdotal observations are illustrated in this figure (Figure 2G in the Lawson et al., manuscript).

Our study is the first to report a sex bias in the number of truncating mutations across four genes (ARID1A, RBM10, STAG2 and CDKN1A), and this finding is robustly supported by the data. This is based on two main differences with respect to this prior study: i) larger number of clones probed per individual (in the order of hundreds or thousands, rather than dozens), and ii) a bigger cohort (71 samples, vs 15). The impact of these two elements in the robustness of the results presented in our study are discussed below at length.

In summary, our study is, to the best of our knowledge, the first to report the existence of a significant sex bias in the degree of positive selection for ARID1A, STAG2, RBM10 and CDKN1A. Other novelties of our study are the finding that there is a strong significant association between the presence of functional mutations in the TERT promoter and smoking, and the description of metrics of natural saturation mutagenesis for the genes under study.

6. Statistical test is missing in several plots, for example, extended Fig 6c for RB1, where the mutation differences between tumor and normal is interesting. Would be also interesting to include some power simulation in the saturation analysis – with how many samples we will be able to observe all driver mutations for clonal expansion?

We have included statistical significance for all regression plots in Extended Data Table 6. Accompanying plots included with the aim of illustrating the distribution of dN/dS values across groups do not include statistical significance, as they would not account for relevant covariables.

The suggestion about saturation mutagenesis is very interesting. Following this comment, we have carried out a new analysis, which is now described in the last section of Results (quoted below) and illustrated in Figure 5b (copied below) and Extended Data Figure 9c.

Figure 5b

Figure 5. Natural saturation mutagenesis

b) Theoretical and observed curves of saturation mutagenesis for TP53 and FGFR3. The black dashed line represents the kinetic of saturation mutagenesis under the theoretical assumption of no selection, in which mutations are observed based only on their neutral probability of occurrence. The red circle denote the degree of saturation achieved by probing the 71 samples in the cohort. The red dashed line is constructed through successive down-samples of the current observation and represents the observed kinetic of natural saturation mutagenesis.

Nevertheless, we have already observed enough mutations to be able to calculate the magnitude of positive selection on sub-genic elements, such as exons and domains. We have also been able to estimate the degree of positive selection at the level of individual sites. These new results are described in the natural saturation mutagenesis section of the Results (quoted below).

Results extract

Next, we asked how many more genomes would need to be sequenced to observe mutations on all amino acid residues of each of these genes. To estimate this, we first calculated the probability to observe each mutation in the gene under neutrality –due to different mutational processes. Thus, we obtained a theoretical curve describing the fraction of all possible mutations that are expected to be observed at different sequencing depths (theoretical kinetic of natural saturation mutagenesis; Fig. 5b continuous curve; Extended Data Fig. 9c; Supplementary Note 11). The shape of this curve is determined by the difference in probability across mutations, governed by their trinucleotide contexts, with some mutations much more likely to occur than others. For example, for TP53, with the accumulated depth provided by the entire cohort (~350.000x), only on the basis of neutral mutagenesis, we expect to have observed less than half of all possible mutations. In theory, being able to detect every possible mutation in the gene would require an accumulated depth of $\sim 10^7$.

Nevertheless, the observation of mutations in normal urothelium is not neutral, but instead strongly affected by selection. To obtain the actual curve of the fraction of observed mutations depending on the number of sequenced genomes (observed kinetic of natural saturation mutagenesis), we subsampled the number of mutations observed in each gene across the cohort respecting their variant allele frequency, but reducing the depth of sequencing at each position (Fig. 5b circle and dots; Extended Data Fig. 9c). The divergence in the shape of the two curves reveals the importance of selection in shaping

the probability to observe a given fraction of mutations in a gene. For example, since missense and truncating TP53 mutations are under positive selection, we have actually observed more mutations than predicted by the theoretical kinetic. For several genes, we observed this faster accumulation of mutations than expected under neutral mutagenesis due to positive selection. Interestingly, the curve for FGFR3 falls below the theoretical one, constructed under purely neutral mutagenesis (Fig. 5c), which supports the notion that FGFR3 mutations are under negative selection in the normal urothelium.

7. Fig3A is hard to visualize, with pTERT being the outlier and other genes are barely visible.

We appreciate and agree with this comment by the reviewer. The distribution of the fraction of urothelium with driver mutations of the different genes is now presented as a heatmap, with the values discretized in Extended Data Figure 7a.

Extended Data Figure 7a

Extended Data Figure 7. Heterogeneity of clonal landscape across individuals

a) Landscape of the fraction of urothelium covered by driver mutations of each gene in the panel. The values of covered urothelium have been discretized. In most cases, the fraction of urothelium covered by driver mutations of each gene falls in the lowest category.

Referee #3 (Remarks on code availability):

Authors indicated code will be available upon publication.

Referee #4 (Remarks to the Author):

In this manuscript, Blanco et al. investigate somatic clonal selection in normal urothelium using targeted duplex sequencing in 42 individuals. This study aims to identify factors that contribute to inter-individual variation in clonal selection and expansion. While the research question is intriguing, it remains largely unanswered by the current study, most likely due to the flaws and limitations in the study design, namely the small targeted gene panel and the

small cohort size. As a result, the study provides very few new insights into somatic mutagenesis and clonal selection in normal urothelium.

We respectfully disagree with the reviewer's statement that the selection of genes to probe in the study and the size of the cohorts are flawed and prevent us from answering our research questions.

The question of the influence of cancer risk factors on the clonal landscape of the urothelium is not left unanswered by our study. Instead, it provides strong evidence of the association between two such factors (sex and smoking) with inter-individual heterogeneity in clonal selection and expansion: i) the sex bias of truncating driver mutations in four genes and ii) the association of activating mutations in TERT promoter with smoking.

We realized that the main result of the study, namely the sex differences in selection of somatic mutations in normal human bladder, was not sufficiently highlighted in the previous version of the manuscript. We have now rewritten the manuscript to highlight this main result and provide a more advanced analysis of TERT promoter mutation and its relation to age and smoking.

With respect to the selection of genes, our aim was not to identify new genes under positive selection in the normal urothelium, as this was already thoroughly done in Lawson et al. 2020 (Science 2020, PMID 33004514). The goal of our study –as clearly stated in the current version of the paper– was to determine whether the clonal landscape of the normal urothelium shows a sex bias, motivated by the big sex bias in the incidence of bladder carcinomas. To achieve this goal, we relied on the detection of the number and nature of expanding clones in the urothelium across individuals, which we regressed against their sex, using as covariables other features that could influence this bias, such as age and smoking history. To detect these expanding clones in the urothelium, we needed to probe genes under positive selection; mutations in genes under neutrality are not expected to confer a selective advantage to the cells where they occur, and therefore, they are not expected to clonally expand. Hence, to achieve this goal, it only made sense to probe the mutations of genes under positive selection in the normal urothelium. This is why we focused on 10 genes which had been found to be under positive selection in the normal urothelium in Lawson et al. 2020 (Science 2020, PMID 33004514), with more than 10 mutations across all microbiopsies analyzed. We included four more genes and the TERT promoter, that are frequently mutated across bladder tumors.

We have now clarified this point in the Introduction of the manuscript. The second section of Results has been completely re-written to explain the novelty of the calculation of positive selection on each gene at the sample level, rather than presenting the re-discovery of these genes in our cohort. The figures illustrating the calculation of positive selection in the pooled samples has subsequently been moved to Extended Data Figure 4.

With respect to the size of the cohort, we have now thoroughly explored the power of our cohort to detect associations between the clonal landscape of the urothelium and cancer risk factors. This exercise and its results are now described in depth in Supplementary Note 10 and Extended Data Figure 8. This analysis demonstrates that this cohort provides enough power to detect these associations, given the distribution of dN/dS values across samples and the effect sizes calculated from the multivariate regressions for all genes.

In addition, we show that it is possible to obtain a fine map of the degree of selection acting on mutations at individual sites or regions of a protein throughout the normal development of tissues of individuals exposed to different external factors. This natural saturation mutagenesis constitutes a potential key development towards personalized cancer medicine.

Below, we answer these points in detail.

Major points:

1. The cohort size is too small to address the major aim of this study effectively. As I understand, the primary goal of this study is to investigate the factors contributing to inter-individual variation in clonal selection and expansion in normal urothelium. This question inherently requires a large or even epidemiological-scale cohort to be addressed. However, the current study only included a cohort of 42 individuals which is highly insufficient to draw meaningful conclusions about the inter-individual variation in clonal selection. What makes the authors reckon that the current cohort size is adequate to answer this question?

The limited sample size appears to prevent the study from providing new insights into the factors influencing clonal selection in normal urothelium beyond what has already been reported.

For example, Figure 3b shows the extremely heterogeneous selection/preference of driver mutations in urothelium across individuals. Similar results were also observed by Lawson et al in their study. The fascinating question of which epidemiological factors drive/associate with the inter-individual variation in driver gene selection remains unanswered here, as the study does not provide new evidence or clues. This limitation seems largely attributable to the small cohort size, which undermines the study's ability to achieve its stated objective.

We explored the power of the cohort to detect the associations between sex and clonal landscape that constitute one of the main findings of the study. The result of this exercise clearly shows that our study design is perfectly powered to address the question of sex differences in clonal selection.

The results of this exercise are described in Supplementary Note 10 and shown in Extended Data Figure 8.

Extended Data Figure 8

Extended Data Figure 8. Power profile for sex association analysis with truncating dN/dS

From simulated datasets reflecting the same distributional features and data dependencies found in the study cohort, we computed the statistical power as the proportion of times the variable of interest (sex) came out significant in the univariate linear mixed-effects regression against truncating dN/dS. In this analysis the female group was picked as the baseline group. We simulated data with different ground truth female baselines (expected truncating dN/dS among females) and between-group differences (effect size). For each baseline-effect combination we can draw a power value, which are represented collectively in the form of these power profiles (see Supp. Note 10). For the two exemplary genes RBM10 and STAG2 we highlight the profile curves corresponding to their observed baselines in the cohort and the projected power given the inferred effect in the cohort.

Supplementary Note 10

Power to discover sex associations

For all the genes and consequence type combinations we conducted a univariate linear mixed-effects regressions with our data and compared the results with the theoretical power and false positive rate that would correspond in each case given the baseline and inferred effect of the variable of interest. We also computed the proportion of simulations with the same baseline and zero ground truth effect that yielded an effect higher or equal than the one inferred, which we refer to as “effect p-value”: it plays the role of a false positive rate metric whereby we assess the likelihood that effects as high as the observed are reached just by chance due to the data composition.

In Supp. Note Table 10.1 we show the results of the comparison for sex association for those genes that we deem robust associations upon univariate, multivariate and FDR correction in the study: ARID1A, CDKN1A, RBM10 and STAG2 in the context of truncating dN/dS and CDKN1A also in the context of missense dN/dS. The power of the tests is moderate or high and none of the 100 simulations with ground truth zero of effect (meaning no difference between males and females) reach an effect size as big as the one observed,

thus indicating that these results are very unlikely to be spurious (p -value <0.01) (Supp. Note Fig. 10.5).

Supplementary Note Figure 10.5. Boxplots represent the distributions of effects for the corresponding baseline values matching each gene when the ground truth effect is zero. Dots represent the reconstructed effect after univariate linear mixed-effects regression for ARID1A, CDKN1A, RBM10 and STAG2 for dN/dS truncating and CDKN1A for dN/dS missense.

This has two important implications regarding our study design: 1) the cohort is adequate to identify associations as strong as the ones observed in these four genes, 2) the structure of the cohort makes it very unlikely that such strong effects show in the regression analysis just by chance, which further reinforces the conclusion that these four associations are real.

Supplementary Note Table 10.1

GENE	CSQN	ESTIMATE	CI_LOW	CI_HIGH	PVAL	INTERCEPT	COVARIATE	BASELINE	POWER	EFFECT_PVAL
RBM10	truncating	36.43	14.50	58.36	0.00	18.58	is_male	20.41	0.91	<0.01
STAG2	truncating	15.07	3.73	26.41	0.01	7.78	is_male	8.12	0.80	<0.01
ARID1A	truncating	14.34	6.23	22.46	0.00	6.73	is_male	7.41	0.77	<0.01
CDKN1A	truncating	13.14	2.61	23.66	0.01	12.88	is_male	12.66	0.57	<0.01
CDKN1A	missense	1.34	0.11	2.58	0.03	2.08	is_male	2.10	0.45	<0.01

This analysis conclusively shows that the cohort analyzed in our study provides the statistical power to detect the associations observed between the magnitude of positive selection in four genes and sex.

2. Targeted sequencing on 16 genes: this study only did targeted sequencing on 15 genes and TRET promoter, which largely limits its power to uncover novel findings that one would expect. This constrained approach likely explains why limited new findings are reported in the paper. For instance, in the section discussing positive selection in the urothelium, all 13 genes identified under positive selection have already been reported in earlier studies.

What rationale do the authors have for believing that focusing on such a limited set of 16 genes would yield new insights? This is particularly also relevant for mutational signature analysis, as the constraints imposed by the targeted panel limit the coverage of trinucleotide sites.

We regret that study design decisions were not clear in the previous version of the manuscript. As explained above (and clarified in the new version of the paper), the goal of our study is not to identify new genes with positive selection in the bladder urothelium, but to ask if there are sex differences in the clonal composition. To address this question it is relevant to focus on genes already known to be important in shaping the clonal landscape of normal urothelium. The decision to focus on 16 genes (15 protein-coding genes and the TERT promoter) was based on the prior knowledge that these were known to be under positive selection in the normal urothelium and/or frequent bladder cancer drivers. Since the goals of our study were to discern the existence of a sex bias in the clonal landscape of the normal urothelium and to approach natural saturation mutagenesis, (and not potentially discovering new genes under positive selection in the normal urothelium) studying genes under positive selection across individuals was the only reasonable choice from the point of view of cost-efficiency.

Including more genes in the study would have increased the costs of sequencing, or decreased the depth of sequencing obtained across samples, thus eroding the power to identify large numbers of clones per sample and test for differences of the clonal landscape across individuals. In addition, the study of natural saturation mutagenesis is relevant in genes that are positively selected and requires ultradeep sequencing to detect a sufficient number of mutations to make it informative. We have now clearly shown this relationship between the depth of sequencing and the power to identify positive selection (that is, expanding clones) through a careful downsampling exercise, described in Supplementary Note 8, with summary results presented in Extended Data Fig. 8.

3. Using cytobrushing samples: The authors collected cells from bladder urothelium by brushing large surface areas. This method inevitably introduces contamination from other cell lineages, such as immune and stromal cells residing in the lamina propria. Have the authors evaluated the contamination from other cell lineages using methods beyond HE staining and pathological examination? It is critical to rigorously evaluate contamination through additional approaches, as the reported observations may be confounded by the presence of non-urothelial cell types.

This is a very important comment. To assess the fraction of epithelial cells in the brushes of the urothelium obtained at autopsy, we used immunohistochemistry staining with the following antibodies for specific cell types: cytokeratin 7 (CK7) for epithelial cells, uroplakin II for urothelial cells (specifically the upper layer of urothelial cells called umbrella cells), CD45 for lymphocytes, and actin for muscle cells (Supplementary Note 2). We observed that the urothelial brushes contained predominantly normal epithelial cells, as demonstrated by almost uniformly positive CK7 staining. Umbrella cells were also observed, as expected

due to the superficial sampling of the tissue. The average percentage of CD45 positive lymphocytes was low (3.8%) and actin staining was negative indicating minimal contamination with blood cells and no contamination with stroma. Overall, these results confirm that the sampling of urothelium with brushes mostly collects epithelial cells, thus validating this approach for the study of clonal expansions in bladder urothelium.

Supplementary Note Figure 2.2

Supplementary Note Figure 2.2. Determination of cell composition of urothelial brushes by immunohistochemistry. A representative urothelial brush from the bladder dome was fixed in Cytolyt, spun down, and embedded in paraffin to make a cell block. Slides sectioned from the cell block were stained with hematoxylin-eosin (H&E) to determine cell morphology, and immunostained with the following antibodies for specific cell types: cytokeratin 7 (CK7) for epithelial cells, uroplakin II specific for upper layer of urothelial cells (strongest in umbrella cells), CD45 for lymphocytes (scattered cells highlighted by red arrows) and actin for muscle cells. Representative pictures at 20x and 40x are shown.

4. Duplex sequencing: As has been raised by Abascal et al in their Nanoseq study previously, duplex sequencing can suffer from errors introduced during end repair, leading to a much higher error rate than the theoretically expected. How did the authors avoid/handle/improve this in their experiments? How accurate is the mutation detection in the study? This requires a thorough evaluation, but the current manuscript provides only limited information, as shown briefly in Extended Data Fig. 1D.

Motivated by this and a similar comment by reviewer 1, we have re-calculated the error rate of the DNA duplex sequencing technology employed in our study directly on three cord blood samples with the same panel used in this study. The results are very similar to those previously obtained on pediatric blood samples on the previous version of the manuscript. In the new version of the manuscript, the result of these experiments are presented in Extended Data Figure 2 (copied below), and discussed in the Supplementary Note 4 (quoted below).

Extended Data Figure 2a

Extended Data Figure 2. Error rate of the DNA duplex sequencing technology

a) Left panel, orange bars, mutation rate detected by the DNA duplex sequencing technology in this study using the same panel of genes in cord blood samples from three donors. The comparison of this observed mutation rate with that expected in these samples yields an estimate of the error rate of the technology, as $\sim 4 \times 10^{-8}$ per sequenced base pair. Blue bars, mutation rate detected by other similar technology (NanoSeq; ref; data taken from ref) in two cord blood samples (Supplementary Note 4). Right panel, comparison of the estimated error rate by both technologies with the mutation rate detected across the 71 normal urothelial samples included in this study (rightmost red bar). It shows that the rate of errors of the DNA duplex sequencing technology used in the study is approximately 25 times smaller than the mutation rate detected in the normal urothelium.

Supplementary Note 4

Estimation of the error rate of the technology in cord blood samples

We estimated the error rate intrinsic to the duplex sequencing technology employed to detect mutations in this study. To that end, we constructed DNA duplex libraries using the same kits and the same capture panel as in this study for three cord blood samples obtained from newborns (StemCell; Extended Data Fig. 3x). The samples were sequenced to depths of 2985x, 4608x and 4308x. We identified 35, 26 and 47 SNVs in these three samples, accounting for mutation rates of 6.1×10^{-8} , 3.1×10^{-8} and 5.7×10^{-8} mutations per sequenced base (Extended Data Fig. 2a). We obtained the average number of mutations observed in newborn blood from two studies^{18,19}, based on the expansion of clones from hematopoietic stem cells. According to these two studies, the number of somatic SNVs in cord blood is between 60 and 105, yielding an expected mutation rate between 2×10^{-8} and 3.5×10^{-8} .

Comparing the rate of mutations detected in these three samples with that expected in newborn blood, we estimated that the rate of errors introduced by the technology using these three samples is not greater than 4.1×10^{-8} . This is the value shown in Extended Data Figure 2a.

We also compared the mutational profile and the mutation rate of our technology with that of whole-exome sequencing of laser capture microbiopsies from Lawson et al 2020, which are also normal urothelium obtained during autopsy. This analysis showed remarkable similarities in mutation profile and mutation rate (Extended Data Fig. 2b,c, copied below; Supplementary Note 4, quoted below), indicating that the error rate of our approach is low.

Extended Data Figure 2b,c

b) Mutational profile of normal urothelium obtained through two orthogonal approaches. Top panel, profile constructed using mutations detected through laser capture microdissection of clonal or quasi-clonal samples followed by regular shallow whole-genome sequencing (data taken from ref). Bottom panel, profile constructed using mutations detected in this study from $\sim 2\text{cm}^2$ brushes followed by ultradeep DNA duplex sequencing.

c) Relationship between the mutation rate calculated in normal bladder urothelium using two orthogonal approaches. The red dots correspond to the rate of mutations detected through laser capture microdissection of clonal or quasi-clonal samples followed by regular shallow whole-exome sequencing (LCM+WES; data taken from ref). The blue dots correspond to the rate of mutations computed for the 71 samples in this study from $\sim 2\text{cm}^2$ brushes followed by ultradeep DNA duplex sequencing. The trend line represented in the plot was calculated from the LCM+WES samples.

Supplementary Note 4

Comparison of ultradeep sequencing with other technology

To have an orthogonal comparison of the error rate we can compare the mutation pattern and mutation rate of our study with that of normal bladder urothelium from autopsies obtained through laser capture microdissection of clonal or quasi-clonal structures (ref). In this case the sequencing approach is bulk whole-genome sequencing, which allows the comparison of duplex and bulk genome sequencing in the same tissue and type of samples.

This comparison revealed strikingly similar mutational profiles (Extended Data Fig. 2b), suggesting the absence of a large number of artifacts with a different profile across the mutations identified in the duplex sequencing of brushes of normal urothelium. In addition, the mutation rate of the normal urothelium in our study revealed a very close match with the values obtained in the laser microdissection study (Extended Data Fig. 2c), again pointing to a low proportion of errors in our study. Moreover, the distribution of mutation rate across both cohorts is very similar, as is the linear relationship between mutation rate and age, which would be incompatible with high error rates across the samples in our cohort.

Based on the reviewer's comment, we also reviewed the potential error rate across all samples in the cohort. We visually compared the mutational profile of samples with low and high error rate, as well as the mutational profile of all samples in this cohort probed through ultradeep sequencing with the normal urothelium samples obtained by Lawson et al. (2020) using laser capture microdissection and sequenced at the whole-genome level. Moreover, we compared the distribution of mutation rate calculated for normal urothelium samples using these two orthogonal technologies. All these analyses (described in Supplementary Note 4, quoted below) suggest that the level of error across samples is not sufficient to alter neither their mutational profile, nor their mutation rate.

Supplementary Note 4

Mutational profile of samples with different mutation rate

This comparison is based on the error rate calculated in cord blood, a scenario in which we do not expect positive selection, in contrast to what we observe in the normal urothelium of adults. In addition, the type of sampling is different in both scenarios. Thus, to understand whether this rate of errors posed a problem in its comparison with different mutation rates across samples, we inspected the mutational profile of the two samples with the lowest mutation rate (01_DO and 27_TR), and of two among those with the highest mutation rate, with the mutational profile of the pooled cohort. We reasoned that a higher prevalence of artifacts in the samples with lower mutation rate should be reflected through the observation of a more different mutational profile. However, the comparison reveals that the profile of these samples is very similar to that of the pooled cohort (cosine similarity = 0.83 and 0.74, respectively; Supplementary Note Fig. 4.1).

Supplementary Note Figure 4.1. Mutational profile of the pooled cohort (top) and the two samples with the lowest (left) and highest (right) mutation rate. Cosine similarity with the mutational profile of the cohort is shown for each sample.

Moreover, we expect that the presence of artifactual mutations across samples in the cohort would result in the extraction of a mutational signature representing them. As discussed in Supplementary Note 5, this is not the case.

Furthermore, we systematically explored the effect of an error rate comparable to or higher than that estimated in cord blood on the estimation of dN/dS values across genes (Extended Data Fig. 6a-c and Supplementary Note 8, copied below). We found that error rates similar to or higher than those estimated in cord blood cause only a mild reduction in the values of dN/dS computed across genes. Moreover, it is not possible that these dN/dS values are spuriously generated by sequencing errors.

Extended Data Figure 6a-c

Extended Data Figure 6. Tolerance of dN/dS values to errors

a) Measurement of the tolerance of RBM10 dN/dS truncating to artifactual mutations (artifacts) following the BotSeq mutational profile (ref.). The boxplots represent the distribution of dN/dS values calculated from 100 synthetic samples with increasing rates of injected artifacts between 0 and 1×10^{-7} (one order of magnitude higher than estimated for the technology), and for increasing values of ground truth dN/dS (between 1 and 50).

b) Average percentage of RBM10 dN/dS truncating reconstructed value across 100 synthetic samples, calculated by computing which fraction of the ground truth dN/dS in a sample is obtained upon calculation.

c) Summary of the results of the experiment of error tolerance. Left panel, average percentage of reconstructed ground truth dN/dS across synthetic samples (for all genes and all ground truth dN/dS explored altogether) that is calculated upon injection of increasing rates (x-axis) of different types of artifacts (color legend). Center plot, average percentage of reconstructed ground truth dN/dS across synthetic samples (for all genes and all artifacts altogether) that is calculated upon injection of increasing rates (x-axis) for different values of ground truth dN/dS (color legend). Right panel, average percentage of reconstructed ground truth dN/dS across synthetic samples (for all ground truth dN/dS explored and all artifacts altogether) for different genes (color legend), that is calculated upon injection of increasing rates (x-axis) of artifacts.

Supplementary Note 8

Tolerance of positive selection calculations to errors

We wondered how artefacts detected in the samples (as they occur randomly across genomic positions) impact the magnitude of positive selection estimated for different genes. We wanted to assess how the dN/dS estimates calculated with Omega across samples could be impacted by artifactual mutations (herein 'errors') generated with the error rates estimated from cord blood.

We simulated synthetic catalogs of mutations (i.e., synthetic samples) whereby two signals are combined: ground truth mutations (obtained through sampling of the mutational profile of a real bladder sample) distributed across synonymous and non-synonymous sites to obtain a pre-defined dN/dS value, and error mutations. The baseline synthetic samples only contained "true" mutations sampled from the mutational profile of the cohort. In different simulations, we then injected increasing levels of artifacts sampled from mutational profiles representing different sources of errors.

Artifacts' mutation probability vectors

We tested the tolerance of Omega to known frequent sequencing artifacts (SBS43, SBS45, SBS52, SBS58 in the Cosmic reference catalog of mutational signatures; ref), and to the BotSeq mutational profile in cord blood reported in Abascal et al. (Nature 2021), which is enriched in artifactual C>A mutations common to this technology (artifacts hereafter) (Supplementary Note Fig. 8.1).

Results of the experiment

We calculated the dN/dS of each of the genes probed in this study across a set of synthetic samples with a given ground truth dN/dS (between 1 and 50), and a rate of errors (between 10^{-9} and 10^{-7} per base pair sequenced) injected on top of the true mutations. This calculation was replicated across 100 synthetic samples. As an example, Supplementary Note Figure 8.2a shows the reconstructed dN/dS truncating for RBM10 after receiving injections of increasing artifactual mutations generated in accordance with the BotSeq C>A mutational profile (ref; Supplementary Note Fig. 8.1). We observed that increasing rates of errors decrease the estimated dN/dS. However, even in the most extreme case of injecting one artifactual mutation every 107 base pairs sequenced, Omega is capable of reconstructing more than 80% of the ground truth dN/dS (Extended Data Fig. 6a,b).

Extended Data Figure 6c shows the overall reconstruction of dN/dS across 5 artifact sources (see Supplementary Note Fig. 8.1), different ground truth values of dN/dS and genes. As expected, artifactual mutations do not have an effect in the dN/dS estimates when the ground truth value is 1 (i.e. no selection). With increasing values of ground truth dN/dS, the reduction in dN/dS increases, although it is comparable across scenarios of high selection (i.e. ground truth dN/dS of 5, 10 and 50). Despite minor differences, the trend of underestimation of dN/dS across artifact sources and genes is maintained.

5. It is reasonable to assume that both germline and environmental factors contribute to somatic clonal selection in normal urothelium across individuals. However, germline variables are not considered at all in the current study. To what extent might the observed variation in clonal selection be driven by germline factors?

This is a very interesting question, but out of the scope of this study. We agree with the reviewer that differences in the germline of individuals can influence the clonal landscape of their normal tissues.

6. The authors reported a high proportion of shared mutations between dome and trigone, which is unexpected and potentially concerning. What are the VAFs of these shared mutations? How many of them are driver mutations? Can the authors validate these findings using orthogonal approaches? The authors appear surprised by the similar patterns of mutation and clonal selection between the dome and trigone. What reasons led the authors to anticipate differences? Are there any previously reported "geographical" differences in mutation or clonal selection patterns in bladder cancer?

These are all interesting questions. We will answer them step by step.

VAF of the mutations shared between dome and trigone tend to be higher than that of non shared mutations. This is not unexpected under scenarios of intraluminal seeding, convergent evolution or cross-contamination during autopsy (see below).

Here, we show the comparison between the VAF distribution of the shared mutations vs the non-shared mutations for a couple of exemplary samples. The p-value annotated in the figure is the result of a Mann-Whitney test comparing the two groups.

Since there is a good agreement between dN/dS values for dome and trigone (including or excluding shared mutations; Fig. 2g and Extended Data Fig. 5), the number of driver mutations of each gene are similar between the two samples of the same donor.

Although we find this higher than expected proportion of shared mutations very intriguing, we cannot easily validate it with orthogonal approaches. For this reason we decided to explain the observation transparently, mentioning the different possibilities for this intriguing observation. One possibility is that it is related to intraluminal seeding, which has been previously proposed as an explanation for multifocal bladder tumors. A second possibility is that it is due to cross-contamination during autopsy sampling. Finally, shared mutations can also be originated through convergent evolution. We decided not to highlight it as an important result of the manuscript, as we cannot rule out these possibilities. In any case, it is important to note that the high correlation of positive selection per gene between dome and trigone remained similar even when accounting only for non-shared mutations (Extended Data Fig. 5d). And none of the other results of the manuscript are affected by the proportion of shared mutations, as all analyses are done at sample level. Moreover, as we employed the linear mixed-effects models for the regressions, the fact that there are samples from the same donor does not affect the results.

The authors appear surprised by the similar patterns of mutation and clonal selection between the dome and trigone. What reasons led the authors to anticipate differences?

The dome and the trigone are functionally and structurally different parts of the bladder, with the trigone having more time with urine exposure given its location at the bottom of the bladder. This motivated us to collect two samples per individual from these two separated regions.

The finding of the striking similarity in the patterns of clonal selection between both samples of the same donors indicates that, despite their geographic, functional and structural differences, the evolutionary dynamics and selective forces in the two regions are equivalent. These results also demonstrate that our approach of ultradeep sequencing urothelial brushes reproducibly measures positive selection and therefore, a brushing of cells from $\sim 2\text{cm}^2$ of the urothelium provides a good representation of the clonal structure of the entire tissue.

Are there any previously reported "geographical" differences in mutation or clonal selection patterns in bladder cancer?

Bladder cancer is classified into muscle invasive and non-muscle invasive but in neither type there is a geographical difference. Bladder cancer is well known for cases of multifocal tumors, its widespread field cancerization, that is, the presence of genetic alterations in the normal urothelium of patients with bladder cancer. Our findings from patients without bladder cancer illustrate that somatic clonal evolution is indeed widespread in the normal bladder and that there are no regional differences, indicating common processes of mutagenesis and selection across the organ.

7. The authors reported sporadically observed large clones in the urothelium, such as two mutations in TERT promoter with a VAF of 14% and 20% in one individual. How confident are the authors about this? Have they validated these results using orthogonal approaches?

We are very confident about the presence of these mutations and their high VAF because: 1) these are not random mutations but canonical hotspot mutations observed in tumors (see new manuscript Fig. 5F). 2) The high VAF indicates that multiple duplex reads support the mutation thus it is very unlikely to be artifactual. 3) For several donors with dome and trigone data, the same pTERT mutation is found at similar high VAF in both samples, providing an internal validation of the results and indicating that most likely these mutations are being repeatedly selected in multiple cells across the tissue.

Additionally, we have validated our results by comparing the strength of site selection observed in our data with the functional impact determined experimentally (ref) for all pTERT positions. We demonstrate that the mutations with most significant site selection have high functional impact in this experiment (Fig. 5f, copied below). In other words, the pTERT mutations that are most frequently found in our data correspond to mutations that affect protein function.

Figure 5f

Figure 5. Natural saturation mutagenesis

f) Natural saturation mutagenesis of the TERT promoter. Top track, needleplot representing the distribution of mutations along its sequence. Second track, site selection computed for each observed mutation. Third track, functional impact values experimentally determined for possible mutations in the TERT promoter. Fourth track, needleplot representing the distribution of mutations observed in the TERT promoter across 8136 tumors. Right plots, top, functional impact values experimentally determined for mutations not observed,

observed, or observed with significant site selection across the 71 samples. Bottom, relationship between the site selection and the experimentally determined functional impact value for all mutations observed in the TERT promoter.

Furthermore, the inspection of the maximum VAF of functional TERT promoter mutations across samples and the number of different mutations found in a sample revealed the co-existence of mutations with different VAF in the same sample, revealing a landscape of strong convergent evolution affecting this genomic region in some donors (Extended Data Fig. 7b).

Extended Data Figure 7b

Extended Data Figure 7. Heterogeneity of clonal landscape across individuals

b) Maximum variant allele frequency detected for TERT promoter mutations in a sample vs the number of functional (see main text) TERT promoter mutations identified in a sample. The observation of different functional TERT promoter mutations in the same sample indicates the existence of convergent evolution of the TERT promoter mutation. This, in turn, suggests that the observation of mutations with large variant allele frequency may also represent convergent evolution rather than very large clones.

This figure exemplifies this for the dome and trigone samples of donor 26, where we identify two and four different TERT mutations with varying VAF.

9. Compared to previous studies, this study provides very limited new insights. Many of the findings have already been reported in earlier research on normal urothelium. For instance, the 13 positively selected genes identified here were previously described, and it is well established that TP53 and FGFR3 are more frequently mutated in bladder cancer than in normal urothelium. Furthermore, all the mutational signatures extracted in this study have been reported before, yet the smoking-related signature SBS92, identified in earlier studies, is notably absent. The lack of novel findings is likely attributable to the study's research strategy and design—specifically, the use of targeted sequencing limited to 16 genes in a small cohort of 42 individuals.

We respectfully disagree with this summary of our study. Two completely novel insights related to how the main cancer bladder risk factors shape clonal composition in normal bladder are reported: i) the sex bias of truncating driver mutations in four genes and ii) the association of TERT promoter driver mutations with smoking.

Furthermore, exploiting the thousands of driver mutations identified across 350,000 haploid genomes, we demonstrate the usefulness of natural saturation mutagenesis to pinpoint regions or individual sites of genes that are key for tumorigenesis and could guide personalized cancer medicine.

Minor points:

1. TERT is missing in Fig 4k while being mentioned in the main text.

There is a new Figure illustrating findings on TERT in the new version of the manuscript (Fig. 4)

2. I fail to see the point of displaying Figure 4A

In agreement with the reviewer's comment, this figure has been removed from the manuscript.

REFEREE COMMENTS

OUR RESPONSE

QUOTES FROM THE MANUSCRIPT OR SUPPLEMENTARY NOTES

Referees' comments:

Referee #1 (Remarks to the Author):

Summary:

The authors have reframed their paper to focus on their novel findings, and they have addressed my comments largely satisfactorily, with some remaining minor comments per below. The mathematical modeling estimating the depth needed to reach saturation mutagenesis I believe is novel and may be a fairly impactful contribution to the field, similar to the early studies that estimated how many samples would be needed to power future GWAS studies.

We thank the reviewer for their appreciation of our work.

1. Extended Data Fig. 2b: the similarit should be calculated as a cosine similarity.

The cosine similarity between the mutational profile observed across the normal urothelium samples and the whole-genome sequenced laser capture microdissection samples processed in Lawson et al. (2020) is indicated in the plot (Cosine similarity: 0.91).

2. Error rate estimates: the authors estimate this as the difference between the observed mutation rate in their samples vs the observed mutation rate in NanoSeq. The authors should consider (if feasible computationally for their pipeline) also calculating an error rate per Abascal et al's method using the single-strand calls (i.e. calls observed in only one of the two strands). Specifically, multiplying the frequencies of the reverse complement single-strand calls on a per-trinucleotide context basis, and summing across all trinucleotides.

We would like to clarify that we do not estimate the error rate as the difference between the observed mutation rate in our samples vs the observed mutation rate in NanoSeq. Instead, we compute it by subtracting the expected mutation rate in cord blood (obtained from expanded colonies of hematopoietic stem and progenitor cells) from that observed in three experiments. To avoid this confusion, this is now more clearly highlighted in Extended Data Figure 2a in the new version of the manuscript.

3. "Some genes found to be under positive selection in Lawson, et al (2020 Science) were not included in this study's panel. What was the reason for that?" -> I recommend adding the authors' explanation for gene choice for the panel to the methods section.

We thank the reviewer for this recommendation. We have edited the corresponding section in Methods, which now reads as follows,

We designed a panel including 10 genes identified in a previous study as being under positive selection in the normal urothelium and with more than 10 mutations across the 1,674 microbiopsies analyzed⁹. We added 5 genes that are frequently mutated in bladder tumors²⁹, and the TERT promoter, also known to be under positive selection in bladder carcinomas^{21,22}. This resulted in a panel containing the entire (or almost entire) coding region of 12 genes (ARID1A, NOTCH2, FOXQ1, CDKN1A, KMT2D, RB1, CREBBP, TP53, EP300, KDM6A, RBM10 and STAG2), 3 genes for which only selected regions were targeted due to clustering of cancer mutations in those regions and/or difficulties for capturing the full gene (PIK3CA, FGFR3, and KMT2C), and the TERT promoter^{9,58}. The panel was constructed by TwinStrand Biosciences (Seattle, WA) and it covered 111,876 bp, including 65,086 bp in coding regions and 46,790 bp in non coding regions (Extended Data Table 2; Supplementary Note 2).

4. Supplementary Note Figure 2.2: The authors should provide more information on the number of such brushes evaluated (ideally several) and would help to have some bar graph of the % for each sample rather than just an average.

We have provided detailed information about the cell counts of the different cell brushes evaluated and have made bar graphs to illustrate the findings. These results are now included in Supplementary Note 2 (copied below).

To formally assess the contribution of each cell type to the brushes, for each specimen we systematically evaluated consecutive fields of view (14 fields at 200x magnification) in the slides stained for CK7, actin, and CD45, and quantified the number of positive cells in each one (Supplementary Note Figure 2.3). Each specimen had more than 1600 cells evaluated, for a total of 8681 cells overall. None of the fields had any actin positive cell identified and the fraction of non-epithelial cells (all CD45 positive cells) was below 0.06 in all fields, with an overall mean of 0.030 across specimens (minimum 0.017, maximum 0.038). Overall, these results indicate that urothelial brushes are composed mostly of urothelial cells with minimal contribution of lymphocytes and no stromal contamination.

50-TR (1670 cells evaluated)					50-DO (4505 cells evaluated)					52-TR (2506 cells evaluated)				
Field of view (200x)	CK7 positive cells	Actin positive cells	CD45 positive cells	Non-epithelial cell fraction	Field of view (200x)	CK7 positive cells	Actin positive cells	CD45 positive cells	Non-epithelial cell fraction	Field of view (200x)	CK7 positive cells	Actin positive cells	CD45 positive cells	Non-epithelial cell fraction
1	94	0	4	0.043	1	289	0	15	0.052	1	240	0	5	0.021
2	135	0	6	0.044	2	325	0	6	0.018	2	175	0	3	0.017
3	106	0	5	0.047	3	256	0	11	0.043	3	181	0	2	0.011
4	110	0	3	0.027	4	319	0	13	0.041	4	169	0	1	0.006
5	87	0	2	0.023	5	333	0	12	0.036	5	174	0	4	0.023
6	130	0	6	0.046	6	384	0	7	0.018	6	211	0	5	0.024
7	128	0	5	0.039	7	351	0	10	0.028	7	205	0	3	0.015
8	116	0	2	0.017	8	296	0	14	0.047	8	196	0	5	0.026
9	121	0	4	0.033	9	285	0	13	0.046	9	233	0	3	0.013
10	118	0	6	0.051	10	279	0	11	0.039	10	238	0	5	0.021
11	95	0	2	0.021	11	350	0	18	0.051	11	215	0	5	0.023
12	132	0	5	0.038	12	309	0	10	0.032	12	226	0	2	0.009
13	119	0	2	0.017	13	290	0	15	0.052	13	na	na	na	na
14	122	0	5	0.041	14	276	0	8	0.029	14	na	na	na	na
mean	115	0	4.1	0.035	mean	310	0	11.6	0.038	mean	205	0	3.6	0.017

Supplementary Note Figure 2.3. Quantification of cell types in urothelial brushes. For each specimen, immunostaining for CK7, actin, and CD45 was evaluated in 14 consecutive fields of view using an objective of 20x, for a total magnification of 200x. Top panel lists the counts by cell type and the non-epithelial cell fraction for each field of view and each sample. Bottom panel quantifies the fraction of lymphocytes (CD45 positive cells) for each sample. Box plots display the quartiles with whiskers extending to the highest and lowest data points within 1.5 times the interquartile range.

5. The methods could use more clarification why rather than only using the medium confidence sets, the high and medium confidence sets of mutations weren't pooled for downstream analyses.

The high quality reads are actually contained within the medium quality reads. This is because reads are considered high quality when at least three copies of each DNA strand are present, and medium when at least two copies of each DNA strand are present, a condition which is fulfilled by all high quality reads. We have added a clarification to this effect in the section of Supplementary Note 3 where the three sets of reads are described.

6. Cross contamination analysis: The authors should more directly calculate/estimate what fraction of their final mutation calls could be due to cross-contamination based on their new germline/somatic contamination analysis. There is also a tool called VerifyBAMID2 that they could run similar to Abascal, et al, though it may not work well with the small panel size they have.

Following the reviewer's recommendation, we have more thoroughly explored potential cross-contamination across the samples in our cohort. We have highlighted one possible (although far from certain) case of cross-contamination and have decided to eliminate somatic mutations identified across samples that are private germline variants in other samples. This is described with more detail in the section of Supplementary Note 3 copied below.

Cross-contamination analysis

We explored the level of potential contamination between samples in the cohort. To this end, we identified all germline variants in each of the samples that were at a VAF > 0.3. We then compared these sets of germline variants between samples to identify, for each pair of samples, which mutations are germline in only one of them. These are the variants that are useful for identifying hypothetical contamination between samples. The only mutation that we excluded from the analysis of contamination is a mutation in one hotspot of the TERT promoter, which was identified as a SNP across several samples and was detected as a somatic mutation with a variant allele frequency above 0.01 in 8 samples. This mutation is likely a recurrent somatic mutation that is present at high VAF in some samples. For each pair of samples we then compared all the non-germline variants to the set of susceptibly contaminating variants and computed the proportion of how many of those are present in each sample.

We reasoned that if a contamination from one sample (contaminant) was present in another (contaminated), we expected to find all SNPs of the contaminant within the contaminated, either as already germline or as somatic mutations. This would be consistent with a small portion of DNA from the contaminant present in the contaminated. Moreover, all the SNPs of the contaminant would be observed at roughly the same variant allele frequency in the

contaminated. Therefore, we next looked at the fraction of germline variants across samples that are identified as somatic mutations across other samples (Supplementary Note Fig. 3.9).

We found one sample (the dome of donor 38) with more than half (0.6) of the SNPs identified in another donor (9) identified as somatic mutations. Yet, there are 4 private SNPs of donor 9 that are not identified as somatic mutations in this sample, casting doubt on whether a sample from donor 9 is really a contaminant of the sample from donor 38. This is still a possibility, since the missing 4 SNPs could be due to chance in the detection. While this is the only case in which cross-contamination could conceivably have occurred in the cohort, we decided to filter out all somatic mutations detected across samples that are germline variants in another donor, except the aforementioned TERT variant, identified recurrently across samples and with a variant allele frequency inconsistent with a contamination.

Supplementary Note Figure 3.9. Germline variants identified as somatic mutations in other samples. The values in the heatmap represent the proportion of germline variants of a sample (in the corresponding column) that have been identified as somatic mutations in another sample (in the row). The proportion is computed taking into account the number of non-shared germline variants to obtain a fair comparison.

General writing comments:

- These sentences are written as a summary of the results, which is somewhat redundant with the beginning of the results section: "Ultra-deep sequencing (~5,000x per sample, amounting to ~350,000 haploid genomes in aggregate) in these genomic regions has proved very powerful to accurately quantify selection at the sample level, and to count and characterize the mutations of each gene driving clonal expansions. The normal urothelium of 71 samples obtained from 41 individuals, profiled at this sequencing depth proved key to accurately compute the magnitude

of positive selection for each gene at the sample level, and carry out a regression analysis that uncovered this sex bias."

Following the reviewer's recommendation, we have now edited the introductory section to avoid as much as possible the overlap with the results sections.

- "game changer": I would rephrase this less formal term

Following the reviewer's suggestion, we have rephrased this sentence as follows,

We also hypothesized that the capacity to identify the driver mutations of thousands of clones in a human tissue could dramatically increase our ability to uncover the functional effect of all mutations in genes relevant in tumorigenesis.

- "We explore this postulate with the data of the normal bladder ultradeep sequencing uncovering an unprecedented power to quantify positive selection at site resolution." -> the grammar and lack of commas is a bit confusing.

Following the reviewer's suggestion, we have rephrased this as follows,

We explore this postulate with the data of the normal bladder ultradeep sequencing. As a result, we have uncovered an unprecedented power to quantify positive selection at site resolution.

- The paragraphs beginning "To explore potential differences in the magnitude of positive selection" and "A correct comparison of the association of sex with the clonal landscape" address sex biases, but then there is a separate section about sex biases. So sex bias is introduced into disparate sections. This structure will be confusing to readers.

We agree with the reviewer and, following their recommendation, we have extensively edited and streamlined this section of Results.

- The paragraph beginning "The number of detected clones driven" doesn't appear to belong in the sex bias section, unless I'm confused about its purpose.

We agree with the reviewer and we have moved this paragraph to the previous section, in accordance with their recommendation.

- The grammar of this sentence is unclear: "That many somatic tissues..."

Following the reviewer's suggestion, this sentence has been edited to,

The reasons behind the wide heterogeneity in clonal landscape observed across individuals' normal tissues^{6-11,38,39} and the potential role of known cancer risk factors are not understood. In this study we demonstrated the suitability of ultradeep DNA duplex sequencing^{12,13}

(approximately 400,000 haploid genomes) to accurately measure the magnitude of positive selection on each of 16 genes in 79 individual samples (Supplementary Note 6).

Referee #2 (Remarks to the Author):

In the current revision, the authors have made substantial effort in improving the robustness of their analyses. The manuscript rigorously demonstrates the technical quality of the ultra-deep targeted sequencing and analysis. As such, I accept that for their purpose of identifying sex bias in selection, their methodology is sufficient and need not be considered as a pilot.

We thank the reviewer for their appreciation of our analyses and the rigorousness of our results.

However, I am still not convinced by the significance of the sex bias in selection for mutation in four genes. Firstly, the cohort is still relatively small with just 41 individuals. While the difference is statistically significant in this cohort, it is not certain that this can be generalised to the general population, particularly given most donors appear to be sampled due to incidental health conditions. Second, even if the sex bias is generalisable, without resolving the mechanism, it could just be an association representing some other factor not captured by the study. For example, even though smoking has been controlled, it is well known that males are generally heavier smokers than females (in terms of cigarettes per day or packs/year), could the amount of smoking contribute to the observed bias? Even with the bias shown in non-smokers, the representativeness of the cohort (5 males versus 8 females) remains questionable.

We would like to clarify that donors were selected for not having any overt bladder pathology. Actually, as described in Methods and Supplementary Note 2, individuals were included if they had no history or evidence (upon autopsy) of gross bladder pathology or bladder cancer. Review of the autopsy report and H&E slides revealed active or chronic inflammation with hemorrhages and mucosal erosions in 3 individuals, who were excluded from the study. We agree with the reviewer that the significant sex bias of clonal landscape observed in our cohort will need to be explored across larger and more diverse cohorts. We acknowledge this point in the Discussion section. We also agree that the reasons behind this bias require further investigation. In our study we have controlled for potential confounders that had been recorded for the donors in the cohort.

Following the point raised by the reviewer about the possibility of smoking as a confounding factor, we have now carried out an experiment accounting for potential differences in the intensity of smoking across males and females. Specifically –as this information was not available for the donors in our cohort– we calculated the bias in smoking intensity (estimated packs per year consumption) between males and females in the UK Biobank cohort. We found that, as a median, males smoke 1.32 more packs than females. Assuming that this rate is the same in our cohort, we carried out a new multivariate regression adjusting for this bias in smoking rather than for a binary annotation of ever smokers and never smokers. The observed sex bias in the magnitude of positive selection was maintained. This experiment and its results are described as one more control of the regressions in Supplementary Note 9.

We have also carried out a new analysis of sex bias in the GENIE cohort of bladder urothelial carcinomas (n=2,965) and have found that truncating mutations in RBM10 and CDKN1A are more frequent in tumors from males than females. This result is consistent with our findings in normal urothelium and provides further support to the sex bias in clonal selection in these genes. This analysis and its results are described in Supplementary Note 11.

Minor comment – regarding Figure 1b, even though I understand that it is intended for descriptive purposes, the comparison is somewhat unusual, as the number of mutations have different meanings for the two cohorts. If the purpose is to demonstrate the superiority of the technique, a comparison with Lawson et al. might be more informative and appropriate.

The purpose of the figure is to illustrate that probing normal tissues using ultradeep sequencing yields a much higher number of mutations in the genes of the panel than sequencing hundreds of tumors. While many clones are identified in a normal tissue with our approach, only one clone is probed in every tumor sequenced sample. This is why we compare with tumors rather than with the laser capture microdissection approach carried out by Lawson et al.

Referee #2 (Remarks on code availability):

No code has been made available.

Referee #3 (Remarks to the Author):

I am satisfied with the revision, although it is still puzzling to me why TERT promoter mutations are strongly associated with smoking in normal tissue, but not the case for tumors. I think this paper is a good starting point.

We thank the reviewer for their assessment of our responses.

We agree that it is interesting and puzzling that TERT promoter mutations, although strongly associated with smoking in normal tissues, do not seem to be in higher frequency in bladder tumors from smokers. To clarify this, we did an analysis with a cohort of 1011 bladder carcinomas sequenced using the MSK-IMPACT panel, which includes the TERT promoter, and confirm that, even though there is a slight increase in frequency of TERT promoter mutations among smokers, this difference is not very big and certainly not significant. It is important to note that TERT promoter mutations are very frequent in bladder cancer (about 70% of tumors bear them).

Given that smoking is also a major risk factor for lung cancer, we performed a similar analysis in a cohort of lung tumors. In this case, where TERT promoter mutations are much less frequent overall, we observed a significantly higher frequency in smokers (2.1%) compared to non-smokers (0.9%). This difference remains statistically significant after adjusting for confounding factors such as age. These results are detailed in Supplementary Note 11.

TERT activating mutations and smoking in tumors

We next asked if activating mutations in the TERT promoter across bladder tumors are associated with smoking, as observed across normal urothelium samples. Leveraging data from 1011 bladder carcinomas sequenced using the MSK-IMPACT panel (which includes the TERT promoter), we found no significant association between TERT promoter mutations and a history of smoking, although we observed a slight increase in the frequency of TERT promoter mutations across smokers (Supplementary Note Figure 11.3)⁵³. It is important to bear in mind that TERT promoter mutations are very frequent (close to 70% in this cohort) across bladder tumors, irrespective of the smoking status of patients. Importantly, a significant association was found between age and the presence of TERT promoter mutations across these tumors.

Supplementary Note Figure 11.3. TERT promoter mutated tumors association with smoking history and age in a bladder cancer cohort.

From left to right, comparison of the frequency of tumors with TERT promoter mutations across ever smokers and never smokers, comparison of the frequency of tumors with TERT promoter mutations across patients in different age groups, distribution of the age of ever smokers and never smokers at the moment of sequencing of their tumors, probability of TERT promoter mutation in tumors from ever smokers and never smokers depending on their age, and multivariate regression analysis of TERT promoter mutations on the age and smoking status of patients. In the first and second panel from the left, the total number of tumors belonging to each group is indicated below the x axis ticks. The specific number of tumors belonging to the pTERT mutated subset per group is shown on top of each bar.

We repeated this analysis in a cohort of lung tumors, since smoking is the main risk factor for lung cancer (MSK-CHORD)⁵⁴. In this case, we identified a significant association of the presence of TERT promoter mutations with the smoking status of patients. In contrast to bladder cancer, where pTERT mutations are found in 70-80% of tumors², pTERT mutations are only found in a small percentage of lung cancers (<5%). However, those with pTERT mutation tend to be smokers and also of older age, as observed for bladder cancer (Supplementary Note Figure 11.4).

Supplementary Note Figure 11.4. *TERT promoter mutated tumors association with smoking history and age in a lung cancer cohort.*

From left to right, comparison of the frequency of tumors with TERT promoter mutations across ever smokers and never smokers, comparison of the frequency of tumors with TERT promoter mutations across patients in different age groups, distribution of the age of ever smokers and never smokers at the moment of sequencing of their tumors, probability of TERT promoter mutation in tumors from ever smokers and never smokers depending on their age, multivariate regression analysis of TERT promoter mutations on the age and smoking status of patients. In the first and second panel from the left, the total number of tumors belonging to each group is indicated below the x axis ticks. The specific number of tumors belonging to the pTERT mutated subset per group is shown on top of each bar.

Referee #4 (Remarks to the Author):

I thank the authors for their responses. My noteworthy concerns during the first round of reviews were (1) the sample size was insufficient for a comprehensive investigation of inter-individual variation in the mutational landscape and clonal selection in normal urothelium; (2) the limited number of targeted genes constrained the potential for new discoveries; (3) contamination from other cell lineages in cytobrush samples; (4) higher-than-expected error rates of duplex sequencing.

In the revision, the authors extensively revamped the manuscript, shifting the primary focus to sex differences in mutation and selection in normal urothelium. Given the higher incidence of bladder cancer in males for largely unknown reasons, it is important for understanding whether mutagenesis and clonal selection differ at the earliest stages of cancer initiation.

We thank the reviewer for their appreciation of our revised manuscript.

I have some follow-up comments and questions:

Regarding concern (1), although I initially expected the authors to expand the cohort in the revision, it is an acceptable move to shift the focus in revised manuscript to sex bias, a question that the current cohort size appears to have enough power to address (as shown in the revision). One of the major findings is that the positive selection on truncating mutations of RBM10, CDKN1A, STAG2 and ARID1A is significantly higher among males. Is the similar sex difference in these four genes seen in bladder cancer? Can authors speculate on how this finding is related to the increased risk of bladder cancer in males? Are there any other mutational parameters showing sex differences, e.g., overall mutational burden, mutational burden of certain mutational signatures, total driver mutation burden, etc?

These are very interesting questions. We have now explored more thoroughly to address the reviewer's comment. Leveraging a cohort of 2965 bladder tumor whole-exomes part of the GENIE initiative, we detected a significant sex bias in RBM10 and CDKN1A similar to those observed across normal tissues in our analysis. These findings are now reported in Supplementary Note 11.

Sex bias in bladder cancer

We asked the question of whether the sex difference observed in the clonal selection of mutations in RBM10, CDKN1A, ARID1A and STAG2 also occurs in bladder tumors. To count with a big enough cohort, we used the exome data of 2,965 bladder urothelial carcinomas (BLCA) obtained from the GENIE project⁵² (Supplementary Note Figure 11.1). We classified tumors in mutated and not mutated for each gene analyzed in the normal bladder study and ran univariate logistic regressions with sex as the dependent variable and the gene mutational status as the independent variables (with the exception of FOXQ1, which does not appear mutated in this cohort). For those genes with a significant association in the univariate analysis we ran a multivariate regression to account for the confounding effect of age. These analyses demonstrated that males have a significantly higher frequency of bladder tumors with mutations in RBM10 and CDKN1A than females (Extended Data Fig. 8a,b and Supplementary Note Figure 11.1).

Supplementary Note Figure 11.1. Analysis of sex bias in a cohort of bladder tumors

a) Percentage of mutated tumors for truncating (above) and missense (below) mutations across males and females for the 15 coding genes sequenced in the normal bladder study. FOXQ1 is not shown because no mutation was identified in the cohort. Absolute number of tumors belonging to each category is shown on top of each bar.

b) Association of sex with the presence of truncating (above) and missense (below) mutations in 14 of the genes sequenced in the normal bladder study. The plots show the effect size and confidence intervals of univariate logistic regressions for all genes except RBM10 and CDKN1A truncating, for which the results

of multivariate regressions accounting for age are shown (q-value < 0.05 indicated with a dark outer circumference).

Regarding the question about other variables that show associations with sex, we found no significant association with overall mutation burden (Extended Data Table 6). However the overall dN/dS truncating (that is, the number of truncating driver mutations across all genes in the panel) is significantly higher in males, reflecting the overall increase in dNdS in the 4 genes with sex differences.

Speculating about the relationship of the bias observed in the dN/dS truncating of RBM10, ARID1A, CDKN1A and, to a lesser extent, STAG2 across males and females with the increased risk of bladder cancer observed in males is very challenging. One possibility, which we raise in Supplementary Note 11 –through the comparison of the mutation frequency of the genes analyzed here across normal urothelium and bladder cancer– is that driver mutations of genes such as RBM10 and ARID1A detected in tumors may reflect the frequency of early clonal expansions driven by these mutations in the normal tissue. In other words, it is possible that the increased number/size of clones driven by these –and perhaps other– genes that we observed in the normal urothelium of males increase the risk of bladder tumors just by providing an increase in the number of cells that may be sensitive to further cancer driver mutations. This and other hypotheses should be contrasted by studies with larger cohorts, and potentially serial samples if non-invasive procedures to collect urothelial cells, such as urine or lavages, can be established.

For the concern (2), the authors claim that the aim of the study is not to identify new genes under positive selection, but rather to understand sex differences in the selection of known genes. This is reasonable, but it still feels like a missed opportunity that the study doesn't leverage the available samples and technology to explore additional findings. For the selection of these 16 genes, beyond sex differences, did authors identify anything that potentially associated with inter-donor variation in selection of mutations in these genes, as obvious inter-donor variation can be seen in Fig 3A regardless of sex?

This is an important question. As explained in the manuscript, we do identify a significant association of the magnitude of positive selection across 13 genes and age, as well as with smoking in the case of potentially activating mutations in the TERT promoter. The complete picture of the associations between risk factors and clonal landscape is presented in Extended Data Table 6. While other significant associations do appear, there are none as systematic and strong as the ones described in the manuscript.

For the concern (3), the authors provided representative IHC staining images showing the cytobrush sampling predominantly collects urothelial cells with minimal contamination from other lineages. This is a stronger piece of evidence than previously included. How many samples were subjected to IHC staining? Can the authors provide a range of contamination besides representative IHC images.

We have provided detailed information about the cell counts of the different cell brushes evaluated and have made bar graphs to illustrate the findings. These results are now included in Supplementary Note 2 (see above, response to point 4 by reviewer 1).

Regarding concern (4), the authors re-evaluated the error rate of duplex sequencing in this study by sequencing 3 cord blood samples with the same gene panel. They observed the expected average mutation burden for newborn blood, which supports the validity of their approach. However, as shown in Extended Data Fig. 2, the error rate of duplex sequencing remains higher than that of NanoSeq. Have the authors performed NanoSeq on these same cord blood samples for comparison? I understand that this may be challenging, given that the NanoSeq protocol is still in-house.

We estimated that implementing NanoSeq with the same level of quality as the authors of the approach is a challenging endeavor, as noted by the reviewer. This is why we decided to directly assess the error rate of the DNA duplex sequencing technology employed in our study, and to thoroughly evaluate its potential implications on our results. This study, as presented in Supplementary Note 8, determined that the metric used to measure positive selection is very tolerant to the rate of errors calculated for the technology.